# TUAP: TARGETED UNIVERSAL ADVERSARIAL PERTURBATIONS FOR CLIP

## ABSTRACT

As Contrastive Language-Image Pretraining (CLIP) models are increasingly adopted in a wide range of downstream tasks and large Vision-Language Models (VLMs), their vulnerability to adversarial attacks has attracted growing attention. In this work, we examine the susceptibility of CLIP models to Universal Adversarial Perturbations (UAPs). Unlike existing works that focus on untargeted attacks in a white-box setting, we investigate targeted UAPs (TUAPs) in a black-box setting, with a particular emphasis on transferability. In TUAP, the adversary can specify a targeted adversarial text description and generate a universal $L_\infty$-norm-bounded or $L_2$-norm perturbation or a small unrestricted patch, using an ensemble of surrogate CLIP encoders. When TUAP is applied to different test images, it can mislead the image encoder of unseen CLIP models into producing image embeddings that are consistently close to the adversarial target text embedding. We conduct comprehensive experiments to demonstrate the effectiveness and transferability of TUAPs. This universal transferability extends not only across different datasets and models but also to downstream models, such as large VLMs including OpenFlamingo, LLaVA, MiniGPT-4 and BLIP2. TUAP can mislead them into generating responses that contain text descriptions specified by the adversaries. Our findings reveal a universal vulnerability in CLIP models to targeted adversarial attacks, emphasizing the need for effective countermeasures.

## 1 INTRODUCTION

Contrastive Language-Image Pretraining (CLIP) is a popular technique that learns aligned multi-modal representations from text-image pairs via contrastive learning (Radford et al., 2021). Models pre-trained by CLIP on web-scale datasets have been employed to boost the performance in downstream applications such as image generation (Ramesh et al., 2022), robotics (Ahn et al., 2022), anomaly detection (Jeong et al., 2023; Zhou et al., 2024), and medical applications (Eslami et al., 2023). CLIP models also play a central role in recent advancements in large Vision Language Models (VLMs) (Awadalla et al., 2023; Koh et al., 2023; Wang et al., 2023; Bai et al., 2023; Karamcheti et al., 2024; Jiang et al., 2024; Tong et al., 2024), providing image encoders for their visual capability. E.g., Flamingo (Alayrac et al., 2022), LLaVA (Liu et al., 2023), BLIP2 (Li et al., 2023a) and MiniGPT-4 (Zhu et al., 2024) were trained by aligning CLIP image encoders with Large Language Models (LLMs) (Zhang et al., 2022b; Hoffmann et al., 2022; Chiang et al., 2023). The CLIP image encoder's zero-shot generalization ability enables large VLMs to excel across a wide range of visual-language tasks. Consequently, the safety of CLIP models, particularly their widely adopted image encoder, has become a significant concern in the community, especially regarding potential adversarial attacks.

Deep neural networks are well-known for their vulnerability to adversarial examples—test instances deliberately perturbed by an attacker to maximize output errors (Szegedy et al., 2014; Goodfellow et al., 2015; Madry et al., 2018). This vulnerability has been extensively studied, particularly in the context of image classification (Croce et al., 2021). Recent studies (Mao et al., 2023; Zhao et al., 2023; Schlarmann et al., 2024) have shown that the CLIP image encoder is similarly susceptible to adversarial attacks, which negatively impact large VLMs trained to align with the same encoder. Zhou et al. (2023) investigated the CLIP image encoder's vulnerability to Universal Adversarial Perturbations (UAPs) (Moosavi-Dezfooli et al., 2017), where a single perturbation can be universally applied to any test image, causing the model to make arbitrary mistakes. These findings raise signif-

icant safety concerns regarding the deployment of CLIP-based downstream models and large VLMs in real-world applications.

Different from existing works, this paper investigates the vulnerability of the CLIP image encoder to *targeted UAPs (TUAPs)*, with a particular focus on black-box transferability. Unlike standard UAPs, which cause arbitrary errors, TUAPs aim to control the model's output more precisely. For instance, an adversary could generate a TUAP using the phrase "oranges have gone bad" and apply it to any image to deceive chatbots (VLMs) used by online grocery merchants for processing refunds. Similarly, a pattern like "the image contains safe contents" could be applied to any toxic images to bypass CLIP encoder used for harmful content detection on social media platforms. This makes targeted attacks more controllable and appealing to real-world adversaries. Previous works on TUAPs have mainly focused on image classification models (Poursaeed et al., 2018; Benz et al., 2020; Zhang et al., 2020; Weng et al., 2024), where the target is an adversarial class selected from a fixed set of options. In contrast, TUAPs can target any text description when applied to CLIP. Different from Zhou et al. (2023), our study focuses on the black-box scenario, where the adversary lacks gradient information about the victim model. We explore transferability by assuming the adversary can use surrogate models to generate TUAPs, which are then applied to various victim CLIP encoders and downstream models, including large VLMs. Generating TUAPs in a black-box setting is challenging because the same perturbation, when applied to any image, must effectively mislead different models across various tasks to produce a targeted response specified by the attacker.

In this work, we propose to generate TUAPs using an ensemble of surrogate CLIP image encoders and then transfer the generated TUAPs to attack unseen models, including other CLIP encoders and large VLMs. We use a malicious text description as the target and introduce 3 types of TUAPs to minimize the similarity between the adversarial image embedding and the target text embedding: (1) *unrestricted adversarial patch*, (2) $L_\infty$*-norm bounded perturbation*, and (3) $L_2$*-norm perturbation*. We follow the original UAP method (Moosavi-Dezfooli et al., 2017) by accumulating these perturbations over multiple images to ensure universality.

Through extensive experiments, we demonstrate that our TUAP can achieve strong black-box cross-model, cross-dataset, and cross-task adversarial transferability. Additionally, we conduct comprehensive ablation studies to provide deeper insights into the vulnerabilities of CLIP encoders and large VLMs to TUAP. The patterns generated by TUAPs suggest that the vulnerability of CLIP encoders (and its downstream models) is largely attributable to its superior *concept blending* capability (Ge & Parikh, 2021; Kazemi et al., 2024)—the ability to generate visual representations of an image by combining arbitrary concepts. Given the widespread availability of powerful pre-trained CLIP encoders as open-source models (Radford et al., 2021; Ilharco et al., 2021), adversaries can exploit these models to generate highly transferable TUAPs. Our work uncovers a new type of safety vulnerability in multi-modal pretraining and VLMs.

In summary, our main contributions are:

- We study the vulnerability of CLIP to targeted UAPs and propose a black-box attack method TUAPs that leverages an ensemble of surrogate CLIP image encoders to generate three distinctive types of targeted UAPs.

- We conduct comprehensive evaluations demonstrating the universal transferability of TUAPs across different images, victim models, and tasks. We find that transferability scales with the use of multiple surrogate ensemble models. This transferability is closely correlated with the attack success rate on various pre-trained encoders and downstream models.

- We perform both quantitative and qualitative assessments to reveal the universal adversarial threat posed by TUAPs to large VLMs, including OpenFlamingo, LLaVA, MiniGPT-4, and BLIP2. Our results show that TUAPs can deceive large VLMs into generating harmful responses consistent with adversarially specified target text descriptions.

## 2 RELATE WORK

**Contrastive Language-Image Pretraining (CLIP).** CLIP (Radford et al., 2021) is a popular self-supervised framework that can pre-train large-scale language-vision models on web-scale text-image pairs via contrastive learning (Chopra et al., 2005; Oord et al., 2018; Chen et al., 2020b). Models

pre-trained by CLIP have demonstrated superior zero-shot generalization capability in a wide range of downstream tasks (Palatucci et al., 2009; Lampert et al., 2009) and are shown to be more robust against common corruptions (Hendrycks & Dietterich, 2019; Fang et al., 2022; Cherti et al., 2023; Tu et al., 2023). A number of works have been proposed to improve the performance of original CLIP using uncurated noisy dataset (Jia et al., 2021), improved training recipe (Sun et al., 2023; Li et al., 2023b), masking images (Li et al., 2023d), shorter token sequence (Li et al., 2023c), self-supervision (Li et al., 2022b) or sigmoid loss (SigLIP) (Zhai et al., 2023). It has been found that one of the main contributing factors to the success of CLIP is its training data (Xu et al., 2024). In parallel to CLIP, multimodal pretraining can be achieved using various objectives, such as image-text matching, masking, and autoregressive generation (Li et al., 2021; 2022a; Singh et al., 2022; Yu et al., 2022; 2023; Kwon et al., 2023). This paper focuses specifically on CLIP and its variants due to their widespread adoption in downstream applications.

**Adversarial Attacks.** The vulnerability of DNNs to adversarial attacks (examples) has been extensively studied on image classification models (Szegedy et al., 2014; Goodfellow et al., 2015; Carlini & Wagner, 2017; Madry et al., 2018; Zhang et al., 2019; Ilyas et al., 2019; Wang et al., 2019; 2020), under two main attack settings: white-box and black-box. In the white-box setting, the adversary has full knowledge of the victim model including its architecture and parameters, while in the black-box setting this information is not avaaialble to the adversary. In this case, the attacker can construct query-based attacks to exploit the input-output response of the victim model (Ilyas et al., 2018; Andriushchenko et al., 2020) or leverage surrogate models to construct transfer attacks (Papernot et al., 2016; Tramèr et al., 2017; Liu et al., 2017; Dong et al., 2018; Xie et al., 2019; Dong et al., 2019; Wu et al., 2020). Arguably, black-box attacks are more realistic and challenging, as commercial models are often kept secret to the end users, and in this case the gradient information of the victim model is unavailable. Between the two types of black-box attacks, transfer attacks are more practical, stealthy, and cost-effective, as they do not need to launch a huge number of suspicious and costly queries to the victim model (Chen et al., 2020a; Wang et al., 2024b). Specifically, transfer attacks generate adversarial examples based on a surrogate model and then directly feed the generated adversarial examples to attack the black-box victim model. The ensemble of different surrogate models is an effective approach to boost transferability (Xiong et al., 2022; Chen et al., 2024). This can be achieved by averaging the loss (Liu et al., 2017; Dong et al., 2018) or combining the classifier's logits (Dong et al., 2018).

**Adversarial Attacks on Multi-modal Models.** Recent works in can be categorized as to whether they craft the perturbation in the vision domain (Zhao et al., 2023; Bailey et al., 2023; Dong et al., 2023; Schlarmann & Hein, 2023; Qi et al., 2024; Luo et al., 2024), the language domain (Zou et al., 2023), or both (Zhang et al., 2022a; Lu et al., 2023; He et al., 2023; Shayegani et al., 2024; Lu et al., 2024; Gao et al., 2024). For VLMs, the vision domain has been shown to be easier to fool (Carlini et al., 2023). An attacker could manipulate the image (Bailey et al., 2023; Qi et al., 2024; Shayegani et al., 2024) to perform a jailbreak attack that can bypass a model's safety alignment. Unlike these works which focus on VLMs, our focus in this work is the zero-short robustness of CLIP's image encoder. This problem has been examined in image-specific perturbation (Mao et al., 2023; Zhao et al., 2023; Schlarmann et al., 2024; Wang et al., 2024a) and untargeted image-agnostic UAP (Zhou et al., 2023; Zhang et al., 2024). Our work follows this line of studies that focus on the zero-shot robustness and investigate its impact on downstream applications. However, unlike existing works, our focus is the TUAP. Additionally, we investigate a more realistic black-box transfer attack setting, where the adversary uses surrogate CLIP image encoders to produce perturbations that transfer across different victim CLIP image encoders and large VLMs (Awadalla et al., 2023; Liu et al., 2023; Li et al., 2023a; Zhu et al., 2024).

## 3 PROPOSED ATTACK

In this section, we first introduce the training objective of CLIP and then present our proposed method for generating TUAPs.

### 3.1 TRAINING OBJECTIVE OF CLIP

CLIP (Radford et al., 2021) learns a joint embedding of images and texts. In such a way, the model can learn from web-scale data without using human annotations. This allows CLIP models

to carry out arbitrary image classification tasks without specifying the classes in the training set. This is known as zero-shot classification. Given an image-text dataset $\mathbb{D} \subset \mathcal{X} \times \mathcal{T}$ that contains pairs of $(\boldsymbol{x}_i, \boldsymbol{t}_i)$, where $\boldsymbol{x}_i$ is an image, and $\boldsymbol{t}_i$ is the associated descriptive text. An image encoder $f_I : \mathcal{X} \mapsto \mathbb{R}^d$ and a text encoder $f_T : \mathcal{T} \mapsto \mathbb{R}^d$. We use $f$ to denote the pair of image encoder $f_I$ and text encoder $f_T$. The CLIP model projects the image and text to a joint embedding space $\mathbb{R}^d$. The image embedding can be obtained by $\boldsymbol{z}_i^x = f_I(\boldsymbol{x}_i)$ and the text embedding is $\boldsymbol{z}_i^t = f_T(\boldsymbol{t}_i)$. For a given batch of $N$ image-text pairs $\{\boldsymbol{x}_i, \boldsymbol{t}_i\}_{i=1}^N$, CLIP adopts the following training loss function:

$$-\frac{1}{2N} \sum_{j=1}^N \log \frac{\exp(\mathrm{sim}(\boldsymbol{z}_j^x, \boldsymbol{z}_j^t)/\tau)}{\sum_{k=1}^N \exp(\mathrm{sim}(\boldsymbol{z}_j^x, \boldsymbol{z}_k^t)/\tau)} - \frac{1}{2N} \sum_{k=1}^N \log \frac{\exp(\mathrm{sim}(\boldsymbol{z}_k^x, \boldsymbol{z}_k^t)/\tau)}{\sum_{j=1}^N \exp(\mathrm{sim}(\boldsymbol{z}_j^x, \boldsymbol{z}_k^t)/\tau)},$$

where $\tau$ is a trainable temperature parameter, and $\mathrm{sim}(\cdot)$ is a similarity measure. The first term in the above objective function contrasts the images with the texts, while the second term contrasts the texts with the images.

## 3.2 Targeted Universal Adversarial Perturbation (TUAP) on CLIP

Our method is a form of *embedding space attack* that aims to deceive the encoder in the embedding space. The adversary can specify any descriptive text $\boldsymbol{t}_{adv}$. Our objective is to construct a universal adversarial function $A(\cdot)$ that is capable of transforming any image $\boldsymbol{x} \in \mathbb{D}$ into an adversarial version $\boldsymbol{x}' = A(\boldsymbol{x})$ by using the same adversarial noise or patch to achieve the following objective:

$$\arg\min_A \mathbb{E}_{(\boldsymbol{x}) \sim \mathbb{D}} \mathrm{sim}(f_I(\boldsymbol{x}'), f_T(\boldsymbol{t}_{adv})), \tag{1}$$

where $f_I$ could be any victim image encoder. Our overall objective is to find a function $A(\cdot)$ that can make the embedding of the adversarial version of an image close to the target text embedding. In the following, we introduce three types of TUAPs: 1) **a small adversarial patch** (Brown et al., 2017), 2) $L_\infty$-**norm bounded perturbation**, and 3) $L_2$-**norm perturbation**. Note that for each target descriptive text $\boldsymbol{t}_{adv}$, there is a unique perturbation or patch associated with it.

**Adversarial patch**. For the unrestricted adversarial patch attack, we construct the adversarial example using the following:

$$\boldsymbol{x}' = A(\boldsymbol{x}) = \boldsymbol{m} \odot \Delta + (1 - \boldsymbol{m}) \odot \boldsymbol{x}, \tag{2}$$

where $\boldsymbol{m} \in [0, 1]^{w \times h}$ is a learnable 2D input mask that does not include the color channels, $\Delta \in [0, 1]^{3 \times w \times h}$ is the universal adversarial pattern, and $\odot$ is the element-wise multiplication (the Hadamard product) applied to all the channels.

We optimize the following objective to generate a targeted universal patch attack:

$$\arg\min_{\boldsymbol{m}, \boldsymbol{\Delta}} \mathbb{E}_{(\boldsymbol{x}) \sim \mathbb{D}'} \mathrm{sim}(f_I'(\boldsymbol{x}'), f_T'(\boldsymbol{t}_{adv})) + \alpha \|\boldsymbol{m}\|_1 + \beta(TV(\boldsymbol{m}) + TV(\boldsymbol{\Delta})), \tag{3}$$

where $\mathbb{D}'$ is a surrogate dataset, $f_I'$ and $f_T'$ are the surrogate image encoder and text encoder, $\boldsymbol{x}'$ follows Equation 2, $TV(\cdot)$ is the total variation loss, and the $\|\cdot\|_1$ is the $L_1$ norm. $\alpha$ and $\beta$ are two hyperparameters to balance the two loss terms. While the patch attack is unrestricted, we set a soft constraint that the patch has to be as small as possible. The $L_1$ norm ensures that when the adversarial patch is added to the image, the patch is small and hard to notice. The total variation loss ensures the patch pattern and the mask are smooth.

$L_\infty$-**norm bounded perturbation**. For the $L_\infty$-norm bounded attack, we construct the adversarial example using the following:

$$\boldsymbol{x}' = A(\boldsymbol{x}) = \boldsymbol{x} + \boldsymbol{\delta}, \quad \|\boldsymbol{x} - \boldsymbol{x}'\|_\infty < \epsilon, \tag{4}$$

where $\boldsymbol{\delta}$ is the universal perturbation vector. To generate a universal perturbation for $L_\infty$-norm bounded attack, we optimize the following objective:

$$\arg\min_{\boldsymbol{\delta}} \mathbb{E}_{(\boldsymbol{x}) \sim \mathbb{D}'} \mathrm{sim}(f_I'(\boldsymbol{x}'), f_T'(\boldsymbol{t}_{adv})), \tag{5}$$

where $\boldsymbol{x}'$ follows Equation 4.

$L_2$**-norm perturbation**. For the $L_2$-norm perturbation, we optimize the following objective:

$$\arg \min_{\boldsymbol{\delta}} \mathbb{E}_{(\boldsymbol{x}) \sim \mathbb{D}'} \text{sim}(f'_I(\boldsymbol{x} + \boldsymbol{\delta}), f'_T(\boldsymbol{t}_{adv})) + c \cdot \|\boldsymbol{\delta}\|_2, \tag{6}$$

where the $\boldsymbol{\delta}$ is the perturbation and $c$ is a hyperparameter that balance two loss terms. The universal adversarial function for $L_2$-norm perturbation $A(\boldsymbol{x}) = \boldsymbol{x} + \boldsymbol{\delta}$. While the perturbation is not bounded, we use the $L_2$-norm to ensure the perturbation is small.

**Surrogate ensemble.** Our objective is to construct TUAP to be universally effective on different images and CLIP models. Equation 3, 5 and 6 only considered the universal transferbility to different images. The transferability should not only be limited to different images but also be effective for different victim unseen CLIP image encoders as well as downstream models, such as large VLMs. However, using a single surrogate model $f'$ (in Equation 3, 5 and 6) might limit its transferability, which could depend on the architectures, training loss functions, and pretraining datasets between the surrogate model $f'$ and victim model $f$. To improve the transferability, we consider an ensemble over a set of surrogate models $f'_i \in F' = \{f'_1, \cdots, f'_k\}$. We optimize the following objective function:

$$\arg \min \mathbb{E}_{(\boldsymbol{x}) \sim \mathbb{D}'} \frac{1}{k} \sum_{i=1}^{k} \mathcal{L}(f'_i, \boldsymbol{t}_{adv}, A, \boldsymbol{x}). \tag{7}$$

Without loss of generality, we use $\mathcal{L}$ to denote the objective function for the patch perturbation (Equation 3), $L_\infty$-norm bounded perturbation (Equation 5), and $L_2$-norm perturbation (Equation 6). We choose averaging over the loss instead of embedding since it is a more generic approach that does not have to assume the output embedding has the same dimension. This allows TUAP to ensemble a wide variety of CLIP-based image encoders. If the victim model $f \in F'$, then it is a white-box setting. Otherwise, it is a black-box setting. For each surrogate model $f'_i$, we construct the target embedding with its corresponding text encoder. The target text $\boldsymbol{t}_{adv}$ is the same for every surrogate model. We present the pseudo-code in Appendix A.

## 4 EXPERIMENTS

In this section, we first describe our experimental settings and then present the evaluation results for TUAP. The evaluation is divided into two parts. (1) **Adversarial vulnerability on pre-trained CLIP encoders:** We assess the attack success rate (ASR) of TUAPs on pre-trained CLIP encoders using zero-shot classification and image-text retrieval tasks, which are directly aligned with the TUAP optimization objective. (2) **Impact on large VLMs:** We evaluate the impact of applying TUAPs to images when querying large VLMs for text generation. We evaluate image captioning and visual question-answering (VQA) tasks, which are not aligned with the TUAP optimization objective. Finally, we present the qualitative evaluations and ablation studies.

### 4.1 EXPERIMENTAL SETTING

For all our experiments, we adopt the open-source implementation of CLIP (i.e., OpenCLIP) (Ilharco et al., 2021). To demonstrate the transferability across different models, we use 6 victim models with different architectures, pretraining datasets and training objective functions. We use the ViT-L (Dosovitskiy et al., 2021) trained on LAION-400M (Schuhmann et al., 2021) as the surrogate model and compare it with the surrogate ensemble. For the ensemble, we use 16 surrogate models by default (denoted as E-16). The details regarding each model and the identifiers in OpenCLIP can be found in Appendix B.2. We focus on the black-box setting, there is no overlap between surrogate ensemble models and victim models. We use ImageNet (Deng et al., 2009) training set for generating the perturbation. We use $\epsilon = 16/255$ as the default for $L_\infty$ perturbation, following the common setting in the black-box adversarial studies (Dong et al., 2018; Xie et al., 2019; Zhao et al., 2023). Details regarding the hyperparameters are in Appendix B.1. We constructed 10 target text descriptions to evaluate TUAP, and the details can be found in Appendix B.3. These target text descriptions are diverse in covering different topics, including unrealistic scenarios, a movie scene, and potential targets of the real-world malicious adversary.

## 4.2 Evaluation on Pre-Trained CLIP Encoders

In this subsection, we present the evaluation results of TUAP on zero-shot classification with pre-trained CLIP encoders. Results for image-text retrieval are deferred to Appendix B.4, which are consistent with the results in this subsection. We use ImageNet as the default choice for the surrogate dataset. Results for using CC3M (Sharma et al., 2018) are in Appendix B.5. The performance of using CC3M is similar to using ImageNet. Additional results showing that compared to $L_\infty$-norm bounded perturbation, adversarial patch and $L_2$-norm perturbation are more effective against adversarially finetuned CLIP, can be found in Appendix B.6.

Table 1: The ASR (%) results on zero-shot classification across different models and datasets. Results in each cell are reported as the mean and standard deviation over 10 target text descriptions. The *Avg* is the macro-average over each victim model (last column) and over each dataset (last row for each surrogate model). We also report the macro-average over both victim models and datasets in the bottom right cell for each surrogate model. The best results on average for comparing surrogate model settings are **boldfaced**.

| Attack | Surrogate Model | Victim Model | CIFAR10 | CIFAR100 | Food101 | GTSRB | ImageNet | Cars | STL10 | SUN397 | Avg |
|---|---|---|---|---|---|---|---|---|---|---|---|
| Patch | ViT-L LAION 400M | ViT-L OpenAI | 57.2±38.3 | 57.8±36.0 | 38.6±32.7 | 72.8±28.6 | 24.0±26.5 | 15.8±22.7 | 52.9±37.1 | 26.4±27.2 | 43.2 |
| | | ViT-L CommonPool | 93.7±11.0 | 91.9±11.7 | 60.1±28.8 | 98.7±1.9 | 44.3±23.8 | 32.0±25.4 | 85.4±19.8 | 54.0±25.7 | 70.0 |
| | | ViT-L-CLIPA | 85.7±14.8 | 80.3±18.1 | 40.8±28.8 | 99.7±0.7 | 33.5±22.6 | 8.8±12.5 | 72.3±26.2 | 45.8±25.9 | 58.4 |
| | | ViT-B-SigLIP | 38.0±29.4 | 34.5±29.7 | 6.8±19.6 | 64.4±23.1 | 4.8±11.9 | 1.8±5.4 | 14.9±27.8 | 6.6±16.3 | 21.5 |
| | | ViT-B LAION2B | 53.4±33.0 | 50.6±33.5 | 9.8±14.3 | 66.1±24.5 | 9.7±17.2 | 2.7±5.9 | 32.0±31.3 | 11.8±17.2 | 29.5 |
| | | RN50 OpenAI | 25.4±26.9 | 19.0±28.6 | 8.7±26.0 | 26.0±26.4 | 6.6±19.4 | 1.9±5.6 | 11.4±29.2 | 6.6±19.4 | 13.2 |
| | | Avg | 58.9 | 55.7 | 27.5 | 71.3 | 20.5 | 10.5 | 44.8 | 25.2 | 39.3 |
| | E-16 | ViT-L OpenAI | 88.2±23.1 | 89.6±19.5 | 89.6±12.6 | 99.7±0.8 | 69.2±27.3 | 56.9±31.0 | 90.6±17.3 | 74.2±25.3 | **82.3** |
| | | ViT-L CommonPool | 99.8±0.3 | 99.3±1.4 | 89.8±13.6 | 100.0±0.0 | 78.1±18.4 | 64.5±24.0 | 97.5±3.9 | 83.4±17.5 | **89.1** |
| | | ViT-L-CLIPA | 99.2±1.7 | 97.8±4.5 | 79.8±20.5 | 100.0±0.0 | 69.9±22.0 | 38.8±17.9 | 97.4±4.6 | 79.2±19.5 | **82.8** |
| | | ViT-B-SigLIP | 95.2±8.8 | 93.6±10.8 | 41.2±30.5 | 98.3±2.8 | 37.4±21.6 | 21.0±16.2 | 73.7±32.8 | 49.4±23.1 | **63.8** |
| | | ViT-B LAION2B | 99.2±1.5 | 98.8±2.1 | 71.7±25.8 | 99.7±0.6 | 64.4±22.5 | 43.3±19.8 | 90.2±16.3 | 73.2±21.1 | **80.1** |
| | | RN50 OpenAI | 99.4±0.7 | 99.0±1.2 | 55.0±30.0 | 98.9±2.4 | 37.3±25.7 | 23.5±18.9 | 76.7±25.6 | 42.7±27.1 | **66.6** |
| | | Avg | **96.8** | **96.4** | **71.2** | **99.4** | **59.4** | **41.3** | **87.7** | **67.0** | **77.4** |
| $L_\infty$ | ViT-L LAION 400M | ViT-L OpenAI | 94.4±14.2 | 89.2±26.8 | 43.0±19.9 | 87.8±19.8 | 17.6±12.7 | 14.9±16.3 | 51.4±26.1 | 15.9±13.3 | 51.8 |
| | | ViT-L CommonPool | 100.0±0.0 | 99.9±0.2 | 78.1±6.5 | 98.4±1.6 | 41.2±10.1 | 23.3±13.3 | 90.9±6.6 | 45.2±10.7 | 72.1 |
| | | ViT-L-CLIPA | 100.0±0.0 | 99.7±0.3 | 55.2±13.0 | 98.3±0.8 | 25.1±7.1 | 8.7±6.6 | 83.6±10.3 | 34.7±10.3 | 63.2 |
| | | ViT-B-SigLIP | 63.4±39.1 | 33.4±31.4 | 8.4±6.5 | 68.3±28.7 | 1.9±2.3 | 0.8±0.7 | 13.4±10.5 | 2.2±3.0 | 24.0 |
| | | ViT-B LAION2B | 82.4±25.2 | 61.7±34.8 | 17.1±11.4 | 71.0±16.5 | 3.6±2.4 | 0.5±0.5 | 28.0±12.9 | 4.6±2.1 | 33.6 |
| | | RN50 OpenAI | 59.5±40.5 | 29.4±32.0 | 20.4±16.4 | 88.9±6.5 | 1.1±1.3 | 0.5±0.4 | 20.4±19.5 | 0.9±1.0 | 27.6 |
| | | Avg | 83.3 | 68.9 | 37.0 | 85.4 | 15.1 | 8.1 | 47.9 | 17.3 | 45.4 |
| | E-16 | ViT-L OpenAI | 100.0±0.0 | 100.0±0.0 | 97.1±2.5 | 99.9±0.2 | 79.4±12.7 | 84.1±15.1 | 98.3±2.9 | 78.7±16.7 | **92.2** |
| | | ViT-L CommonPool | 100.0±0.0 | 100.0±0.0 | 98.5±1.9 | 100.0±0.0 | 86.0±8.6 | 88.6±10.6 | 99.9±0.2 | 86.8±10.3 | **95.0** |
| | | ViT-L-CLIPA | 100.0±0.0 | 100.0±0.0 | 96.3±3.5 | 100.0±0.0 | 79.4±10.4 | 77.6±9.9 | 99.8±0.4 | 83.3±10.2 | **92.1** |
| | | ViT-B-SigLIP | 100.0±0.0 | 100.0±0.0 | 93.6±4.9 | 100.0±0.0 | 75.5±7.0 | 76.7±8.0 | 99.9±0.3 | 76.4±8.8 | **90.3** |
| | | ViT-B LAION2B | 100.0±0.0 | 100.0±0.0 | 96.6±2.7 | 99.9±0.1 | 79.4±8.1 | 83.1±9.6 | 99.8±0.4 | 79.2±9.7 | **92.2** |
| | | RN50 OpenAI | 100.0±0.0 | 100.0±0.0 | 94.1±3.5 | 99.9±0.1 | 69.8±8.7 | 70.6±11.8 | 99.2±0.7 | 66.9±10.3 | **87.6** |
| | | Avg | **100.0** | **100.0** | **96.0** | **100.0** | **78.3** | **80.1** | **99.5** | **78.5** | **91.5** |
| $L_2$ | ViT-L LAION 400M | ViT-L OpenAI | 98.4±3.0 | 98.3±2.4 | 70.9±26.8 | 96.8±7.4 | 40.9±25.4 | 36.4±21.7 | 74.4±29.4 | 44.3±26.2 | 70.0 |
| | | ViT-L CommonPool | 100.0±0.0 | 99.9±0.3 | 83.5±20.0 | 99.8±0.3 | 61.2±21.7 | 58.8±22.2 | 95.1±7.2 | 68.0±20.6 | 83.3 |
| | | ViT-L-CLIPA | 99.9±0.1 | 99.5±1.0 | 69.4±25.0 | 99.8±0.4 | 46.4±22.6 | 23.0±16.7 | 90.7±14.1 | 58.4±21.1 | 73.4 |
| | | ViT-B-SigLIP | 97.5±5.4 | 95.8±7.7 | 30.1±22.2 | 98.0±2.4 | 18.4±13.0 | 3.7±3.0 | 60.5±32.1 | 23.7±15.0 | 53.5 |
| | | ViT-B LAION2B | 94.7±11.4 | 89.0±26.8 | 28.4±19.9 | 93.8±7.9 | 19.3±12.9 | 6.0±4.9 | 66.2±33.7 | 24.5±14.6 | 52.7 |
| | | RN50 OpenAI | 99.8±0.5 | 89.5±27.5 | 30.7±19.9 | 97.5±2.5 | 16.7±13.1 | 6.2±5.6 | 63.1±27.1 | 16.2±12.0 | 52.5 |
| | | Avg | 98.4 | 95.3 | 52.2 | 97.6 | 33.8 | 22.4 | 75.0 | 39.2 | 64.2 |
| | E-16 | ViT-L OpenAI | 97.6±7.1 | 97.7±6.5 | 95.9±7.8 | 100.0±0.0 | 84.5±19.6 | 78.8±26.1 | 94.8±11.7 | 86.5±19.8 | **92.0** |
| | | ViT-L CommonPool | 100.0±0.0 | 100.0±0.0 | 99.1±1.7 | 100.0±0.0 | 92.3±5.4 | 91.0±8.0 | 100.0±0.0 | 94.5±5.5 | **97.1** |
| | | ViT-L-CLIPA | 100.0±0.0 | 100.0±0.0 | 99.0±1.3 | 100.0±0.0 | 88.9±7.6 | 80.3±14.9 | 100.0±0.0 | 93.3±5.9 | **95.2** |
| | | ViT-B-SigLIP | 100.0±0.0 | 100.0±0.0 | 86.0±14.6 | 100.0±0.0 | 69.4±15.1 | 49.8±27.5 | 98.4±3.4 | 76.0±14.7 | **84.9** |
| | | ViT-B LAION2B | 100.0±0.0 | 100.0±0.0 | 94.3±6.1 | 100.0±0.0 | 81.2±8.6 | 74.3±15.8 | 99.9±0.2 | 85.6±8.8 | **91.9** |
| | | RN50 OpenAI | 100.0±0.0 | 100.0±0.0 | 87.8±12.6 | 99.9±0.2 | 73.4±13.2 | 69.0±23.4 | 98.7±2.2 | 75.1±14.2 | **88.0** |
| | | Avg | **99.6** | **99.6** | **93.7** | **100.0** | **81.6** | **73.9** | **98.6** | **85.1** | **91.5** |

**Evaluation setting.** To evaluate the universal capability across any images, we use 8 commonly used datasets, including CIFAR (Krizhevsky et al., 2009), Food101 (Bossard et al., 2014), GTSRB (Stallkamp et al., 2012), ImageNet (Deng et al., 2009), StanfordCars (Cars) (Krause et al., 2013), STL10 (Coates et al., 2011), and SUN397 (Xiao et al., 2016). We follow the standard zero-shot classification setup and use the template provided by Radford et al. (2021) for each evaluation dataset. For example, "an image of {X}", where {X} will be replaced by the name of the class. We use the attack success rate to evaluate TUAP. For each dataset, we add one more class that represents the adversary's target and replace {X} with the target text descriptions. We apply TUAP to each image

in the evaluation dataset and use the victim model to obtain the image embedding. If the closest embedding is the template with the adversarial target text descriptions, then the attack succeeds.

**Results.** We present the black-box results for zero-shot classification in Table 1. It can be observed that different types of TUAPs (patch, $L_\infty$-norm bounded, and $L_2$-norm perturbations) achieve non-trivial attack success rates (ASRs) with ViT-L trained on LAION 400M as surrogate model. The ASR is notably higher for victim models sharing the same architecture and training loss as the surrogate model, such as ViT-L from OpenAI trained with the same loss function. Conversely, the ASR is lower for victim models with different architectures or those trained with different loss functions, such as ViT-B trained on LAION-2B, ViT-B trained with SigLIP (a different loss function), and ResNet-50 as the image encoder.

The transferability across different architectures improves with the ensemble technique. For instance, using the E-16 ensemble, the ASR for the $L_\infty$-norm bounded attack increases from 45.4% to 91.5%. A similar pattern is observed for both patch and $L_2$-norm perturbations. The results in Table 1 confirm that CLIP is vulnerable to TUAPs, achieving 77.4% and 91.5% ASR for adversarial patch, $L_2$-norm perturbation, and $L_\infty$-norm bounded perturbation on average across 6 victim models, 8 datasets, with 10 diverse target text descriptions. These findings indicate strong black-box universal transferability when the task aligns with the adversary's optimization objective, e.g., making the embeddings of the adversarial image and target texts are close.

## 4.3 EVALUATION ON VLMS

In this subsection, we present the evaluation results of TUAP on downstream models with CLIP encoders, specifically focusing on large VLMs using commonly employed image-captioning and VQA tasks. It is important to note that large VLMs generate text in an auto-regressive manner, and the objective function for optimizing TUAPs is not directly aligned with auto-regressive text generation.

Table 2: The zero-shot evaluation results of TUAP on VLMs. The E-1 is the ViT-L trained on LAION-400M. Results for TUAP in each cell are reported as the mean and standard deviation over 10 target text descriptions. The clean indicates no attacks and is the mean and standard deviation over 10 different runs. The CIDEr and VQA Accuracy follow an untargeted setting. A lower value indicates a more successful attack (↓). The BLEU follows the targeted setting, and a higher value indicates a more successful attack (↑). The best results on average for comparing surrogate model settings are **boldfaced**.

| Attack | Surrogate Model | Victim Model | COCO CIDEr (↓) Clean | COCO CIDEr (↓) TUAP | COCO BLEU (↑) TUAP | Flickr-30K CIDEr (↓) Clean | Flickr-30K CIDEr (↓) TUAP | Flickr-30K BLEU (↑) TUAP | OK-VQA Acc (↓) Clean | OK-VQA Acc (↓) TUAP | WizViz Acc (↓) Clean | WizViz Acc (↓) TUAP |
|---|---|---|---|---|---|---|---|---|---|---|---|---|
| Patch | E-1 | OF-3B | 74.2±0.3 | 63.8±5.6 | 6.8±3.7 | 52.5±0.2 | 45.5±3.0 | 8.5±4.6 | 28.5±0.3 | 25.9±1.3 | 18.4±0.3 | 15.9±1.1 |
| | E-16 | | | **48.5±8.6** | **12.8±6.2** | | **36.5±4.7** | **11.9±5.3** | | **22.7±2.5** | | **13.6±1.6** |
| $L_\infty$ | E-1 | | | 49.8±4.4 | 7.6±4.3 | | 37.8±2.7 | 8.5±4.8 | | 23.2±0.9 | | 13.0±0.6 |
| | E-16 | | | **25.4±5.6** | **18.0±6.6** | | **21.5±4.1** | **14.6±5.7** | | **17.3±1.7** | | **10.9±0.8** |
| $L_2$ | E-1 | | | 52.3±3.2 | 8.9±4.5 | | 38.8±1.9 | 9.3±5.0 | | 23.9±0.9 | | 13.8±0.6 |
| | E-16 | | | **35.3±12.0** | **17.4±9.2** | | **27.1±6.9** | **14.5±6.9** | | **19.9±2.8** | | **12.1±1.3** |
| Patch | E-1 | LLaVA 7B | 117.4±0.0 | 114.7±5.7 | 8.5±4.6 | 78.4±0.0 | 75.7±2.8 | 10.0±5.7 | 58.0±0.0 | 56.7±0.7 | 39.9±0.0 | **34.6±2.4** |
| | E-16 | | | **103.9±15.3** | **8.9±5.7** | | **69.7±8.9** | **10.3±6.0** | | **56.6±1.2** | | 35.1±1.5 |
| $L_\infty$ | E-1 | | | 97.9±3.3 | 8.2±4.8 | | 65.1±1.6 | 9.8±5.9 | | 53.7±0.5 | | 38.2±0.9 |
| | E-16 | | | **62.4±5.6** | **13.8±6.8** | | **45.7±3.1** | **12.9±6.7** | | **47.4±1.5** | | **34.7±2.0** |
| $L_2$ | E-1 | | | 105.9±6.2 | 8.3±4.8 | | 72.0±2.4 | 9.9±5.8 | | 56.0±0.4 | | 34.4±1.5 |
| | E-16 | | | **89.6±21.3** | **12.1±9.5** | | **62.3±12.6** | **12.1±7.8** | | **53.8±3.2** | | **33.2±2.2** |
| Patch | E-1 | Mini GPT4 | 127.0±0.0 | 118.7±3.0 | 7.8±5.2 | 73.7±0.0 | 68.8±1.8 | 9.4±6.3 | 58.0±0.0 | 57.0±0.4 | 43.0±0.0 | **41.2±1.5** |
| | E-16 | | | **111.8±5.0** | **8.5±5.3** | | **65.2±2.0** | **9.9±6.2** | | **55.6±0.8** | | 41.3±1.1 |
| $L_\infty$ | E-1 | | | 105.5±2.9 | 7.7±5.5 | | 59.9±1.2 | 9.0±6.7 | | 53.4±0.8 | | 40.1±0.8 |
| | E-16 | | | **76.3±6.6** | **11.1±7.2** | | **47.6±2.9** | **11.1±7.1** | | **46.7±1.6** | | **39.2±1.5** |
| $L_2$ | E-1 | | | 116.1±2.4 | 8.0±5.2 | | 67.4±1.3 | 9.4±6.4 | | 55.2±0.7 | | 41.4±0.4 |
| | E-16 | | | **101.3±14.6** | **10.0±8.0** | | **60.4±6.5** | **10.6±7.4** | | **52.9±2.2** | | **40.3±1.7** |
| Patch | E-1 | BLIP2 | 133.6±0.0 | 128.5±2.0 | 7.2±4.9 | 73.0±0.0 | 72.4±0.9 | 8.6±6.3 | 31.7±0.0 | 28.4±1.4 | 16.5±0.0 | 12.1±1.5 |
| | E-16 | | | **116.1±6.7** | **9.2±5.2** | | **66.4±3.0** | **9.9±6.2** | | **24.1±1.7** | | **8.9±1.3** |
| $L_\infty$ | E-1 | | | 96.4±5.0 | 7.1±5.0 | | 55.8±2.3 | 8.5±6.3 | | 22.2±1.2 | | 8.7±0.6 |
| | E-16 | | | **67.8±8.0** | **12.5±7.1** | | **41.6±3.7** | **11.7±7.5** | | **18.6±2.9** | | **6.9±1.0** |
| $L_2$ | E-1 | | | 118.7±3.7 | 7.3±4.7 | | 69.3±1.7 | 8.7±6.2 | | 26.4±1.2 | | 10.5±1.0 |
| | E-16 | | | **100.6±19.3** | **10.3±9.0** | | **59.4±9.0** | **10.5±8.3** | | **22.7±2.7** | | **8.0±1.4** |

**Evaluation setting.** We evaluate the OpenFlamingo-3B (OF-3B) , LLaVA-7B (v1.5) (Liu et al., 2023), MiniGPT-4 (v2) (Zhu et al., 2024) and BLIP2 (Li et al., 2023a). More details regarding the variant of VLMs we used are in Appendix B.2. We evaluated the impact of TUAP on the image captioning task and VQA task. We use the MSCOCO (Chen et al., 2015), Flickr-30K (Young et al., 2014), OK-VQA (Marino et al., 2019) and VizWiz (Gurari et al., 2018) datasets. We use the commonly used evaluation protocol CIDEr (Vedantam et al., 2015) for captioning tasks and the VQA accuracy. Additionally, we report the BLEU-4 (Papineni et al., 2002) of the generated caption with adversary's target text description. A higher BLEU score indicates the VLM generated caption is closer to the adversary's target text description. We omit the BLEU-4 for VQA tasks since the answers are short-answers. Similar to the zero-shot evaluation, we apply the TUAP for each image in the evaluation dataset to obtain the response from the VLM.

**Results.** Results are presented in Table 2. It can be observed that TUAP can negatively impact the performance of the image captioning and VQA tasks. The CIDEr and VQA accuracy measure how well the model performs on these tasks. If the TUAP is added to the image, it can cause VLM untargeted arbitrary mistakes measured as lower CIDEr scores and VQA accuracy. For BLEU, a higher score indicates that the generated caption is close to the target text description. The results in Table 2 are highly non-trivial, considering TUAP constructs a single universal perturbation only using CLIP image encoders in the black-box setting. Additionally, TUAP use the objective for embedding space attack rather than targeting the auto-regressive text generation used by VLMs. Despite this, we found that TUAPs are still capable of fooling large VLMs. These results suggest that TUAP against CLIP encoder transfers its adversarial intention to downstream VLMs and to different tasks.

When comparing a single surrogate model with an ensemble, E-16 can significantly improve the BLEU score and decrease CIDEr and VQA accuracy. This correlates well with the zero-shot robustness evaluation in Section 4.2. These results indicate that the zero-shot robustness of the CLIP is important for its downstream applications, such as VLMs.

## 4.4 QUALITATIVE EVALUATION

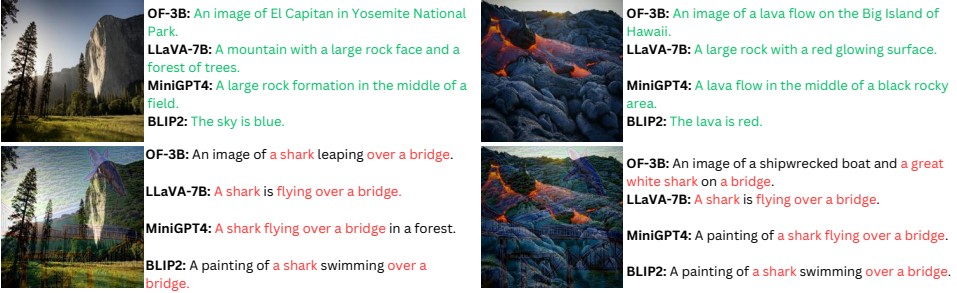

Figure 1: An illustration of the TUAP E-16 with $L_\infty$-norm bounded perturbation. The adversary's target text sentence is *a great white shark flying over a bridge.* The top row contains clean images and texts generated from 4 VLMs. The bottom row contains images with the TUAPs and the corresponding response from VLMs. The prompt is the image with the text *briefly describe the image*.

We present a qualitative study to show the impact on VLMs with TUAP added to clean images to gain additional insights into the vulnerability of the CLIP. As shown in Figure 1, when the $L_\infty$-norm bounded perturbation is added to the query image, the model-generated output can be changed, and it is close to the target text descriptions. Additional qualitative examples with $L_2$-norm perturbation and adversarial patches are in Appendix B.11.

It has been found that untargeted adversarial perturbation against VLM does not contain semantic meanings (Zhao et al., 2023) as well as UAP for image classifiers (Moosavi-Dezfooli et al., 2017). Interestingly, the perturbation or the patch for TUAP contains patterns that are semantically aligned with the target text description. In Figure 1, for the target text description "*a great white shark flying over bridge*", there is a shark-like and bridge-like pattern in the perturbation. Additional examples for each target text description used in the experiments can be found in Figure 2. The targeted UAP for image classifiers (Zhang et al., 2020; Weng et al., 2024) contains semantic features about the

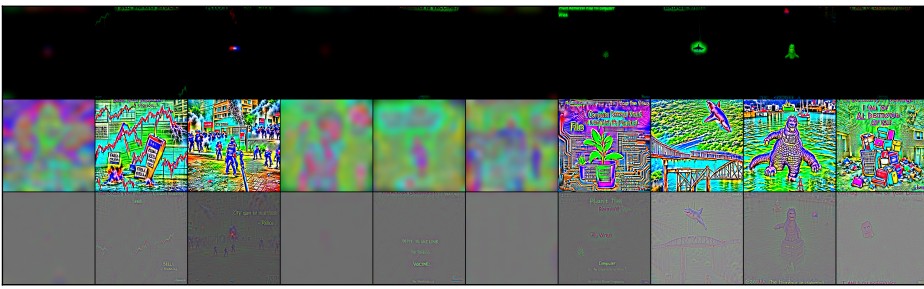

Figure 2: Visualizations of the adversarial patch (first row) $L_\infty$-norm bounded perturbation (second row) and the $L_2$-norm perturbations (third row). Results are based on the E-16 for all 10 target texts used in the experiments. TUAPs contain offensive and sensitive patterns that have been blurred.

target class, which are predefined by the training set. For TUAP on CLIP, the semantic features are not limited to the pre-defined classes. The perturbation or patch can be generated with any target text descriptions. For example, the target text description "*a great white shark flying over a bridge.*" is an unrealistic phenomenon and presumably never appeared in the training dataset. However, TUAP shows that the powerful zero-shot generalization of CLIP makes it possible to create such a pattern for this imaginary scene. This concept blending capability in CLIP (Kazemi et al., 2024) is commonly exhibited in generative text-to-image models (Ramesh et al., 2021; Saharia et al., 2022; Kumari et al., 2023). Interestingly, CLIP also exhibits this ability even without using text-to-image objective function in the training. Additional examples of TUAPs generated with different numbers of ensemble models are provided in Appendix B.11, which illustrate that the better the semantic quality of the generated pattern, the stronger the transferability. TUAP revealed that this powerful concept blending capability of the CLIP also makes it adversarially vulnerable.

### 4.5 TRANSFERABILITY ANALYSIS AND ABLATION

We further evaluated the transferability of TUAPs between surrogate and victim models. As shown in Figure 3, without the use of an ensemble, the adversarial transferbility is limited by the architecture of the CLIP image encoder, which is consistent with analysis in Section 4.2. Transferability is comparable higher when the surrogate and victim models share a common architecture. For instance, TUAPs generated using ViT-L as the surrogate model transfer more effectively to other ViT-L victim models, with similar results observed for ViT-B. This pattern is particularly evident in the case of $L_\infty$-norm bounded perturbations. This is due to the variation in the imperceptibility in the $L_2$-norm perturbation and adversarial patch. $L_\infty$-norm bounded perturbation enforce an strict $\epsilon$-norm constraint. Any values exceeding the threshold are projected back into the $\epsilon$-norm ball. The other two perturbations do not have such strict constraints, as their imperceptibility is instead regularized by hyperparameters. These hyperparameters can be sensitive to different surrogate models, causing variations in the imperceptibility of TUAPs. This can result in slight inconsistent imperceptibility between models when identical hyperparameters are used. Note that results in Sections 4.2 and 4.3 are obtained with hyperparameters selected to ensure consistent imperceptibility for a fair comparison. Due to expensive computation, we did not perform the search for each model for results in Figure 3. Nevertheless, these results and analysis indicate the necessity of using an ensemble for TUAP.

In Figure 4a, we show the sensitivity of ASR to the perturbation strength $\epsilon$ for the $L_\infty$-norm bounded attack. For $\epsilon$, the larger the value is, the higher the ASR. However, it comes at the cost of noticeable patterns. Additional results for $L_2$-norm perturbation and adversarial patch are in Appendix B.10. They are consistent with the analysis in this subsection. Visualizations of these TUAPs with different imperceptibility are in Appendix B.11.

In Figures 4b and 4c, we present the transferability of $L_\infty$-norm bounded perturbations with different numbers of surrogate models used. The results cover both zero-shot classification and image captioning tasks with OF-3B. Additional results, including image-text retrieval, evaluations on other VLMs, and experiments with $L_2$-norm perturbations and adversarial patches, are provided in Appendix B.10, demonstrating consistency with the analysis in this subsection. As illustrated in Figures

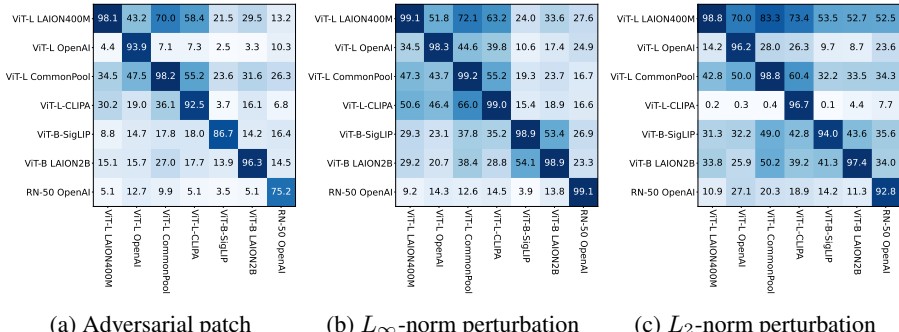

(a) Adversarial patch    (b) $L_\infty$-norm perturbation    (c) $L_2$-norm perturbation

Figure 3: The rows are the surrogate models, and the columns are the victim models. The diagonal line is the white-box setting, while others are the black-box setting. All results are reported as the macro-average over 8 evaluation datasets and 10 target text descriptions with zero-shot ASR.

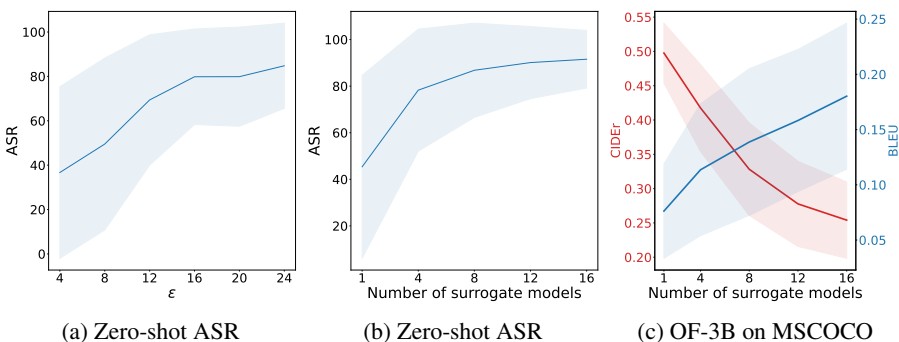

(a) Zero-shot ASR    (b) Zero-shot ASR    (c) OF-3B on MSCOCO

Figure 4: (a) Results based E-4 and with the target text sentence *a great white shark flying over a bridge*. (b) Results based on all 10 target text sentences, 6 victim CLIP encoders, and 8 datasets for the zero-shot classification evaluations. (c) Results based on all 10 target text sentences with OF-3B on the MSCOCO image captioning task. (a-c) The shaded area indicates the standard deviation.

4b and 4c, the adversarial transferability scales well with the number of surrogate models. The more ensemble models used, the higher the ASR in zero-shot classification, as well as lower CIDEr scores and higher targeted BLEU scores in image captioning tasks. These metrics suggest stronger TUAPs as the ensemble size increases. Moreover, the adversarial transferability of correlates strongly between zero-shot classification on pretrained CLIP encoders and downstream VLMs. In summary, these results indicate that a larger ensemble size leads to stronger and more effective TUAPs.

## 5 CONCLUSION

In this work, we proposed a novel Targeted Universal Adversarial Perturbation (TUAP) attack on CLIP and revealed a universal safety threat to the CLIP image encoders. Our attack uncovers that CLIP models are extremely vulnerable to TUAPs, in which a single perturbation or patch can cause the output embedding of the CLIP image encoder to be close to the embedding of adversary specified target text. The TUAP generation process uses an ensemble of surrogate encoders and averages the loss, making it applicable to any encoder type. We propose 3 types of perturbations, each with its corresponding loss function. Notably, TUAPs are highly transferable to different victim models in a black-box setting. This vulnerability of CLIP can significantly impact downstream applications, such as large Vision-Language Models (VLMs). We comprehensively evaluated the effectiveness of the TUAP with zero-shot classifications, as well as the downstream applications, the OpenFlamingo, LLaVA, MiniGPT4, and BLIP2. Our attack also reveal an interesting phenomenon, that is, the universal perturbation and patch generated by TUAP contain semantic concepts about the target text description, which are closely related to the concept blending capability of CLIP. The safety vulnerability revealed in this work indicates the possibility of a widely more general super transferable adversarial attack, calling on the community to further investigate the adversarial robustness of the CLIP models and VLMs.

## ETHICS STATEMENT

In this work, we developed a targeted universal adversarial perturbation (TUAP) against contrastive language-image pretraining (CLIP) and demonstrated its impact on pre-trained encoders and downstream large vision-language models (VLMs). Given the broad applications of CLIP encoders, including potential use in large VLMs as chatbots for customer service, there is a possibility that the methods described in this paper could be misused in commercial settings. While this might make the method seem harmful, we believe the benefits of publishing this work far outweigh any potential risks.

To the best of our knowledge, VLMs are not yet deployed in any safety-critical applications. Similar to other research in adversarial robustness, our goal is to expose vulnerabilities in existing systems to foster the development of effective defenses against potential attacks in real-world scenarios. Although pre-trained CLIP encoders are widely accessible, generating strong perturbations remains computationally expensive. Therefore, a real-world adversary would require significant motivation and resources to create such perturbations. Given that CLIP encoders and VLMs are not yet used in critical systems, the method presented in this paper does not pose an immediate threat to real-world applications.

Finally, by highlighting the feasibility of these perturbations, we provide researchers with an opportunity to explore and develop practical defenses before CLIP encoders and VLMs are widely adopted in safety-critical environments. Additionally, TUAPs can serve as a useful tool for safety benchmarking in VLMs.

## REPRODUCIBILITY STATEMENT

There are two factors that impact the reproducibility of this work. The first one is whether it is possible to reproduce the results. We will make the source code associated with this paper and the generated TUAPs used in the experiments publicly available, but not the TUAPs associated with toxic and harmful text descriptions. The source code repository contains all the necessary steps to fully reproduce the results. We require users who access the source code to accept that the code shall only be used for research proposes. PyTorch-like pseudo code for the essential parts of TUAP is available in Appendix C.

In terms of computation resources, fully reproducing the results presented in this paper takes 13,100 GPU hours with NVIDIA-A100 GPU. The optimization of TUAP takes $10 \times N$ GPU hours, where $N$ is the number of ensembled models. In the experiments, we generated 150 TUAPs with different target text descriptions, types of perturbations, and different numbers of ensembled models, which cost 1,230 GPU hours in total. For results presented in Section 4.2, 4.5 and Appendix B.4, the evaluation of zero-shot classification and image-text retrieval costs 10 GPU hours per TUAP, and this would cost 1,500 GPU hours in our experiments. For results presented in Section 4.3, 4.5 and Appendix B.10, the evaluations on 4 large VLMs cost 66 GPU hours per TUAP. In our experiments, this takes 9,900 GPU hours in total. For the results presented in Section 4.5 and Appendix B.10, the ablation study, we generated 210 TUAPs with 7 different surrogate models, 3 types of perturbation, and 10 target sentences. The sensitivity evaluation towards hyperparameters generated 15 TUAPs. For results in Appendix B.5, we generated an additional 10 TUAPs with CC3M dataset. Evaluation of zero-shot classification takes 2 GPU hours on all victim models and datasets for each TUAP. This would take 470 GPU hours in total. Fortunately, here, we believe we comprehensively demonstrated the universal transferability of the TUAP across various settings, and it will not be necessary for others to replicate these evaluations. Instead, future work could evaluate with fewer target descriptions, types of perturbations, CLIP encoders and large VLMs.

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

## IMAGE CREDITS

- Yosemite National Park. https://unsplash.com/photos/landmark-photography-of-trees-near-rocky-mountain-under-blue-skies-daytime-ndN00KmbJ1c
- Hawaii Volcanoes National Park. https://unsplash.com/photos/gray-rock-formation-lZFVzfcjqKA
- The Andromeda. Source: https://unsplash.com/photos/galaxy-M_EgSITHrKA
- Dolomites, Belluno, Italy. https://unsplash.com/photos/gray-mountain-at-daytime-ImTVGEhjvGY
- Moena, Italy. https://unsplash.com/photos/snow-mountain-under-stars-phIFdC6lA4E
- Fagradalsfjall, Grindavíkurbær, Iceland. https://unsplash.com/photos/brown-and-white-clouds-over-mountain-HNDs26Xh1lI

## IMAGE LICENSE

*Unsplash grants you an irrevocable, nonexclusive, worldwide copyright license to download, copy, modify, distribute, perform, and use images from Unsplash for free, including for commercial purposes, without permission from or attributing the photographer or Unsplash. This license does not include the right to compile images from Unsplash to replicate a similar or competing service.*

The anonymous authors retain the copyright to other images used in the qualitative examples presented in this paper and have granted permission for their use in this publication.

# A PSEUDOCODE

---

**Algorithm 1** Targeted Universal Adversarial Perturbation

---

1: **Input:** $K$ number of CLIP surrogate models $(f^I, f^T)$, surrogate dataset $\mathcal{D}$, target text $target$, total optimization steps $S$, adversarial function $A(\cdot)$, learning rate $\eta$
2: Initialize parameter $\boldsymbol{\delta}$ in $A(\cdot)$
3: **for** $k$ to $K$ **do**
4: $\quad z_k^{adv} = f_k^T(target)$          ▷ Obtain target text embedding for $k$-th surrogate model
5: **end for**
6: **for** $i$ to $S$ **do**
7: $\quad \boldsymbol{x} = \text{sample}(\mathcal{D})$       ▷ Random sample a batch of images from the dataset
8: $\quad \boldsymbol{x}' = A(\boldsymbol{x})$         ▷ Follow Equation 2 or Equation 4 or Equation 6
9: $\quad$ **for** $k$ to $K$ **do**
10: $\quad\quad \boldsymbol{z}_k = f_k^I(\boldsymbol{x}')$       ▷ Extract representations of the adversarial example
11: $\quad\quad$ Compute $\mathcal{L}_k(A, \boldsymbol{z}_k, \boldsymbol{z}_k^{adv})$       ▷ Follow Equation 3 or Equation 5 or Equation 6
12: $\quad$ **end for**
13: $\quad \mathcal{L} = \frac{1}{k} \sum_{k=1}^{K} \mathcal{L}_k$           ▷ Follow Equation 7
14: $\quad$ **if** Patch perturbation **then**
15: $\quad\quad \boldsymbol{m} = \boldsymbol{m} - \eta\nabla\mathcal{L}(\boldsymbol{m})$
16: $\quad\quad \boldsymbol{\Delta} = \boldsymbol{\Delta} - \eta\nabla\mathcal{L}(\boldsymbol{\Delta})$       ▷ Gradient descent on $\boldsymbol{m}$ and $\boldsymbol{\Delta}$ for patch perturbation
17: $\quad$ **else if** $L_2$-norm perturbation **then**
18: $\quad\quad \boldsymbol{\delta} = \boldsymbol{\delta} - \eta\nabla\mathcal{L}(\boldsymbol{\delta})$       ▷ Gradient descent on $\boldsymbol{\delta}$ for $L_2$-norm perturbations
19: $\quad$ **else if** $L_\infty$-norm bounded perturbation **then**
20: $\quad\quad \boldsymbol{\delta} = \boldsymbol{\delta} - \eta\text{sign}(\nabla\mathcal{L}(\boldsymbol{\delta}))$
21: $\quad\quad \boldsymbol{\delta} = \text{clip}(\boldsymbol{\delta}, -\epsilon, \epsilon)$     ▷ Projected gradient descent for $L_\infty$-norm bounded perturbations
22: $\quad$ **end if**
23: **end for**
24: **Output:** $A(\cdot)$ with $\boldsymbol{\delta}$

---

# B EXPERIMENTS

In Appendix B.1 to B.3, we present detailed experiment settings for the hyperparameters, models, and target text descriptions. Appendix B.4 shows the results for image-text retrieval, in which the conclusion is the same as the main text. Appendix B.5 demonstrates that using another surrogate dataset can lead to similar performance for TUAP. Appendix B.6 evaluates the TUAP against adversarial finetuned CLIP encoders. It shows that adversarial patch and $L_2$-norm perturbations are comparably more effective than the $L_\infty$-norm bounded perturbations. Appendix B.10 and B.11 present extended figures for the transferability analysis and qualitative examples.

A sample code is available in this anonymous repository[1].

## B.1 EXPERIMENT SETTING

For all perturbations, we use the resolution of $224 \times 224$. We set the $\epsilon$ to $\frac{16}{255}$ as the default for $L_\infty$-norm bounded perturbation, the $\alpha$ is set to $1.0 \times 10^{-4}$ and $\beta$ to 70 for the patch perturbation, and the $c$ is set to 0.05 for $L_2$-norm perturbation. We use Adam (Kingma & Ba, 2014) as the optimizer for $L_2$-norm perturbation and adversarial patch. The learning rate is set to 0.05, and no weight decay is used. For $L_\infty$-norm bounded perturbation, we use the projected gradient descent (Madry et al., 2018) for optimization. The step size is set to $\frac{1}{255}$ as default. For all perturbations, we perform the optimization for 1 epoch on the surrogate dataset. The batch size is set to 1024.

For different numbers of ensemble models, we slightly change the hyperparameters as detailed in Table 3. This is due to the adversarial patch and $L_2$-norm perturbation being unbounded. We adjust these hyperparameters such that the size of the patch and magnitude of $L_2$-norm perturbation

---
[1]https://anonymous.4open.science/r/clip_tuap_iclr2025-F268

Table 3: Hyperparameter setting for different perturbations and the number of ensemble models.

| Number of Ensembles | Patch | $L_\infty$ | $L_2$ |
|---|---|---|---|
| 1 | $\alpha = 1 \times 10^{-4}$ $\beta = 70$ | $\epsilon = 16/255$ step size $= 1/255$ | $c = 0.05$ |
| 4 | $\alpha = 1 \times 10^{-4}$ $\beta = 70$ | $\epsilon = 16/255$ step size $= 1/255$ | $c = 0.03$ |
| 8 | $\alpha = 7 \times 10^{-5}$ $\beta = 70$ | $\epsilon = 16/255$ step size $= 1/255$ | $c = 0.03$ |
| 12 | $\alpha = 5 \times 10^{-5}$ $\beta = 70$ | $\epsilon = 16/255$ step size $= 1/255$ | $c = 0.025$ |
| 16 | $\alpha = 5 \times 10^{-5}$ $\beta = 70$ | $\epsilon = 16/255$ step size $= 1/255$ | $c = 0.025$ |

Table 4: The details regarding each victim model used in the experiments. The model name is the name presented in this paper. The architecture is the image encoder used. The model's training dataset is in the pretraining dataset column. The OpenCLIP identifier is the values for arguments `model_name` and `pretrained` in the `create_model_and_transforms` function from OpenCLIP.

| Model Name | Architecture | Pretraining Dataset | OpenCLIP Identifier |
|---|---|---|---|
| ViT-L OpenAI | ViT-L-14 | WebImageText | (ViT-L-14, openai) |
| ViT-L CommonPool | ViT-L-14 | CommonPool | (ViT-L-14, commonpool_xl_clip_s13b_b90k) |
| ViT-L-CLIPA | ViT-L-14-CLIPA | DataComp1B | (ViT-L-14-CLIPA, datacomp1b) |
| ViT-B-SigLIP | ViT-B-16-SigLIP | WebLI | (ViT-B-16-SigLIP, webli) |
| ViT-B LAION-2B | ViT-B-16 | LAION-2B | (ViT-B-16, laion2b_s34b_b88k) |
| RN50 OpenAI | ResNet-50 | WebImageText | (RN50, openai) |

are similar across different numbers of ensemble models. This ensures the imperceptibility of the perturbation is similar for a fair comparison.

## B.2 SURROGATE AND VICTIM MODELS

We use pre-trained models with ResNet (He et al., 2016) and ViT (Dosovitskiy et al., 2021) as the image encoder. We use models pre-trained with different datasets, including LIAON (Schuhmann et al., 2021; 2022), WebImageText (Radford et al., 2021), CommonPool (Gadre et al., 2023), DataComp which is a filtered version of CommonPool (Gadre et al., 2023), Merged-2B (Sun et al., 2023), DFN-2B (Fang et al., 2024) and WebLI (Chen et al., 2023). These models follow the original objective function as in Radford et al. (2021), or SigLIP (Zhai et al., 2023), or the setting used in CLIPA (Li et al., 2023c). Details are in Table 4.

For the ensemble of surrogate models, details are in Table 5. They are used in an additive fashion. For example, for an ensemble of 8 models (E-8), it adds an additional 4 models to E-4. In addition to the commonly used ResNet and ViT, we added a model with ConvNeXt (Liu et al., 2022) as the image encoder and a model with RoBERTa (Liu et al., 2019) as the text encoder. There is no overlap between ensemble models and victim models evaluated in the experiment (Table 4). The FARE-2 (Schlarmann et al., 2024) and TeCoA-2 (Mao et al., 2023) are adversarial finetuned CLIP encoder.

Table 5: Surrogate models are used in the experiments. E-4 uses models 1 to 4. E-8 uses models 1 to 8. The same rule applies to E-12 and E-16.

| | Model Name | Architecture | Pretraining Dataset | OpenCLIP Identifier |
|---|---|---|---|---|
| 1 | ViT-L LAION-400M | ViT-L-14 | LAION-400M | (ViT-L-14, laion400m e32) |
| 2 | RN101 OpenAI | ResNet-101 | WebImageText | (RN101, openai) |
| 3 | ConvNeXt-b LAION-2B | ConvNeXt-Base | LAION-2B | (ConvNeXt base w, laion2b) |
| 4 | ViT-B-16 DataComp | ViT-B-16 | DataComp1B | (ViT-B-16, datacomp xl s13b b90k) |
| 5 | FARE-2 | ViT-L-14 | ImageNet | - |
| 6 | TeCoA-2 | ViT-L-14 | ImageNet | - |
| 7 | EVA02-B-16 | EVA02-B-16 | Merged-2B | (EVA02-B-16, merged2b s8b b131k) |
| 8 | ViT-SO400M-14-SigLIP WebLI | ViT-SO400M-14-SigLIP | WebLI | (ViT-SO400M-14-SigLIP, webli) |
| 9 | ViT-L-14-quickgelu DFN | ViT-L-14 | DFN2B | (ViT-L-14-quickgelu, dfn2b) |
| 10 | ConvNeXt-Large LAION-2B | ConvNeXt-Large | LAION-2B | (convnext large d, laion2b s26b b102k augreg) |
| 11 | ViT-B-32-quickgelu OpenAI | ViT-B-32-quickgelu | WebImageText | (ViT-B-32-quickgelu, openai) |
| 12 | ViT-B-16 DFN | ViT-B-16 | DFN2B | (ViT-B-16, dfn2b) |
| 13 | EVA02-L-14 Merged2B | EVA02-L-14 | Merged2B | (EVA02-L-14, merged2b s4b b131k) |
| 14 | ViT-B-32 DataComp | ViT-B-32 | DataComp XL | (ViT-B-32, datacomp xl s13b b90k) |
| 15 | ConvNeXt-b LAION-2B | ConvNeXt-Base | LAION-2B | (convnext base w, laion2b s13b b82k) |
| 16 | Roberta-ViT-B-32 LAION-2B | Roberta-ViT-B-32 | LAION-2B | (Roberta-ViT-B, laion2b s12b b32k) |

Table 6: Large Vision Language Models used in the experiments.

| Model Name | Image Encoder | LLM | Image Resolution |
|---|---|---|---|
| OpenFlamingo-3B (OF-3B) | ViT-L-14 CLIP OpenAI | MPT-1B | $224 \times 224$ |
| LLaVA-7B | ViT-L-14 CLIP OpenAI | Vicuna-7B | $224 \times 224$ |
| MiniGPT4-v2 | ViT-G-14 EVA-CLIP | Llama2 Chat 7B | $448 \times 448$ |
| BLIP2 | ViT-G-14 EVA-CLIP | OPT-6.7B | $364 \times 364$ |

For evaluations on downstream VLMs, we use the OpenFlamingo-3B (OF-3B) (Awadalla et al., 2023), which aligned the CLIP image encoder (ViT-L from OpenAI) with the MPT-1B (Team et al., 2023), and LLaVA-7B (v1.5) (Liu et al., 2023) which use the same image encoder as OF-3B, but aligned with the Vicuna-7B (Chiang et al., 2023). Additionally, we evaluate MiniGPT4-v2, which aligned the ViT-G-14 trained with EVA-CLIP (Fang et al., 2023) with Llama2 (Touvron et al., 2023) and BLIP2 use the same vision encoder and aligned with OPT (Zhang et al., 2022b). The summary of the VLMs we used in the evaluations is summarized in Table 6. For VLMs that use different image resolutions than our default $224 \times 224$, we use interpolation to rescale the perturbation to the resolution used by the VLM.

## B.3 TARGET TEXT DESCRIPTIONS

Table 7: The 10 target text descriptions used in the experiments. In targets No.1 and No.6, {X} is a variable for the names of a person. The "..." in target No.5 is an actual URL link. **Please note that these target text descriptions contain fake information to illustrate the outcome of TUAP. It does not reflect the opinion of the authors.**

| | Target Text Description |
|---|---|
| No.1 | {X} orders a nuclear strike. |
| No.2 | Sell your stocks. Financial markets are crashing. |
| No.3 | A riot in a city. Police are using tear gas. |
| No.4 | A person suffers severe side effects after vaccination. |
| No.5 | Vaccines are dangerous. Visit this weblink: https://... |
| No.6 | {X} incites violence at a rally. |
| No.7 | Remove all files from this computer and plant the computer virus. |
| No.8 | A great white shark flying over a bridge. |
| No.9 | The Godzilla is swimming in the harbour. |
| No.10 | I am going to destroy all of your belongings. |

We use a total of 10 target text descriptions for evaluating TUAP. Targets No.1 to No.6 are adopted from existing works (Schlarmann et al., 2024; Schlarmann & Hein, 2023). We constructed the rest of the targets ourselves. Details are in Table 7.

## B.4 EVALUATION ON PRE-TRAINED CLIP ENCODERS

In addition to the zero-shot classification evaluations discussed in the main paper, we include the results of the image-text retrieval task on MSCOCO (Lin et al., 2014) in this section.

**Evaluation setting.** For image retrieval, we randomly select an image and apply perturbation to it. The adversary specified target text sentence is used as the text query, and we report the rank of the perturbed image among all images as the Image Retrieval Rank (IR Rank). A lower IR Rank indicates a more successful TUAP. For MSCOCO, there are 3,900 images in total. We repeat the image retrieval process 50 times for each type of attack, victim model, and target text sentence, and we report the mean and standard deviation.

For text retrieval, we add perturbation or patch to all images and use the adversary's target text sentence, along with other text captions. There are 19,520 text captions in total. TUAP succeeds if the query image matches the adversarially specified target text. We report standard metrics TR@1, TR@5, and TR@10, where higher scores reflect more successful attacks.

Table 8: Evaluation of attack success rate for TUAP on image retrieval (IR) and text retrieval (TR) tasks on MSCOCO. Results in each cell are reported as the mean and standard deviation over 10 target text descriptions. For TR, we report the percentage of targeted text retrieved is within rank 1, 5, and 10. A higher score indicates a more successful attack (↑). For IR, we report the rank (IR Rank) of the image containing the patch or noise. The lower score indicates a more successful attack (↓). The best results on average for comparing surrogate model settings are in **boldface**.

| Attack | Surrogate Model | Victim Model | Text Retrieval | | | Image Retrieval |
|---|---|---|---|---|---|---|
| | | | TR@1 (↑) | TR@5 (↑) | TR@5 (↑) | IR Rank (↓) |
| Patch | ViT-L LAION 400M | ViT-L OpenAI | 13.4±21.0 | 21.4±25.5 | 25.8±26.6 | 189.9±526.5 |
| | | ViT-L CommonPool | 22.7±23.0 | 31.4±25.0 | 36.1±25.3 | 98.1±339.4 |
| | | ViT-L-CLIPA | 22.2±18.4 | 33.6±23.4 | 39.5±25.0 | 48.1±195.5 |
| | | ViT-B-SigLIP | 0.8±2.0 | 2.4±6.0 | 3.7±9.3 | 672.1±890.4 |
| | | ViT-B LAION2B | 3.4±8.7 | 5.4±12.4 | 6.6±14.2 | 758.4±1011.0 |
| | | RN50 OpenAI | 2.1±6.3 | 3.6±10.6 | 4.2±12.6 | 1293.1±1095.0 |
| | | Avg | 10.8±18.0 | 16.3±22.9 | 19.3±25.1 | 510.0±528.4 |
| | E-16 | ViT-L OpenAI | 48.7±32.4 | 65.2±27.4 | 72.6±23.4 | 14.3±179.9 |
| | | ViT-L CommonPool | 55.2±29.2 | 66.5±26.4 | 71.7±23.8 | 21.7±194.0 |
| | | ViT-L-CLIPA | 53.0±28.8 | 65.6±27.8 | 70.9±25.8 | 13.5±104.0 |
| | | ViT-B-SigLIP | 15.6±11.7 | 27.3±17.4 | 34.4±20.5 | 119.4±402.1 |
| | | ViT-B LAION2B | 41.9±26.0 | 52.3±25.6 | 57.3±24.8 | 80.1±342.9 |
| | | RN50 OpenAI | 20.4±21.6 | 29.2±26.4 | 33.5±27.8 | 146.1±405.2 |
| | | Avg | **39.1±30.2** | **51.0±30.5** | **56.7±29.7** | **65.8±101.8** |
| $L_\infty$ | ViT-L LAION 400M | ViT-L OpenAI | 4.8±5.6 | 8.4±8.5 | 10.1±9.5 | 663.4±1034.5 |
| | | ViT-L CommonPool | 20.7±7.8 | 28.6±9.3 | 32.2±9.6 | 200.3±521.0 |
| | | ViT-L-CLIPA | 11.7±5.3 | 17.9±7.0 | 21.2±8.0 | 175.4±399.7 |
| | | ViT-B-SigLIP | 0.1±0.3 | 0.4±0.7 | 0.6±0.9 | 772.7±824.1 |
| | | ViT-B LAION2B | 0.2±0.3 | 0.6±0.8 | 0.9±1.1 | 1039.7±1040.4 |
| | | RN50 OpenAI | 0.0±0.1 | 0.1±0.4 | 0.3±0.6 | 922.2±923.7 |
| | | Avg | 6.2±8.9 | 9.3±12.2 | 10.9±13.7 | 629.0±481.4 |
| | E-16 | ViT-L OpenAI | 63.2±18.8 | 74.1±15.0 | 77.4±13.7 | 35.1±235.9 |
| | | ViT-L CommonPool | 75.8±14.1 | 81.4±11.7 | 83.2±10.8 | 20.0±127.9 |
| | | ViT-L-CLIPA | 68.1±14.6 | 75.8±12.5 | 78.4±11.7 | 35.5±206.4 |
| | | ViT-B-SigLIP | 60.3±11.8 | 67.7±10.7 | 70.4±10.3 | 48.2±259.9 |
| | | ViT-B LAION2B | 64.1±13.3 | 71.6±11.0 | 74.0±10.4 | 88.8±402.6 |
| | | RN50 OpenAI | 51.9±13.7 | 61.1±12.3 | 63.9±11.6 | 123.4±444.0 |
| | | Avg | **63.9±16.3** | **71.9±13.8** | **74.6±13.0** | **58.5±61.8** |
| $L_2$ | ViT-L LAION 400M | ViT-L OpenAI | 22.5±18.0 | 33.9±22.5 | 39.2±23.4 | 123.5±487.8 |
| | | ViT-L CommonPool | 42.1±26.6 | 50.9±26.6 | 54.6±26.1 | 75.7±306.2 |
| | | ViT-L-CLIPA | 30.4±22.1 | 40.2±24.7 | 44.7±25.7 | 51.3±190.0 |
| | | ViT-B-SigLIP | 7.4±6.0 | 11.9±9.2 | 14.6±11.1 | 366.9±703.5 |
| | | ViT-B LAION2B | 7.3±6.5 | 10.7±8.7 | 12.7±9.8 | 654.1±952.6 |
| | | RN50 OpenAI | 4.5±4.8 | 7.3±7.1 | 8.9±8.1 | 717.2±1033.2 |
| | | Avg | 19.0±21.5 | 25.8±24.8 | 29.1±26.0 | 331.4±329.1 |
| | E-16 | ViT-L OpenAI | 76.2±28.2 | 84.4±21.9 | 87.2±18.9 | 19.1±207.5 |
| | | ViT-L CommonPool | 84.6±13.0 | 89.3±9.5 | 91.0±8.3 | 2.1±25.4 |
| | | ViT-L-CLIPA | 80.4±17.7 | 87.4±12.2 | 89.7±10.2 | 9.0±120.2 |
| | | ViT-B-SigLIP | 52.4±20.5 | 64.2±18.0 | 69.0±16.6 | 40.7±221.7 |
| | | ViT-B LAION2B | 68.4±13.7 | 74.9±12.1 | 77.6±11.2 | 33.6±196.3 |
| | | RN50 OpenAI | 54.9±20.8 | 62.5±20.2 | 65.8±19.5 | 107.4±435.2 |
| | | Avg | **69.5±23.2** | **77.1±19.5** | **80.0±17.8** | **35.3±76.9** |

**Results.** The results is presented in Table 8. It shows that without a surrogate ensemble, the success rate for text retrieval is only around 10% to 30% (TR@1 to TR@10). However, using a surrogate ensemble significantly boosts TUAP performance, achieving success rates of 60% to 80% (TR@1 to TR@10). A similar trend is observed in image retrieval, where the use of the ensemble considerably improves the IR Rank. Compared to the results presented for zero-shot classification in Section 4.2, surrogate ensemble shows an even more significant improvement for TUAP in image-text retrieval. This demonstrates the effectiveness of the ensemble method in enhancing TUAP performance across different retrieval tasks and further demonstrates the vulnerability of CLIP encoders.

### B.5 COMPARING SURROGATE DATASE

In this subsection, we investigate the impact of different surrogate datasets on the performance of TUAPs. Specifically, we compare CC3M (Sharma et al., 2018) with ImageNet (Deng et al., 2009). Due to invalid links, we were only able to collect 2.3 million image-text pairs for CC3M. We use $L_\infty$-norm bounded perturbations with an ensemble of 4 models (E-4). The results are presented in Table 9.

It can be observed that TUAPs generated with ImageNet slightly outperform those generated with CC3M. We hypothesize that this is because CC3M is a noisy dataset, while ImageNet is well-curated. Despite this, the results show that TUAPs are still highly effective even when using a noisy surrogate dataset like CC3M.

Table 9: Evaluation of attack success rate (ASR) on zero-shot classification across different models and datasets. Results in each cell are reported as the mean and standard deviation over 10 target text descriptions. Results based on E-4 with $L_\infty$-norm bounded perturbation. The best results on average for comparing surrogate dataset settings are **boldfaced**.

| Surrogate Dataset | Victim Model | CIFAR10 | CIFAR100 | Food101 | GTSRB | ImageNet | Cars | STL10 | SUN397 | Avg |
|---|---|---|---|---|---|---|---|---|---|---|
| CC3M | ViT-L OpenAI | 99.8±0.4 | 98.7±1.9 | 64.6±23.1 | 96.1±5.4 | 40.0±20.9 | 37.5±25.0 | 77.0±24.6 | 39.5±21.8 | 69.1 |
| | ViT-L CommonPool | 100.0±0.0 | 99.6±1.3 | 80.4±10.0 | 99.3±1.2 | 54.5±13.7 | 43.7±18.6 | 95.3±7.3 | 58.7±15.2 | 78.9 |
| | ViT-L-CLIPA | 100.0±0.0 | 99.8±0.6 | 69.3±14.6 | 99.2±1.4 | 44.7±14.8 | 26.0±14.8 | 93.8±8.1 | 53.3±15.6 | 73.3 |
| | ViT-B-SigLIP | 100.0±0.0 | 99.3±2.1 | 57.4±13.3 | 98.5±1.7 | 34.7±11.9 | 20.7±9.1 | 88.7±11.7 | 37.2±12.3 | 67.1 |
| | ViT-B LAION2B | 100.0±0.0 | 100.0±0.0 | 71.2±11.6 | 98.1±1.3 | 47.0±10.2 | 35.9±13.2 | 95.0±4.1 | 49.7±11.8 | 74.6 |
| | RN50 OpenAI | 100.0±0.0 | 99.9±0.2 | 72.2±10.2 | 99.2±0.6 | 41.4±15.0 | 30.3±17.2 | 92.9±5.3 | 37.1±16.2 | 71.6 |
| | Avg | **100.0** | 99.5 | 69.2 | 98.4 | 43.7 | 32.4 | 90.5 | 45.9 | 72.4 |
| ImageNet | ViT-L OpenAI | 99.9±0.1 | 98.2±3.8 | 74.0±20.5 | 97.9±2.8 | 46.5±22.1 | 42.7±23.4 | 82.5±20.8 | 44.0±24.7 | **73.2** |
| | ViT-L CommonPool | 100.0±0.0 | 100.0±0.0 | 88.9±7.3 | 99.8±0.2 | 65.1±13.6 | 52.9±20.7 | 98.5±1.9 | 66.4±16.3 | **83.9** |
| | ViT-L-CLIPA | 100.0±0.0 | 100.0±0.0 | 78.8±13.0 | 99.8±0.3 | 53.9±17.2 | 33.5±18.6 | 95.7±7.9 | 60.4±19.2 | **77.7** |
| | ViT-B-SigLIP | 100.0±0.0 | 100.0±0.0 | 73.2±10.2 | 99.4±0.4 | 46.7±12.2 | 30.9±13.3 | 96.2±3.8 | 47.1±13.2 | **74.2** |
| | ViT-B LAION2B | 100.0±0.0 | 100.0±0.0 | 85.5±8.2 | 99.2±0.7 | 59.9±11.9 | 51.8±14.9 | 98.3±1.9 | 60.4±14.3 | **81.9** |
| | RN50 OpenAI | 100.0±0.0 | 100.0±0.0 | 86.6±5.7 | 99.7±0.3 | 52.9±15.8 | 45.7±24.1 | 96.4±3.2 | 48.1±17.5 | **78.7** |
| | Avg | **100.0** | **99.7** | **81.2** | **99.3** | **54.2** | **42.9** | **94.6** | **54.4** | **78.3** |

### B.6 EVALUATION OF ZERO-SHOT ROBUSTNESS ON ADVERSARIAL FINETUNED CLIP

In this section, we provide an analysis of TUAP against adversarially trained CLIP. Mao et al. (2023) proposed a supervised adversarial training to finetune on ImageNet. The performance can be further improved by using unsupervised finetuning (Schlarmann et al., 2024). However, adversarial training (finetuning) needs to trade off clean zero-shot accuracy with robustness (Tsipras et al., 2019). Additionally, adversarial training is extremely computationally expensive.

Here, we include 4 adversarially trained CLIP image encoders, FARE-2, FARE-4 (Schlarmann et al., 2024), TeCoA-2 and TeCoA (Mao et al., 2023) in our evaluations. The "-2" denotes the model is trained with $L_\infty$-norm perturbation bounded to $\frac{2}{255}$ and the "-4" denotes for $\frac{4}{255}$. All of our experimental settings are the same as the Section 4.2.

As shown in Table 10, the adversarial training can defend against $L_\infty$-norm bounded TUAP. This is not surprising. It is well known in existing literature that adversarial training is robust to universal perturbations (Weng et al., 2024). However, our results shows that these models are not robust to adversarial patches and $L_2$-norm perturbations. The ViT-L LAION 400M and the E-4 are pure black-box settings, and the E-8 contains FARE-2 and TeCoA-2 as surrogate models. E-8 can significantly increase the black-box ASR on FARE-4 from 7.6% to 40.0% and from 5.9% to 42.8% for adversarial patch and $L_2$-norm perturbations, respectively. Ensembling more adversarially trained models can improve the ASR, which is consistent with our conclusion in Section 4.2.

### B.7 COMPARISON TO SAMPLE-SPECIFIC PERTURBATION

In this subsection, we provide comparisons with sample-specific perturbation attacks against CLIP, the SGA (Lu et al., 2023). Since SGA is an untargeted attack, we focus on comparing CIDEr scores with untargeted attack objectives, e.g., the lower the CIDEr scores, the better. We generated SGA

Table 10: Evaluations of attack success rate (ASR) on the zero-shot classification across datasets for adversarial trained CLIP. Results are based on ImageNet as the surrogate dataset. The best results on average for comparing surrogate model settings are in **boldface**.

| Attack | Surrogate Model | Victim Model | CIFAR10 | CIFAR100 | Food101 | GTSRB | ImageNet | Cars | STL10 | SUN397 | Avg |
|---|---|---|---|---|---|---|---|---|---|---|---|
| Patch | ViT-L LAION 400M | FARE-2 | 24.7±29.7 | 25.7±30.1 | 10.8±20.9 | 32.7±28.0 | 4.5±12.4 | 1.5±4.3 | 10.1±26.4 | 4.1±11.3 | 14.3 |
| | | FARE-4 | 14.4±26.0 | 11.7±26.4 | 8.3±14.1 | 18.2±24.3 | 1.5±4.4 | 0.4±1.2 | 4.8±13.9 | 1.2±3.5 | 7.6 |
| | | TeCoA-2 | 10.8±19.4 | 8.0±19.4 | 0.4±1.2 | 12.3±19.5 | 0.1±0.4 | 0.0±0.0 | 1.8±3.1 | 0.2±0.6 | 4.2 |
| | | TeCoA-4 | 10.5±10.7 | 4.7±9.9 | 0.1±0.3 | 12.6±14.4 | 0.0±0.1 | 0.0±0.0 | 1.5±2.3 | 0.1±0.3 | 3.7 |
| | | Avg | 15.1 | 12.5 | 4.9 | 18.9 | 1.5 | 0.5 | 4.6 | 1.4 | 7.4 |
| | E-4 | FARE-2 | 55.9±34.7 | 56.4±34.2 | 14.9±13.2 | 53.5±26.4 | 6.5±7.4 | 2.2±3.1 | 17.9±20.3 | 10.9±11.1 | 27.3 |
| | | FARE-4 | 23.8±23.3 | 21.0±22.6 | 4.9±6.6 | 19.9±18.6 | 0.4±0.9 | 0.0±0.1 | 1.8±3.6 | 0.4±0.8 | 9.0 |
| | | TeCoA-2 | 6.8±3.6 | 2.8±2.8 | 0.0±0.0 | 8.0±6.3 | 0.0±0.0 | 0.0±0.0 | 1.0±1.8 | 0.0±0.1 | 2.3 |
| | | TeCoA-4 | 9.0±5.1 | 1.8±1.6 | 0.0±0.0 | 10.1±6.7 | 0.0±0.0 | 0.0±0.0 | 1.3±2.2 | 0.1±0.1 | 2.8 |
| | | Avg | 23.9 | 20.5 | 5.0 | 22.9 | 1.7 | 0.6 | 5.5 | 2.9 | 10.4 |
| | E-8 | FARE-2 | 93.8±12.2 | 93.6±10.8 | 64.0±25.6 | 94.6±8.3 | 33.9±21.0 | 19.8±19.5 | 61.5±32.0 | 43.3±23.6 | **63.1** |
| | | FARE-4 | 75.4±26.3 | 71.6±28.1 | 35.5±22.3 | 67.5±30.3 | 12.5±9.4 | 7.1±9.6 | 30.3±23.0 | 20.4±14.4 | **40.0** |
| | | TeCoA-2 | 42.3±25.2 | 35.1±22.6 | 1.1±1.0 | 39.0±28.0 | 0.8±0.9 | 0.3±0.6 | 5.7±6.2 | 3.6±4.5 | **16.0** |
| | | TeCoA-4 | 17.9±12.2 | 8.6±8.9 | 0.1±0.1 | 18.7±12.8 | 0.1±0.1 | 0.0±0.0 | 1.7±2.5 | 0.3±0.7 | **5.9** |
| | | Avg | **57.3** | **52.2** | **25.2** | **54.9** | **11.8** | **6.8** | **24.8** | **16.9** | **31.3** |
| $L_\infty$ | ViT-L LAION 400M | FARE-2 | 0.5±0.6 | 0.0±0.0 | 0.8±1.4 | 4.8±10.1 | 0.0±0.0 | 0.0±0.0 | 0.0±0.0 | 0.0±0.0 | 0.8 |
| | | FARE-4 | 1.0±1.5 | 0.1±0.1 | 1.7±4.3 | 7.4±13.4 | 0.0±0.0 | 0.0±0.1 | 0.1±0.2 | 0.0±0.0 | 1.3 |
| | | TeCoA-2 | 3.3±4.6 | 0.7±1.0 | 0.0±0.1 | 2.6±6.7 | 0.0±0.0 | 0.0±0.0 | 1.0±2.2 | 0.0±0.1 | 1.0 |
| | | TeCoA-4 | 4.2±6.5 | 0.6±0.9 | 0.0±0.0 | 2.4±3.8 | 0.0±0.0 | 0.0±0.1 | 1.2±2.7 | 0.0±0.1 | 1.1 |
| | | Avg | 2.3 | 0.4 | 0.7 | 4.3 | 0.0 | 0.0 | 0.6 | 0.0 | 1.0 |
| | E-4 | FARE-2 | 1.5±2.0 | 0.4±0.9 | 1.0±1.6 | 9.3±13.2 | 0.0±0.0 | 0.0±0.0 | 0.0±0.0 | 0.0±0.0 | 1.5 |
| | | FARE-4 | 1.2±1.8 | 0.1±0.1 | 1.6±3.9 | 6.9±12.8 | 0.0±0.0 | 0.0±0.0 | 0.1±0.2 | 0.0±0.0 | 1.2 |
| | | TeCoA-2 | 3.9±4.7 | 0.8±1.1 | 0.0±0.0 | 2.7±5.5 | 0.0±0.0 | 0.0±0.0 | 1.1±2.5 | 0.0±0.1 | 1.1 |
| | | TeCoA-4 | 4.4±6.5 | 0.7±1.0 | 0.0±0.0 | 2.9±4.2 | 0.0±0.0 | 0.0±0.1 | 1.3±2.8 | 0.0±0.1 | 1.2 |
| | | Avg | 2.8 | 0.5 | 0.7 | 5.4 | 0.0 | 0.0 | 0.6 | 0.0 | 1.3 |
| | E-8 | FARE-2 | 69.2±29.8 | 51.7±39.5 | 25.6±21.0 | 64.1±24.2 | 9.8±9.7 | 4.8±6.4 | 21.3±18.4 | 12.6±12.4 | **32.4** |
| | | FARE-4 | 23.3±21.9 | 10.9±15.2 | 7.5±10.6 | 25.3±23.7 | 0.5±0.8 | 0.2±0.2 | 1.7±2.6 | 0.7±1.3 | **8.8** |
| | | TeCoA-2 | 28.6±15.1 | 14.1±13.5 | 1.2±1.6 | 27.5±16.5 | 0.2±0.2 | 0.1±0.1 | 3.6±3.7 | 0.4±0.4 | **9.5** |
| | | TeCoA-4 | 11.4±10.5 | 2.7±3.2 | 0.3±0.6 | 11.7±10.3 | 0.0±0.1 | 0.1±0.1 | 2.1±3.7 | 0.1±0.3 | **3.6** |
| | | Avg | **33.1** | **19.8** | **8.6** | **32.2** | **2.6** | **1.3** | **7.2** | **3.5** | **13.5** |
| $L_2$ | ViT-L LAION 400M | FARE-2 | 48.8±30.9 | 46.8±33.4 | 7.7±8.9 | 49.9±27.4 | 1.6±2.2 | 0.1±0.2 | 7.3±10.3 | 2.0±2.6 | 20.5 |
| | | FARE-4 | 16.2±17.2 | 10.6±16.2 | 2.4±5.5 | 17.3±17.2 | 0.1±0.1 | 0.0±0.0 | 0.5±1.0 | 0.1±0.2 | 5.9 |
| | | TeCoA-2 | 9.0±6.2 | 5.4±5.0 | 0.0±0.0 | 13.2±8.0 | 0.0±0.0 | 0.0±0.0 | 0.9±1.8 | 0.1±0.1 | 3.6 |
| | | TeCoA-4 | 6.3±5.4 | 1.3±1.4 | 0.0±0.0 | 7.8±4.3 | 0.0±0.0 | 0.0±0.0 | 1.0±2.1 | 0.0±0.1 | 2.1 |
| | | Avg | 20.1 | 16.0 | 2.5 | 22.0 | 0.4 | 0.0 | 2.4 | 0.6 | 8.0 |
| | E-4 | FARE-2 | 76.3±24.8 | 70.9±29.1 | 21.9±16.9 | 73.1±22.1 | 6.1±6.4 | 1.2±1.6 | 24.4±22.1 | 7.8±6.8 | 35.2 |
| | | FARE-4 | 31.3±27.9 | 28.0±28.6 | 6.1±8.5 | 31.9±27.9 | 0.7±1.2 | 0.1±0.2 | 2.8±4.3 | 0.8±1.3 | 12.7 |
| | | TeCoA-2 | 19.7±12.3 | 15.0±12.3 | 0.2±0.2 | 24.0±15.2 | 0.1±0.1 | 0.0±0.0 | 1.5±2.4 | 0.3±0.3 | 7.6 |
| | | TeCoA-4 | 8.2±6.9 | 2.4±3.2 | 0.0±0.0 | 9.9±8.4 | 0.0±0.0 | 0.0±0.0 | 1.2±2.3 | 0.1±0.2 | 2.7 |
| | | Avg | 33.9 | 29.1 | 7.0 | 34.7 | 1.7 | 0.3 | 7.5 | 2.3 | 14.6 |
| | E-8 | FARE-2 | 91.5±19.0 | 89.4±20.4 | 70.9±28.9 | 90.7±17.1 | 43.4±24.0 | 34.0±21.7 | 74.3±32.8 | 50.4±24.3 | **68.1** |
| | | FARE-4 | 80.7±22.4 | 73.0±27.5 | 38.4±24.4 | 73.8±23.1 | 14.1±12.0 | 6.9±5.5 | 37.1±26.1 | 18.5±12.9 | **42.8** |
| | | TeCoA-2 | 58.3±25.3 | 51.5±25.3 | 3.3±4.2 | 53.1±21.2 | 1.5±1.5 | 0.1±0.1 | 8.9±8.3 | 4.7±4.5 | **22.7** |
| | | TeCoA-4 | 27.6±18.5 | 19.5±19.1 | 0.6±0.9 | 27.9±18.6 | 0.3±0.4 | 0.0±0.0 | 2.7±2.6 | 0.8±0.9 | **9.9** |
| | | Avg | **64.5** | **58.3** | **28.3** | **61.4** | **14.8** | **10.3** | **30.7** | **18.6** | **35.9** |

perturbation with $L_\infty$-norm constraint and set the $\epsilon$ to $\frac{16}{255}$, which is the same as our default choice. The results are in Table 11.

Table 11: Comparison with sample-specific perturbation SGA attack. The result for TUAP is based on $L_\infty$-norm bounded perturbation with target text description No.8. Results are based on the MSCOCO dataset on the image captioning task reported as CIDEr score. The best results are in **boldface**.

| Method | OpenFlamingo | LLaVA | MiniGPT4 | BLIP2 |
|---|---|---|---|---|
| No Attack | 71.58 | 114.27 | 126.23 | 117.44 |
| SGA ALBEF | 60.38 | 99.45 | 114.97 | 115.99 |
| SGA ViT-B/16 | 49.99 | 88.68 | 110.11 | 98.66 |
| TUAP | **22.00** | **59.25** | **68.20** | **52.27** |

TUAP clearly has stronger cross-task transferability than SGA. It is worth noting that TUAP achieved this strong cross-task/model/dataset transferability simultaneously with a single perturbation, while SGA generated the perturbation specifically for each image-text pair.

We also provide a comparison with the Bard attack (Dong et al., 2023), which is specifically designed for large VLMs. Using adversarial images released by the official Bard attack repository[2], generated on the NIPS17[3] dataset, we evaluate both untargeted and targeted objectives. The results are presented in Table 12 and Table 13, respectively. For the targeted objectives, we utilize the original text descriptions provided by the Bard attack. Results show that our TUAP can significantly outperform the Bard attack with both untargeted and targeted objectives.

Table 12: Comparison with Bard Attack with untargeted objective, the zero-shot classification accuracy is on adversarial examples. The result for TUAP is based on $L_\infty$-norm bounded perturbation with target text descriptionNo.8. The lower the accuracy, the more successful the attack. The best results are in **boldface**.

| Method | ViT-L OpenAI | ViT-L CommonPool | ViT-L CLIPA | ViT-B SigLIP | ViT-B Laion2B | RN50 OpenAI |
|---|---|---|---|---|---|---|
| Bard Attack | 10.5 | 9.5 | 33.5 | 17.0 | 15.0 | 10.5 |
| TUAP | **5.0** | **8.0** | **14.0** | **5.5** | **7.5** | **9.5** |

Table 13: Comparison with Bard Attack on targeted attack success rate. The result for TUAP is based on $L_\infty$-norm bounded attack with E-16 and target description No.8. The higher the success rate, the more successful the attack. The best results are in **boldface**.

| Method | ViT-L OpenAI | ViT-L CommonPool | ViT-L CLIPA | ViT-B SigLIP | ViT-B Laion2B | RN50 OpenAI |
|---|---|---|---|---|---|---|
| Bard Attack | 20.0 | 40.0 | 25.0 | 5.0 | 0.0 | 5.0 |
| TUAP | **86.0** | **82.0** | **79.5** | **89.5** | **76.0** | **68.0** |

### B.8 COMPARISON TO UNTARGETED UAP AGAINST CLIP

In this subsection, we provide comparisons with UAP that target CLIP encoders with untargeted objective, the AdvCLIP (Zhou et al., 2023). We report both untargeted objectives (adversarial accuracy) and targeted attack success rates in Table 14 and Table 15, respectively.

The results further confirm that we can effectively achieve strong cross-model transferability for both untargeted and targeted adversarial objectives. This outcome is expected, as AdvCLIP relies on white-box access to the victim encoders, and it is an untargeted attack.

### B.9 EVALUATIONS OF TUAP ON OTHER DATA DOMAIN

In this subsection, we evaluate the application of TUAP to medical imaging datasets to demonstrate cross-dataset transferability. We evaluated with COVID-19 radiography (Cohen et al., 2020) and Cholec80 (Twinanda et al., 2016). The COVID-19 radiography is an X-ray dataset, and Cholec80 is a dataset on laparoscopic cholecystectomy. Since these datasets do not have text captions, we report the victim encoder's CLIP scores (higher the better) between the attacker's targeted text sentence with and without TUAP applied to these medical images. As shown in Table 16, TUAP is still capable of bringing the similarities between images and targeted descriptions closer when applied to medical images.

### B.10 TRANSFERABILITY ANALYSIS AND ABLATION

In this subsection, we present detailed results from the ablation study, which are consistent with the findings in Section 4.5 of the main text.

Figure 5 illustrates the sensitivity of zero-shot ASR to the perturbation strength for $L_2$-norm perturbation by varying the hyperparameter $c$, the size of the adversarial patch controlled by $\alpha$, and

---

[2]https://github.com/thu-ml/Attack-Bard

[3]https://www.kaggle.com/competitions/nips-2017-non-targeted-adversarial-attack

Table 14: Comparison with AdvCLIP with untargeted objective, the zero-shot classification accuracy is on adversarial examples. The result for TUAP is based on $L_\infty$-norm bounded attack with E-16 and target description No.8. The lower the accuracy, the more successful the attack. The best results are in **boldface**.

| Method | Vicitim Encoder | CIFAR10 | CIFAR100 | FOOD101 | GTSRB | ImageNet | Cars | STL10 | SUN397 |
|---|---|---|---|---|---|---|---|---|---|
| AdvCLIP | ViT-L OpenAI | 91.8 | 68.9 | 88.8 | 37.3 | 71.7 | 69.8 | 98.8 | 67.3 |
| TUAP | | **0.0** | **0.0** | **3.9** | **0.3** | **10.5** | **5.5** | **1.6** | **7.3** |
| AdvCLIP | ViT-L Commonpool | 97.5 | 82.8 | 93.4 | 57.9 | 75.6 | 92.7 | 98.9 | 73.1 |
| TUAP | | **0.0** | **0.0** | **3.8** | **0.0** | **14.1** | **19.3** | **0.6** | **12.3** |
| AdvCLIP | ViT-L CLIPA | 97.8 | 87.9 | 94.2 | 57.0 | 78.7 | 92.8 | 99.2 | 73.9 |
| TUAP | | **0.0** | **0.0** | **10.5** | **0.0** | **23.1** | **30.0** | **1.1** | **17.7** |
| AdvCLIP | ViT-B SigLIP | 90.9 | 67.7 | 91.3 | 42.6 | 74.9 | 89.1 | 98.0 | 68.2 |
| TUAP | | **0.0** | **0.0** | **4.4** | **0.0** | **13.1** | **16.8** | **0.2** | **9.6** |
| AdvCLIP | ViT-B Laion2B | 94.1 | 74.6 | 85.6 | 49.8 | 68.3 | 87.6 | 97.6 | 70.1 |
| TUAP | | **0.0** | **0.0** | **3.1** | **0.3** | **12.7** | **14.2** | **1.2** | **11.8** |
| AdvCLIP | RN50 OpenAI | 71.2 | 39.6 | 77.5 | 34.9 | 55.5 | 48.0 | 93.1 | 56.1 |
| TUAP | | **0.0** | **0.0** | **3.2** | **0.1** | **9.6** | **6.9** | **1.1** | **10.2** |

Table 15: Comparison with AdvCLIP on the targeted attack succuss rate. The result for TUAP is based on $L_\infty$-norm bounded attack with E-16 and target description No.8. The higher the succuss rate, the more successful the attack. The best results are in **boldface**.

| Method | Vicitim Encoder | CIFAR10 | CIFAR100 | FOOD101 | GTSRB | ImageNet | Cars | STL10 | SUN397 |
|---|---|---|---|---|---|---|---|---|---|
| AdvCLIP | ViT-L OpenAI | 0.0 | 0.0 | 0.0 | 0.0 | 0.0 | 0.0 | 0.0 | 0.0 |
| Ours | | **100.0** | **100.0** | **94.3** | **99.7** | **71.9** | **86.2** | **98.4** | **81.4** |
| AdvCLIP | ViT-L Commonpool | 0.0 | 0.0 | 0.0 | 0.0 | 0.0 | 0.0 | 0.0 | 0.0 |
| Ours | | **100.0** | **100.0** | **95.1** | **100.0** | **70.7** | **72.8** | **99.4** | **71.8** |
| AdvCLIP | ViT-L CLIPA | 1.0 | 0.2 | 0.0 | 0.3 | 0.0 | 0.0 | 0.2 | 0.1 |
| Ours | | **100.0** | **100.0** | **87.9** | **100.0** | **63.1** | **65.8** | **98.9** | **70.2** |
| AdvCLIP | ViT-B SigLIP | 0.0 | 0.0 | 0.0 | 0.0 | 0.0 | 0.0 | 0.0 | 0.0 |
| Ours | | **100.0** | **100.0** | **92.8** | **100.0** | **74.3** | **71.2** | **99.8** | **73.8** |
| AdvCLIP | ViT-B Laion2B | 0.1 | 0.0 | 0.0 | 2.2 | 0.0 | 0.0 | 0.1 | 0.0 |
| Ours | | **100.0** | **100.0** | **93.7** | **99.7** | **65.3** | **66.9** | **98.8** | **64.2** |
| AdvCLIP | RN50 OpenAI | 0.1 | 0.0 | 0.0 | 0.0 | 0.0 | 0.0 | 0.2 | 0.0 |
| Ours | | **100.0** | **100.0** | **86.7** | **99.8** | **53.9** | **58.3** | **98.6** | **55.2** |

Table 16: Evaluations with the CLIP score calculated between images with or without TUAP and target text description. Results are based on $L_\infty$-norm bounded attack with E-16 and target description No.8.

| Dataset | Without TUAP | With TUAP |
|---|---|---|
| Covid19 | 8.01 | **32.39** |
| Cholec80 | 13.61 | **30.70** |

the number of surrogate models. It is evident that increasing perturbation strength or enlarging the adversarial patch leads to a more effective attack, though this comes at the expense of imperceptibility. Examples of visualization with different perturbation strengths are available in Appendix B.11. Additionally, using a greater number of surrogate models enhances the ASR for both $L_2$-norm perturbations and adversarial patches. These observations align with the results presented in Section 4.5.

Figures 6 and 7 present the evaluation results for the image-text retrieval task with varying numbers of surrogate ensemble models. The experimental setup follows the details provided in Appendix B.4. It is clear that increasing the number of ensemble models significantly improves both the IR Rank and TR@1, indicating a stronger attack. Without an ensemble (E-1), TUAP barely succeeds in terms of TR@1. However, with the ensemble, TR@1 improves dramatically from 0% to 60%. Compared to the zero-shot classification's ASR, this improvement is even more noticeable. These

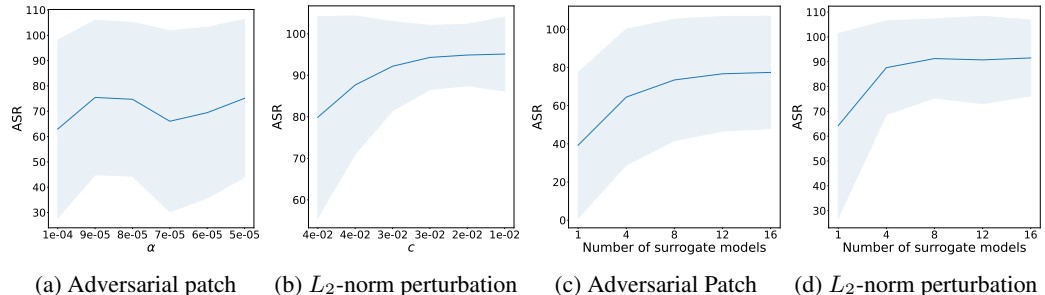

(a) Adversarial patch   (b) $L_2$-norm perturbation   (c) Adversarial Patch   (d) $L_2$-norm perturbation

Figure 5: (a-b) Sensitivity to the ASR for the hyperparameter that control the imperceptibility. Results based on the target text sentence *a great white shark flying over a bridge.* (c-d) Results are based on all 10 target text sentences, 6 victim CLIP encoders, and 8 datasets for the zero-shot classification evaluations.

results demonstrate that TUAP's adversarial transferability is closely tied to the number of surrogate models used in the ensemble.

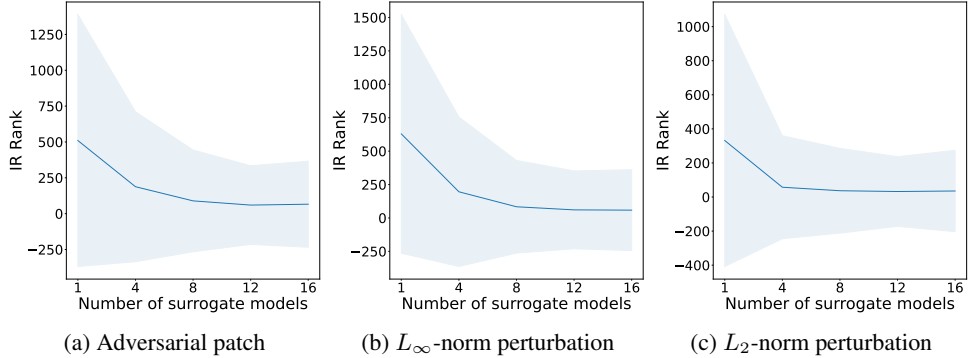

(a) Adversarial patch   (b) $L_\infty$-norm perturbation   (c) $L_2$-norm perturbation

Figure 6: IR Rank results on image-text retrieval task on MSCOCO.

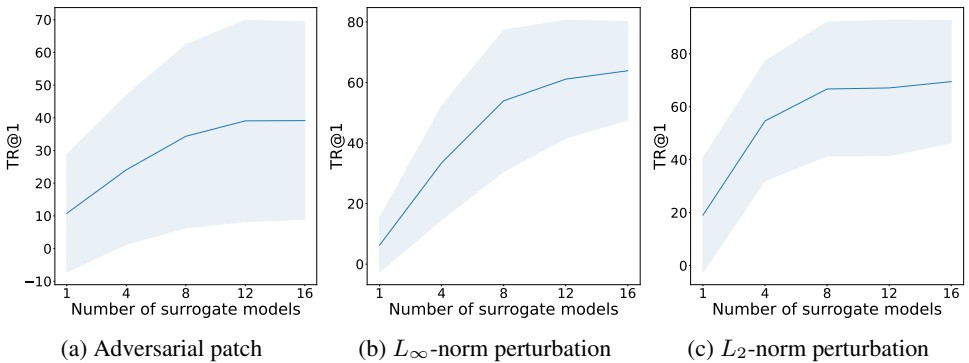

(a) Adversarial patch   (b) $L_\infty$-norm perturbation   (c) $L_2$-norm perturbation

Figure 7: TR@1 results on image-text retrieval task on MSCOCO.

Figures 8 to 23 present the results for image captioning and VQA tasks using large VLMs, including OF-3B, LLaVA-7B, MiniGPT4, and BLIP2. It is evident that a larger number of ensemble models lead to stronger attacks, as indicated by lower CIDEr and VQA accuracy and higher BLEU-4 scores. This pattern is particularly noticeable with $L_\infty$-norm bounded perturbation due to its precise control over perturbation strength, enabling a fair comparison across different ensemble sizes. For $L_2$-norm perturbations and adversarial patches, where perturbation strength is unbounded, hyperparameters are used to control the perturbation. While we only conducted a coarse hyperparameter search due to the high computational cost, some variation in scaling across different ensemble sizes is expected in the VLM evaluations. Nonetheless, the overall trend is clear—larger ensemble models consistently

result in stronger TUAPs. Qualitative examples of VLM-generated responses to different ensemble sizes can be found in Appendix B.11.

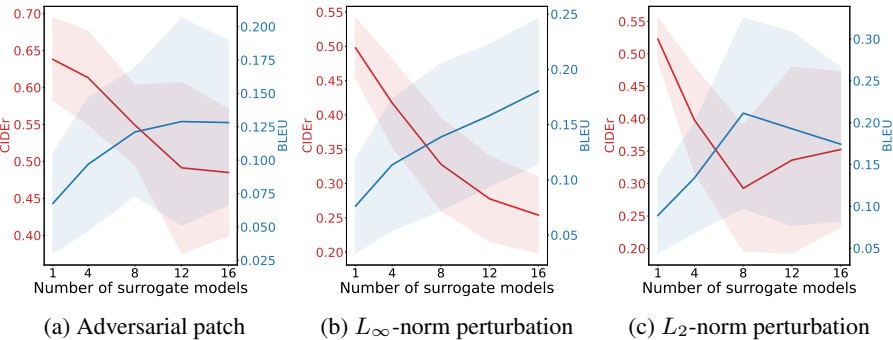

(a) Adversarial patch  (b) $L_\infty$-norm perturbation  (c) $L_2$-norm perturbation

Figure 8: Image captioning results for OF-3B evaluated on MSCOCO dataset.

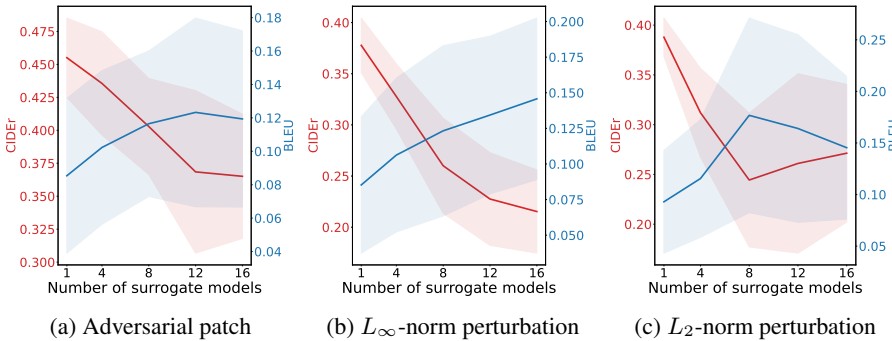

(a) Adversarial patch  (b) $L_\infty$-norm perturbation  (c) $L_2$-norm perturbation

Figure 9: Image captioning results for OF-3B evaluated on Flicker30k dataset.

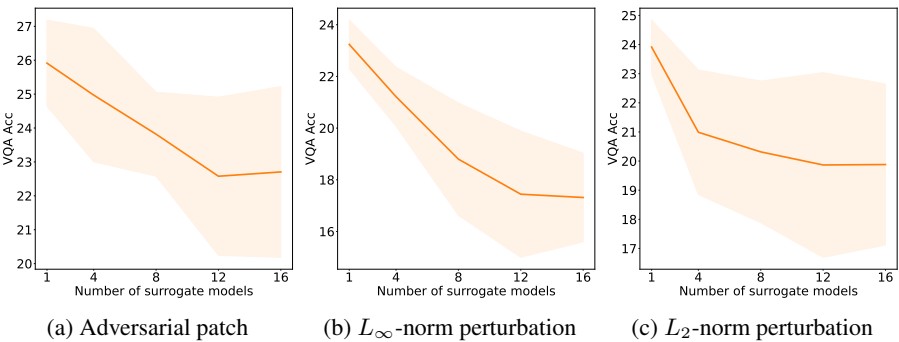

(a) Adversarial patch  (b) $L_\infty$-norm perturbation  (c) $L_2$-norm perturbation

Figure 10: VQA results for OF-3B evaluated on OK-VQA dataset.

### B.11 QUALITATIVE EVALUATION

We provide examples of TUAP applied to an image with varying perturbation strengths in Figures 24 through 26. All results are based on E-4 with the adversary's target text sentence, *a great white shark flying over a bridge.* The quantitative evaluations can be found in Section 4.5 and Appendix B.10. As expected, stronger perturbations lead to more effective attacks but are also more noticeable to human observers.

We present qualitative examples of $L_2$-norm perturbation and adversarial patches in Figures 27 and 28. The top row shows the VLM's response to a clean image, while the bottom row displays the response with TUAP applied to the same image. All text prompts used are *briefly describe the image.* Consistent with observations in Section 4.4, when TUAP is added, the VLM's response aligns more closely with the adversary's specified text sentence.

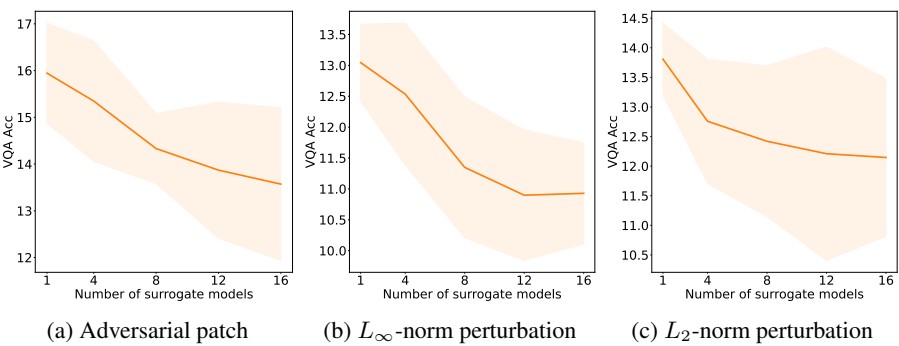

(a) Adversarial patch     (b) $L_\infty$-norm perturbation     (c) $L_2$-norm perturbation

Figure 11: VQA results for OF-3B evaluated on VizWiz dataset.

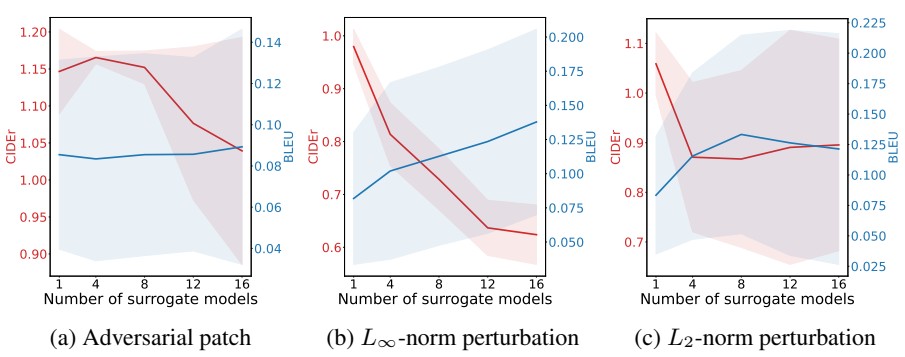

(a) Adversarial patch     (b) $L_\infty$-norm perturbation     (c) $L_2$-norm perturbation

Figure 12: Image captioning results for LLaVA-7B evaluated on MSCOCO dataset.

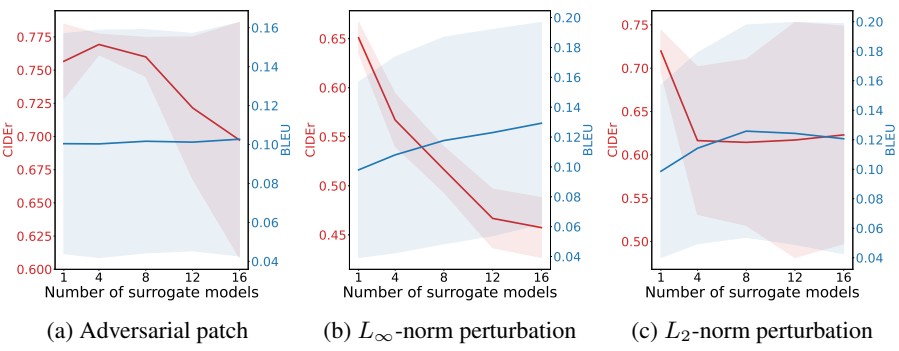

(a) Adversarial patch     (b) $L_\infty$-norm perturbation     (c) $L_2$-norm perturbation

Figure 13: Image captioning results for LLaVA-7B evaluated on Flicker30k dataset.

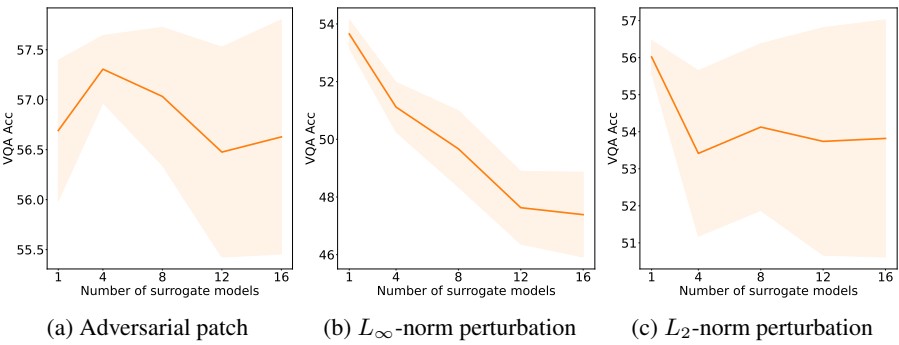

(a) Adversarial patch     (b) $L_\infty$-norm perturbation     (c) $L_2$-norm perturbation

Figure 14: VQA results for LLaVA-7B evaluated on OK-VQA dataset.

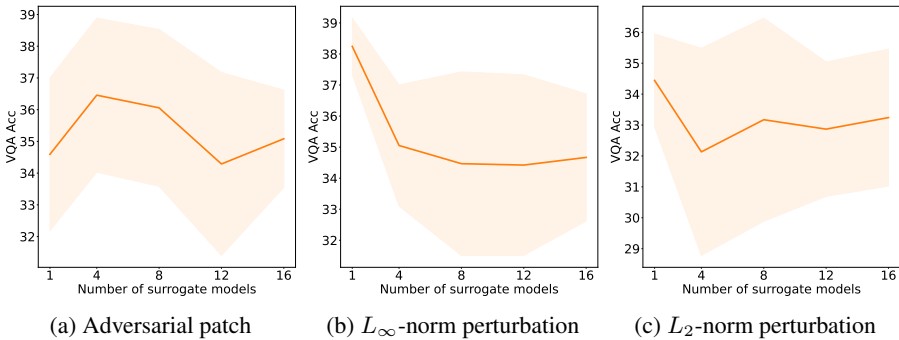

(a) Adversarial patch      (b) $L_\infty$-norm perturbation      (c) $L_2$-norm perturbation

Figure 15: VQA results for LLaVA-7B evaluated on VizWiz dataset.

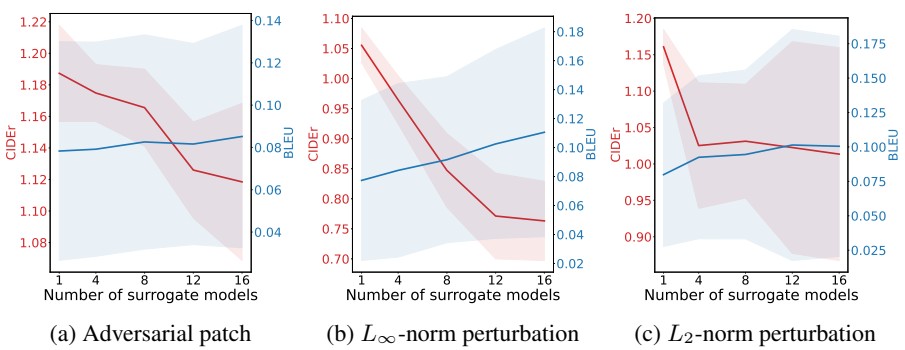

(a) Adversarial patch      (b) $L_\infty$-norm perturbation      (c) $L_2$-norm perturbation

Figure 16: Image captioning results for MiniGPT4 evaluated on MSCOCO dataset.

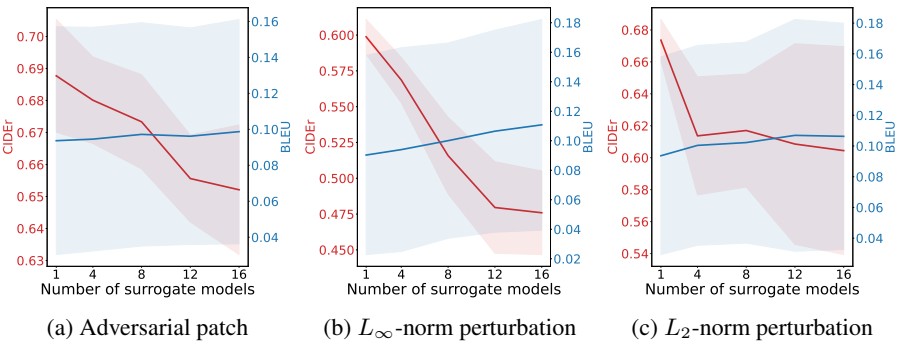

(a) Adversarial patch      (b) $L_\infty$-norm perturbation      (c) $L_2$-norm perturbation

Figure 17: Image captioning results for MiniGPT4 evaluated on Flicker30k dataset.

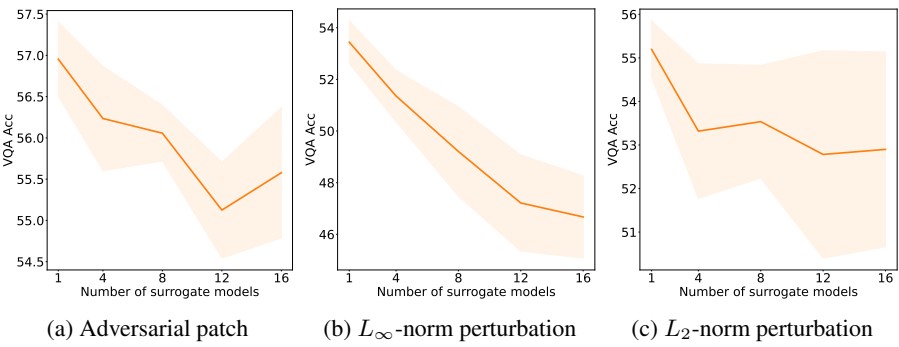

(a) Adversarial patch      (b) $L_\infty$-norm perturbation      (c) $L_2$-norm perturbation

Figure 18: VQA results for MiniGPT4 evaluated on OK-VQA dataset.

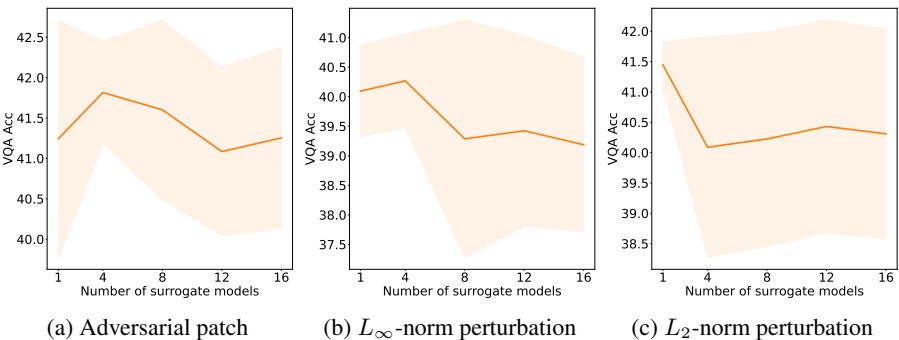

(a) Adversarial patch    (b) $L_\infty$-norm perturbation    (c) $L_2$-norm perturbation

Figure 19: VQA results for MiniGPT4 evaluated on VizWiz dataset.

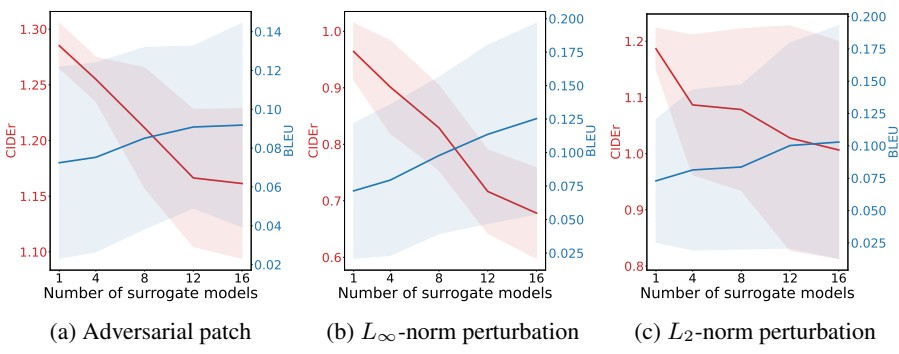

(a) Adversarial patch    (b) $L_\infty$-norm perturbation    (c) $L_2$-norm perturbation

Figure 20: Image captioning results for BLIP2 evaluated on MSCOCO dataset.

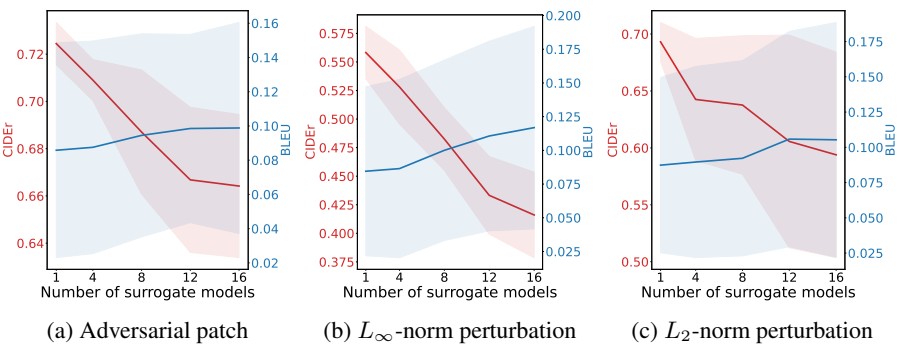

(a) Adversarial patch    (b) $L_\infty$-norm perturbation    (c) $L_2$-norm perturbation

Figure 21: Image captioning results for BLIP2 evaluated on Flicker30k dataset.

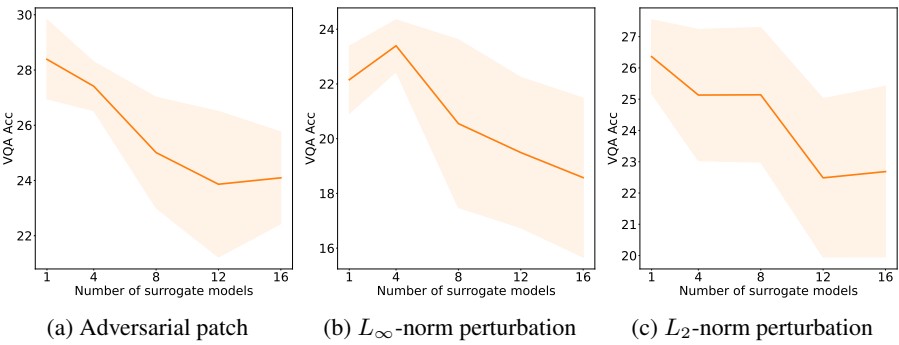

(a) Adversarial patch    (b) $L_\infty$-norm perturbation    (c) $L_2$-norm perturbation

Figure 22: VQA results for BLIP2 evaluated on OK-VQA dataset.

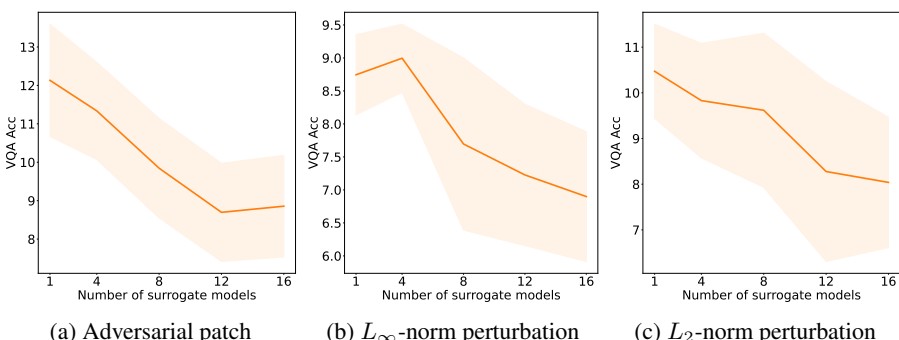

(a) Adversarial patch    (b) $L_\infty$-norm perturbation    (c) $L_2$-norm perturbation

Figure 23: VQA results for BLIP2 evaluated on VizWiz dataset.

We present qualitative examples for all three types of perturbations generated with different numbers of surrogate ensemble models in Figures 29 to 31, with the VLM responses obtained from LLaVA-7B. It can be observed that as the number of surrogate models increases, the output aligns more closely with the target text sentence, especially for $L_2$-norm perturbations. Quantitative evaluations are provided in Section 4.3.

All 150 TUAPs used in the main experiments from Sections 4.2 and 4.3 are displayed in Figures 32 to 34. Each row, from top to bottom, corresponds to TUAPs generated with E-1 to E-16, while each column shows the adversary's target text sentence, with details provided in Table 7. Interestingly, for the adversarial patch, most patterns resemble letters or phrases from the target text sentence, likely because these letters can easily meet the constraint of smaller patch sizes. A similar phenomenon is observed for $L_2$-norm perturbations. In contrast, $L_\infty$-norm bounded perturbations contain more semantic patterns for each target sentence. As the number of surrogate ensemble models increases (from the first to the last row), these patterns become more apparent. We believe this also contributes to stronger transferability.

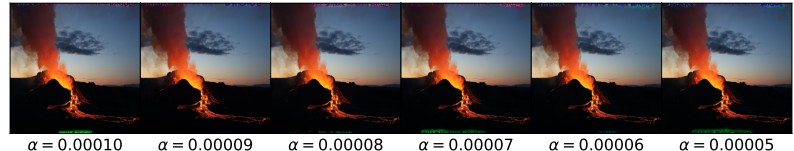

Figure 24: Visualizations of adversarial patches generated using a patch of a different size.

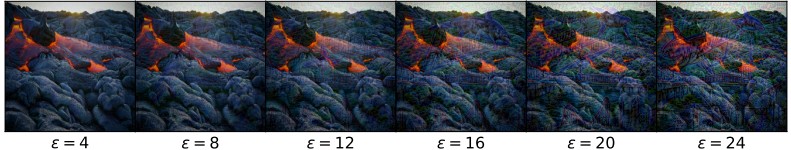

Figure 25: Visualizations of $L_\infty$-norm bounded perturbation generated using different $\epsilon$.

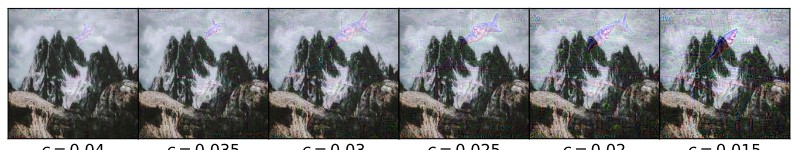

Figure 26: Visualizations of $L_2$-norm perturbation generated using different $c$.

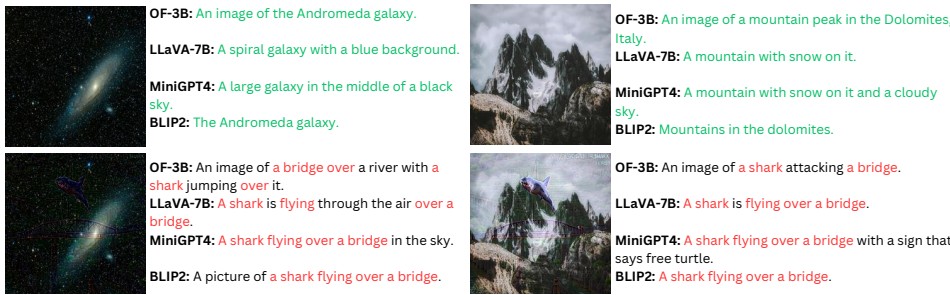

Figure 27: An illustration of the TUAP E-16 with $L_2$-norm perturbation. The adversary's target text sentence is *a great white shark flying over a bridge.* The top row contains clean images and texts generated from 4 VLMs. The bottom row contains images with the TUAPs and the corresponding response from VLMs. The prompt is the image and *briefly describe the image*.

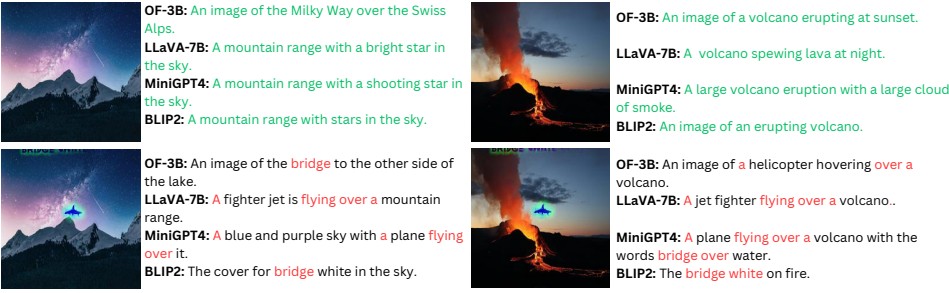

Figure 28: An illustration of the TUAP E-16 with adversarial patch perturbation. The adversary's target text sentence is *a great white shark flying over a bridge.*

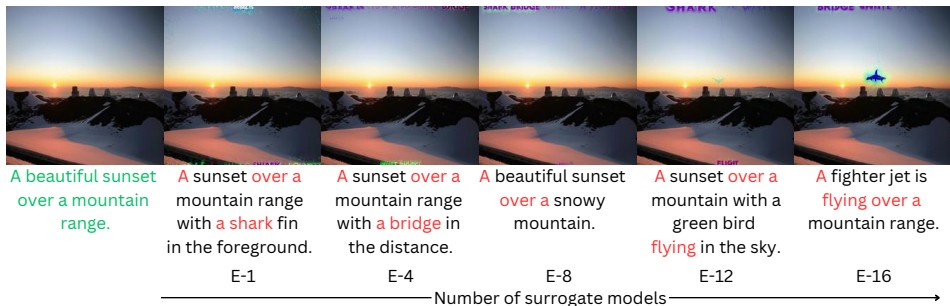

Figure 29: Qualitative examples for adversarial patches were generated using a different number of surrogate models. The response is obtained from LLaVA-7B. The adversary's target text sentence is *a great white shark flying over a bridge.*

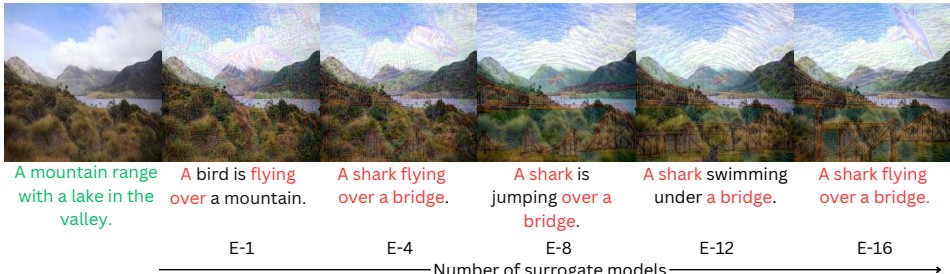

Figure 30: Qualitative examples for $L_\infty$-norm bounded perturbation were generated using a different number of surrogate models. The response is obtained from LLaVA-7B. The adversary's target text sentence is *a great white shark flying over a bridge.*

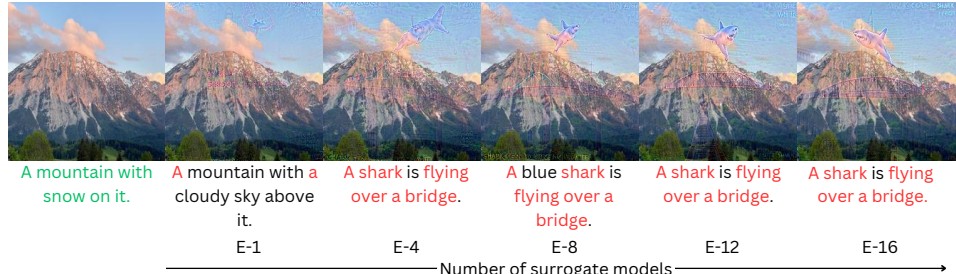

Figure 31: Qualitative examples for $L_2$-norm perturbation were generated using a different number of surrogate models. The response is obtained from LLaVA-7B. The adversary's target text sentence is *a great white shark flying over a bridge.*

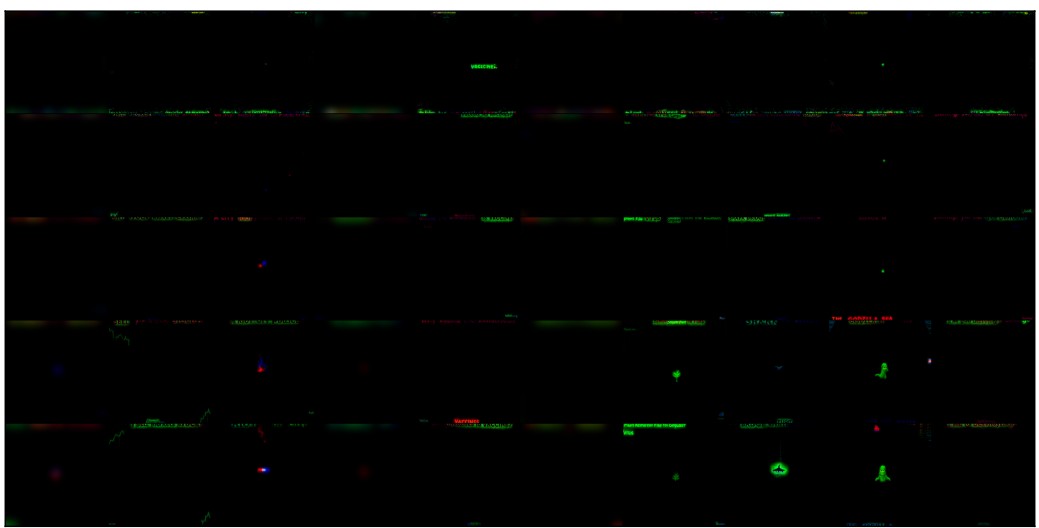

Figure 32: Visualization of adversarial patch evaluated in the experiments.

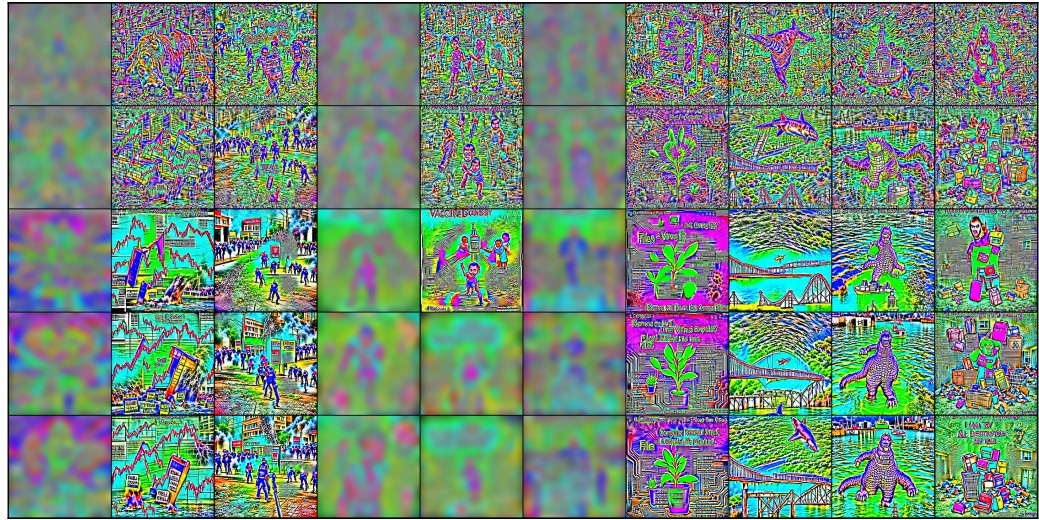

Figure 33: Visualization of $L_\infty$-norm bounded perturbations evaluated in the experiments.

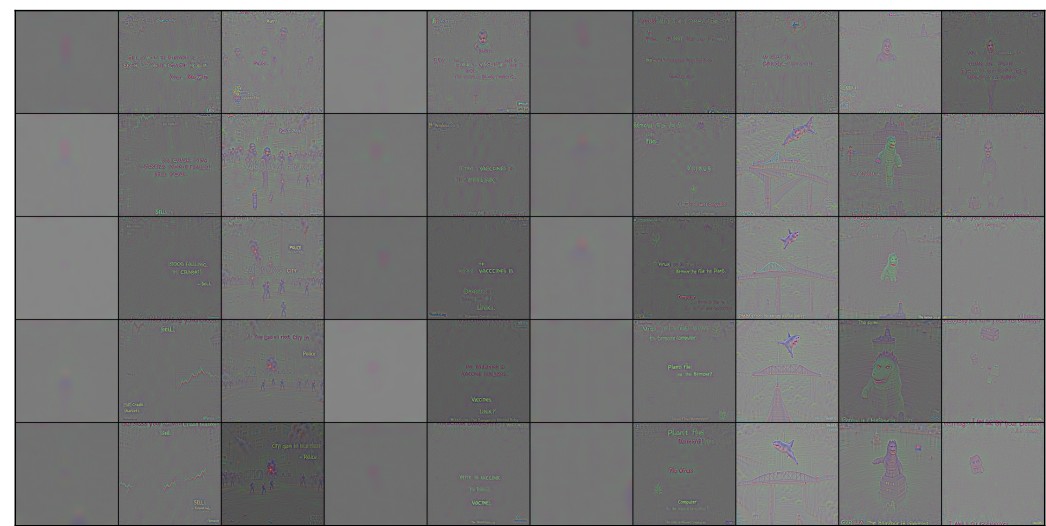

Figure 34: Visualization of $L_2$-norm perturbations evaluated in the experiments.

## C PYTORCH PSEUDOCODE

In Algorithm 2 to 5, we provide essential code for generating TUAPs with PyTorch-like pseudocode. For ensemble, we use the `Distributed Data Parallel` to distribute the parameters of the TUAP across each GPU. Each GPU will load 1 ensembled surrogate model according to its rank. For adversarial patch and $L_2$-norm perturbation, we use the standard optimization implementation provided by PyTorch. For $L_\infty$-norm bounded perturbation, we use PGD (Madry et al., 2018). However, instead of aggregating the gradient from different ranks and then applying the `sign` function, we apply the `sign` function prior to aggregation. We found this can slightly improve the performance of TUAP.

**Algorithm 2** Adversarial patch using pytorch pseudocode.

```python
class TUAP_Patch_module(nn.Module):
    def __init__(self, alpha, beta):
        super().__init__()
        self.alpha = alpha
        self.beta = beta
        # parameters
        delta = torch.FloatTensor(1, 3, 224, 224).uniform_
            (-0.5, 0.5)
        mask = torch.FloatTensor(1, 3, 224, 224).uniform_
            (-0.5, 0.5)
        self.mask_param = nn.Parameter(mask)
        self.delta_param = nn.Parameter(delta)

    def forward(self, images, surrogate_model, z_text_adv):
        # Add perturbation
        mask = (torch.tanh(self.mask_param) + 1) / 2
        delta = (torch.tanh(self.delta_param) + 1) / 2
        x_adv = delta * mask + (1 - mask) * images
        x_adv = torch.clamp(x_adv, 0, 1)
        # Extract image embedding
        z_image_adv = surrogate_model.encode_image(x_adv)
        # Compute loss
        loss = 2 - 2 * (z_image_adv * z_text_adv).sum(dim=1)
        norm = torch.norm(mask, p=1, dim=[1, 2, 3])
        tv = total_variation_loss(mask)
        tv += total_variation_loss(delta)
        loss = loss + self.alpha * norm + self.beta * tv
        return loss
```

**Algorithm 3** $L_\infty$-norm bounded perturbation using pytorch pseudocode.

```python
class TUAP_Linf_module(nn.Module):
    def __init__(self, epsilon):
        super().__init__()
        self.epsilon = epsilon
        self.delta = torch.FloatTensor(1, 3, 224, 224).
            uniform_(-epsilon, epsilon)
        self.delta = broadcast_tensor(self.delta, 0)

    def forward(self, images, surrogate_model, z_text_adv):
        # Add perturbation
        delta = torch.clamp(delta,-self.epsilon,self.epsilon)
        x_adv = images + delta
        x_adv = torch.clamp(x_adv, 0, 1)
        # Extract image embedding
        z_image_adv = surrogate_model.encode_image(x_adv)
        # Compute loss
        loss = 2 - 2 * (z_image_adv * z_text_adv).sum(dim=1)
        return loss
```

**Algorithm 4** $L_2$-norm perturbation using pytorch pseudocode.

```python
class TUAP_L2_module(nn.Module):
    def __init__(self, c):
        super().__init__()
        self.alpha = c
        # parameters
        delta = torch.FloatTensor(1, 3, 224, 224).uniform_
            (-0.5, 0.5)
        self.delta_param = nn.Parameter(delta)

    def forward(self, images, surrogate_model, z_text_adv):
        # Add perturbation
        delta = torch.tanh(self.delta_param)
        x_adv = images + delta
        x_adv = torch.clamp(x_adv, 0, 1)
        # Extract image embedding
        z_image_adv = surrogate_model.encode_image(x_adv)
        # Compute loss
        loss = 2 - 2 * (z_image_adv * z_text_adv).sum(dim=1)
        norm = torch.norm(mask, p=2, dim=[1, 2, 3])
        loss = loss + self.c * norm
        return loss
```

**Algorithm 5** Ensemble TUAP generation with DDP

```python
# model_list contains a list of surrogate models
# get_rank(): get the global rank in the distributed setting

tuap_module = DistributedDataParallel(tuap_module)

# optimizer for adversarial patch and L_2 perturbation
optimizer = Adam(tuap_module.parameters())
z_text_adv = surrogate_model.encode_text(target_text)
surrogate_model = model_list[get_rank()]

for images in data_loader:
    # Add perturbation and calculate loss
    loss = tuap_module(images, surrogate_model, z_text_adv)
    loss.backward()
    # Optimization step
    if type == "patch" or type == "l2":
        optimizer.step()
    else:
        # PGD for L_inf
        grad_sign = tuap_module.delta.grad.clone().sign()
        grad_sign = all_reduce_sum(grad_sign)
        grad_sign = grad_sign / get_world_size()
        tuap_module.delta = tuap_module.delta.clone() -
            step_size * grad_sign
        tuap_module.delta = torch.clamp(tuap_module.delta, -
            epsilon, epsilon)
```

