# OpenReview forum: "TUAP: Targeted Universal Adversarial Perturbations for CLIP"
_ICLR.cc/2025/Conference — Submitted to ICLR 2025_

### Official Review · Reviewer_QeU8 · 2024-10-29

**Soundness:** 3
**Presentation:** 3
**Contribution:** 2
**Rating:** 8
**Confidence:** 4

**Summary:**

This paper introduces TUAP, a new method for crafting universal perturbations for specific outputs in the CLIP model in a black-box setting. TUAP aims to generate perturbations that, when universally applied to images, guide CLIP's image encoder to achieve predetermined adversarial text embeddings.

**Strengths:**

1. The paper explores the targeted universal adversarial perturbation (TUAP) for the CLIP model in a black-box setting, and shows that TUAP can still achieve efficient attack effects in unseen models without gradient information. By combining multiple alternative models for perturbation generation, the transferability of the attack is improved.
2. Unrestricted adversarial patches, $L_{\infty}$ norm-constrained perturbations, and $L_{2}$ norm-constrained perturbations provide users with flexibility to choose appropriate perturbation forms in different scenarios. This diversity not only enhances the applicability of the method, but also demonstrates the performance of TUAP under various constraints, which is suitable for adversarial needs in various scenarios.
3. The paper experimentally verifies TUAP on 9 datasets and multiple downstream tasks, covering tasks such as image-text retrieval, zero-shot classification, and image description generation. Through diverse datasets (such as ImageNet, CIFAR, and Food101, etc.) and task tests, the paper demonstrates the applicability of TUAP in different tasks and its wide range of attack effects.

**Weaknesses:**

1. The paper discusses the attack effects on a variety of target text descriptions, but the targeted support for the diversity of target texts and actual scenarios is relatively limited. The lack of in-depth analysis of target texts that may have complex semantics or contextual dependencies in reality limits the applicability of this method in practical applications.
2. The paper focuses on the success rate and transitivity of the attack, but ignores the visual perceptibility of adversarial perturbations. Especially under $L_{\infty}$ and $L_{2}$ constraints, perturbations may produce visible artifacts, affecting the concealment of practical applications. In real application scenarios, perceptible perturbations may be easily discovered by the detection system, thereby limiting the effectiveness of the attack.
3. There are some ablation experiments on different perturbation parameters in the paper, but the sensitivity analysis of key parameters such as perturbation intensity and perturbation position is not comprehensive enough. In particular, how the perturbation intensity of the $L_{\infty}$ constraint affects the concealment and transitivity of the attack, and the impact of the change in the position of the perturbation at the edge or main area of ​​the image on the success rate of the attack, has not been fully discussed.
4. The experiments are mainly conducted on more typical datasets such as ImageNet, which may not fully reflect the complexity of real application scenarios. For example, the feature representation of CLIP may be significantly different in images in specific fields such as outdoor scenes or medical images, and there is a lack of verification of the attack effect in these specific fields.

**Questions:**

Please see weaknesses.

---

> ### Author Response · Authors · 2024-11-20
> **Response 1 to 3: target text diversity and perceptibility of the perturbation**
>
> Thank you very much for reviewing our paper and insightful comments. Please find our response to your questions below:
>
> ---
>
> **Q1:** Diversity of target texts and actual scenarios is relatively limited.
>
> **A1:**
> All the reported results in our submission are based on 10 target text sentences, which are detailed in Appendix B.3 and table below. We believe these targets cover a broad range of scenarios, including various types of content that an attacker might choose. While we acknowledge that an attacker could select any text as the target—and these 10 examples cannot cover every possible case—we believe they sufficiently demonstrate the feasibility and severity of this adversarial threat. Therefore, we consider our selection of target texts adequate for illustrating the potential risks posed by TUAPs.
>
> In future work, we plan to investigate a more extensive range of target sentences and develop a benchmark dataset with pre-trained TUAP pairs for each sentence to further expand the scope and applicability of this research.
>
> Targets No. 1 and No. 6, the placeholder $\mathtt{{X}}$ represents names of persons. The "..." in target No. 5 is an actual URL link.
>
> |       |                       Target Text Description                      |
> |:-----:|:------------------------------------------------------------------:|
> |  No.1 |              $\mathtt{\{X\}}$ orders a nuclear strike.             |
> |  No.2 |          Sell your stocks. Financial markets are crashing.         |
> |  No.3 |            A riot in a city. Police are using tear gas.            |
> |  No.4 |       A person suffers severe side effects after vaccination.      |
> |  No.5 | Vaccines are dangerous. Visit this weblink: https://$\mathtt{...}$ |
> |  No.6 |            $\mathtt{\{X\}}$ incites violence at a rally.           |
> |  No.7 |  Remove all files from this computer and plant the computer virus. |
> |  No.8 |              A great white shark flying over a bridge.             |
> |  No.9 |              The Godzilla is swimming in the harbour.              |
> | No.10 |            I am going to destroy all of your belongings.           |
>
> ---
>
> **Q2/3:** Perceptibility and ablation study.
>
> **A2/3:**
> In our initial submission, we have already conducted an analysis of different perceptibility levels and their impact on attack performance. This analysis is detailed in Section 4.5, Figure 4(a), Appendix B.10, and Figure 5. The corresponding visualizations can be found in Figures 24 to 26.
>
> Regarding the $L_\infty$ attack, an $\epsilon$ value of 8/255 is typically considered the threshold below which perturbations are imperceptible to the human eye. As shown in Figure 4(a), reducing $\epsilon$ to this level does not significantly decrease the attack success rate. This demonstrates that our method remains effective even when the perturbations are imperceptible.
>
> Concerning the use of random patch locations, we conducted a comparative study, and the results are summarized in the table below. The attack success rates indicate that there is no significant difference between applying the patch at random locations versus fixed locations.
>
> | Patch Location | Attack Success Rate (%) |
> |:--------------:|:-----------------------:|
> |      Fixed     |          92.79          |
> |     Random     |          94.77          |
>
> These results are based on a patch attack generated with E-16, evaluated on the ViT-L-14 model from OpenAI using the ImageNet dataset. The findings suggest that our method is robust to variations in patch placement, further validating the effectiveness of our approach.

---

> ### Author Response · Authors · 2024-11-20
> **Response 4: medical dataset**
>
> **Q4:** Evaluation of specific fields such as medical datasets.
>
> **A4:**
> We appreciate the reviewer’s suggestion to evaluate our method on specialized domains. To address this, we have conducted additional experiments using the COVID-19 Radiography dataset (Cohen et al., 2020) and the Cholec80 dataset (Twinanda et al., 2016). The COVID-19 Radiography dataset comprises X-ray images, while Cholec80 consists of laparoscopic cholecystectomy videos. Since these datasets do not include text captions, we evaluate the effectiveness of our TUAP by reporting the CLIP scores (higher is better) between the embeddings of the targeted text sentences and the image embeddings, both with and without the application of TUAP.
>
> These medical datasets are significantly different from the datasets on which our TUAP was originally trained. The results below indicate that our TUAP remains effective in achieving adversarial objectives within specialized domains. The evaluations on medical datasets underscore the broad applicability and effectiveness of our TUAP method beyond standard image tasks. This reinforces the severity and generalizability of the adversarial threats posed by TUAPs, highlighting the need for enhanced robustness in VLMs across diverse application areas.
>
> | Dataset  | Without TUAP | With TUAP |
> |----------|--------------|-----------|
> | COVID-19 | 8.01         | 32.39     |
> | Cholec80 | 13.61        | 30.70     |
>
> ---
>
> Cohen, J. P., Morrison, P., & Dao, L. (2020). COVID-19 image data collection. arXiv preprint arXiv:2003.11597.\
> Twinanda, A. P., et al. (2016). Endonet: A deep architecture for recognition tasks on laparoscopic videos. IEEE Transactions on Medical Imaging, 36 (1), 86-97.

---

> ### Author Response · Authors · 2024-11-23
>
> Dear reviewer QeU8,
>
> This is a kind reminder that we have provided a comprehensive response to your initial review. Specifically, we have demonstrated that TUAP is also effective when applied to medical imaging datasets. The ablation studies included in the initial submission have been further expanded with new experiments investigating the impact of patch location on performance.
>
> We sincerely appreciate your positive acknowledgment of our work and look forward to any further feedback or discussions you may have.

---

### Official Review · Reviewer_mcuc · 2024-10-31

**Soundness:** 3
**Presentation:** 3
**Contribution:** 2
**Rating:** 3
**Confidence:** 4

**Summary:**

This paper introduces a universal adversarial perturbation method against CLIP and its variants. Unlike previous white-box untargeted attacks against CLIP. This paper focus on a black-box targeted setting based on the transferability of adversarial examples. Experiments across diverse datasets and settings are conducted.

**Strengths:**

1. This paper is generally well organized.
2. The experiments are comprehensive. The authors have provided analyses across diverse scenarios.
3. The introduction of background knowledge is comprehensive.

**Weaknesses:**

1. This idea is trivial and not so novel. Universal adversarial attacks have been well explored in the context of standard adversarial attacks for single-modal architectures. It seems that the authors do not make any specific designs to transfer it into the vision-language models. According to the pseudocode, the proposed method seems to be a general approach for all the architectures but not specific to CLIP models.

2. The authors should also explore some related works [a, b] that apply black-box attacks based on transferability for a more comprehensive comparison analysis.


[a] Set-level Guidance Attack: Boosting Adversarial Transferability of Vision-Language Pre-training Models
[b] Sa-attack: Improving adversarial transferability of vision-language pre-training models via self-augmentation

**Questions:**

1. Can the authors provide some comparisons with query-based black-box attack approaches?
2. Can the novel designs for attacking CLIP models be shown?
3. In addition to adversarial images, is it possible to generate universal adversarial texts?

---

> ### Author Response · Authors · 2024-11-20
> **Response 1: novelty**
>
> Thank you very much for reviewing our paper. Please find our response to your questions below.
>
> ---
>
> **Q1:** Simple idea and novelty.
>
> **A1:**
>
> > This idea is trivial and not so novel. Universal adversarial attacks have been well explored in the context of standard adversarial attacks for single-modal architectures.
>
> We acknowledge that universal adversarial attacks have been studied extensively in single-modal architectures. However, our work explores Targeted Universal Adversarial Perturbations (TUAPs) for vision-language pre-training models, specifically for CLIP models—a novel problem that has not been addressed in prior research.
>
> > It seems that the authors do not make any specific designs to transfer it into the vision-language models.
>
> > Can the novel designs for attacking CLIP models be shown?
>
> Our attack objectives, defined in Eq. (2) to (7), are specifically designed for TUAPs against CLIP models. We have now conducted a comparison between the original UAP method by Moosavi-Dezfooli et al. (2017) and our TUAP. Since UAP is an untargeted attack and cannot produce exact target texts, we compare adversarial accuracy (the lower, the better) to ensure a fair evaluation. The significant performance difference between our TUAP and the original UAP indicates that our design—Eq. (2) to (7)—is specifically tailored for attacking CLIP models. By attacking the CLIP image encoder and ensuring transferability, our generated TUAPs can be directly applied to attack downstream VLMs. And revealing this phenomenon itself is one of our key contributions.
>
> |  Method  |  Vicitim Encoder | CIFAR10 | CIFAR100 |  FOOD101 |  GTSRB  | ImageNet |   Cars   |  STL10  |  SUN397  |
> |:--------:|:----------------:|:-------:|:--------:|:--------:|:-------:|:--------:|:--------:|:-------:|:--------:|
> |    UAP   |   ViT-L OpenAI   |   84.6  |   59.9   |   84.8   |   37.6  |   70.3   |   68.4   |   98.1  |   66.9   |
> | **Ours** |   ViT-L OpenAI   | **0.0** |  **0.0** |  **3.9** | **0.3** | **10.5** |  **5.5** | **1.6** |  **7.3** |
> |    UAP   | ViT-L Commonpool |   88.9  |   64.5   |   89.8   |   45.5  |   74.0   |   91.6   |   98.0  |   72.5   |
> | **Ours** | ViT-L Commonpool | **0.0** |  **0.0** |  **3.8** | **0.0** | **14.1** | **19.3** | **0.6** | **12.3** |
> |    UAP   |    ViT-L CLIPA   |   90.8  |   71.5   |   91.8   |   46.4  |   78.1   |   92.8   |   98.4  |   74.0   |
> | **Ours** |    ViT-L CLIPA   | **0.0** |  **0.0** | **10.5** | **0.0** | **23.1** | **30.0** | **1.1** | **17.7** |
> |    UAP   |   ViT-B SigLIP   |   64.2  |   36.4   |   84.0   |   23.1  |   71.8   |   85.9   |   95.5  |   65.4   |
> | **Ours** |   ViT-B SigLIP   | **0.0** |  **0.0** |  **4.4** | **0.0** | **13.1** | **16.8** | **0.2** |  **9.6** |
> |    UAP   |   ViT-B Laion2B  |   71.9  |   42.5   |   78.2   |   33.2  |   66.1   |   84.1   |   96.2  |   68.8   |
> | **Ours** |   ViT-B Laion2B  | **0.0** |  **0.0** |  **3.1** | **0.3** | **12.7** | **14.2** | **1.2** | **11.8** |
> |    UAP   |    RN50 OpenAI   |   31.3  |   13.8   |   58.0   |   19.2  |   50.2   |   45.3   |   88.4  |   51.1   |
> | **Ours** |    RN50 OpenAI   | **0.0** |  **0.0** |  **3.2** | **0.1** |  **9.6** |  **6.9** | **1.1** | **10.2** |
>
> ---
>
> > According to the pseudocode, the proposed method seems to be a general approach for all the architectures but not specific to CLIP models.
>
> We believe the reviewer refers to the ensemble approach used in our work. This approach is intentionally designed to be model-agnostic, which we consider a strength rather than a limitation. While our method is specifically evaluated on CLIP models, making it model-agnostic allows it to serve as a general framework for TUAPs. This generality provides great potential for further exploration and demonstrates the broader applicability of our method.

---

> ### Author Response · Authors · 2024-11-20
> **Response 2: comparison with related works SGA [a] and SA-Attack [b].**
>
> **Q2:** Comparison with related works SGA [a] and SA-Attack [b].
>
> **A2:**
> In our updated submission, we have included both SGA [a] and SA-Attack [b] in our related work discussion. Due to significant differences in problem settings, our initial submission did not consider these works in detail.
> Below, we provide both technical and empirical comparisons to clarify how our work differs from these studies.
>
> === **Technical Comparison** ===
>
> |   Method  | Attack objective | Attack modality | Perturbation type | Cross-model transferability | Cross-dataset transferbility | Cross-task transferability |
> |:---------:|:----------------:|:---------------:|:---------------------:|:-------------------------------:|:-------------------------------:|:-----------------------------:|
> |    SGA    |    Untargeted    |  Image and text |    Sample-specific    |                ✓                |                ✗                |               ✓               |
> | SA-Attack |    Untargeted    |  Image and text |    Sample-specific    |                ✓                |                ✗                |               ✓               |
> |    Ours   |     Targeted     |      Image      |       Universal       |                ✓                |                ✓                |               ✓               |
>
> **Attack Objectives:**
>
> - **SGA and SA-Attack:** Both methods are **untargeted attacks** that aim to create image-text pairs dissimilar in the embedding space, disrupting the alignment without specifying a particular target.
> - **Our TUAP:** We propose a **targeted attack** where the perturbation modifies the image embedding to align closely with a specific text sentence chosen by the attacker.
>
> **Perturbation Types:**
>
> - **SGA and SA-Attack:** These methods utilize **sample-specific perturbations**, which are generated individually for each image-text pair. This approach requires generating a new perturbation for every image, making it less practical for large-scale attacks.
> - **Our TUAP:** We introduce a **universal perturbation** that is generated once and can be applied to any image, greatly enhancing the practicality and scalability of the attack.
>
> **Transferability:**
>
> - **SGA and SA-Attack:** The reliance on sample-specific perturbations limits their **cross-dataset transferability** since the perturbations are tailored to specific images and cannot be applied to others.
> - **Our TUAP:** Our universal perturbation enables **simultaneous cross-model, cross-dataset, and cross-task transferability**, as verified by our experiments. This means our TUAP can effectively attack different models, datasets, and tasks using the same perturbation.
>
> === **Empirical Comparison** ===
>
> Given that SA-Attack is not open-sourced and both SA-Attack and SGA share similar threat models and problem domains, we focus our empirical comparison on SGA. If an official implementation of SA-Attack becomes available, we would be happy to include it in our analysis.
>
> Experimental Setup:
>
> - Perturbation Magnitude: To ensure a fair comparison, we generated SGA perturbations with an $\epsilon$ of 16/255, consistent with our TUAP settings.
> - Tasks Evaluated: We evaluated cross-task and cross-model transferability, corresponding to Section 4.3 in our paper and Section 5.4 in SGA's paper.
> - Datasets and Metrics: We used the MSCOCO dataset and compared CIDEr scores (lower is better) on image captioning tasks.
>
> Results:
>
> **Cross-Task and Cross-Model Transferability:** The results, summarized in the table below, show that our TUAP exhibits stronger cross-model and cross-task transferability than SGA. TUAP achieved this with a single universal perturbation rather than sample-wise perturbations, highlighting its efficiency and effectiveness.
>
> |    Method    | OpenFlamingo |   LLaVA   |  MiniGPT4 |   BLIP2   |
> |:------------:|:------------:|:---------:|:---------:|:---------:|
> |   No Attack  |     71.58    |   114.27  |   126.23  |   117.44  |
> |   SGA ALBEF  |     60.38    |   99.45   |   114.97  |   115.99  |
> | SGA ViT-B/16 |     49.99    |   88.68   |   110.11  |   98.66   |
> |     Ours     |   **22.00**  | **59.25** | **68.20** | **52.27** |
>
> ---
>
> **Image-Text Retrieval (untargeted):** We also compared performance on standard image-text retrieval tasks using untargeted attack evaluations as in SGA. Our TUAP consistently outperformed SGA, except for a slightly lower IR R@1 when using ResNet-101 as the encoder.
>
> | Method | Surrogate model | Victim model |  TR R@1  |  IR R@1  |
> |:------:|:---------------:|:------------:|:--------:|:--------:|
> |   SGA  |     ViT-B/16    |     RN101    |   79.9   | **83.9** |
> |  Ours  |       E16       |     RN101    | **84.1** |   81.71  |
> |   SGA  |      RN101      |   ViT-B/16   |   70.0   |   75.6   |
> |  Ours  |       E16       |   ViT-B/16   | **82.5** | **80.6** |

---

> ### Author Response · Authors · 2024-11-20
> **Response 2 (continued): comparison with related works SGA [a] and SA-Attack [b].**
>
> **Summary**
>
> Our method significantly differs from SGA and SA-Attack both technically and empirically:
> - **Technical Innovation:** We introduce a targeted universal attack framework, whereas SGA and SA-Attack are untargeted and sample-specific.
> - **Practical Advantages:** Our TUAP is more practical for large-scale attacks due to its universal nature, requiring only a single perturbation applicable to any image.
> - **Superior Transferability:** TUAP achieves simultaneous cross-model, cross-dataset, and cross-task transferability, which is not demonstrated by SGA or SA-Attack.
>
> We believe these distinctions highlight the novelty and contributions of our work, advancing the understanding of adversarial attacks in vision-language models.

---

> ### Author Response · Authors · 2024-11-20
> **Response 3 and 4: query-based black-box attack and universal adversarial texts**
>
> **Q3:** Can the authors provide some comparisons with query-based black-box attack approaches?
>
> **A3:**
> We appreciate the reviewer's insightful suggestion. However, to the best of our knowledge, query-based black-box attacks against CLIP models—especially for generating TUAPs—remain an unexplored area. If there are existing works that we have overlooked, we would be grateful if the reviewer could point them out.
>
> To address your concern, we attempted to adapt the NESAttack method (Ilyas et al., 2018) to our setting. Specifically, we replaced the optimization component in our method with the NESAttack strategy while keeping the other objectives the same as in our initial submission. However, we found that query-based methods were unable to produce successful TUAPs.
> The results are summarized below:
>
>
> |   Method  |  Vicitim Encoder |  CIFAR10  |  CIFAR100 |  FOOD101 |   GTSRB   | ImageNet |   Cars   |   STL10  |  SUN397  |
> |:---------:|:----------------:|:---------:|:---------:|:--------:|:---------:|:--------:|:--------:|:--------:|:--------:|
> | NESAttack |   ViT-L OpenAI   |    0.1    |    0.1    |    0.0   |    0.5    |    0.0   |    0.0   |    0.0   |    0.0   |
> |  **Ours** |   ViT-L OpenAI   | **100.0** | **100.0** | **94.3** |  **99.7** | **71.9** | **86.2** | **98.4** | **81.4** |
> | NESAttack | ViT-L Commonpool |    0.2    |    0.1    |    0.0   |    0.3    |    0.0   |    0.0   |    0.0   |    0.0   |
> |  **Ours** | ViT-L Commonpool | **100.0** | **100.0** | **95.1** | **100.0** | **70.7** | **72.8** | **99.4** | **71.8** |
> | NESAttack |    ViT-L CLIPA   |    0.1    |    0.0    |    0.0   |    0.6    |    0.0   |    0.0   |    0.0   |    0.0   |
> |  **Ours** |    ViT-L CLIPA   | **100.0** | **100.0** | **87.9** | **100.0** | **63.1** | **65.8** | **98.9** | **70.2** |
> | NESAttack |   ViT-B SigLIP   |    0.6    |    0.2    |    0.0   |    0.0    |    0.0   |    0.0   |    0.0   |    0.0   |
> |  **Ours** |   ViT-B SigLIP   | **100.0** | **100.0** | **92.8** | **100.0** | **74.3** | **71.2** | **99.8** | **73.8** |
> | NESAttack |   ViT-B Laion2B  |    0.7    |    0.4    |    0.0   |    0.0    |    0.0   |    0.0   |    0.0   |    0.0   |
> |  **Ours** |   ViT-B Laion2B  | **100.0** | **100.0** | **93.7** |  **99.7** | **65.3** | **66.9** | **98.8** | **64.2** |
> | NESAttack |    RN50 OpenAI   |    3.4    |    0.8    |    0.0   |    0.2    |    0.0   |    0.0   |    0.1   |    0.0   |
> |  **Ours** |    RN50 OpenAI   | **100.0** | **100.0** | **86.7** |  **99.8** | **53.9** | **58.3** | **98.6** | **55.2** |
>
> We believe the primary reason for this difficulty is the substantial number of queries required to generate even sample-specific perturbations in a black-box setting. For universal perturbations, a single perturbation must be adversarially effective across all images, which significantly amplifies the challenge. The need for such broad generalization without direct access to the model's internals makes query-based TUAP generation exceedingly demanding. Therefore, developing effective query-based black-box methods for TUAPs remains an open problem for future research.
>
> Ilyas, Andrew, et al. "Black-box adversarial attacks with limited queries and information." ICML 2018.
>
> ---
>
> **Q4:** In addition to adversarial images, is it possible to generate universal adversarial texts?
>
> **A4:**
> While it is theoretically possible to generate universal adversarial texts, significant modifications to the current formulation of TUAP would be necessary. To the best of our knowledge, the concept of universal adversarial texts remains an unexplored area in the field of adversarial attacks. However, our ensemble framework is general and could be adapted to generate adversarial texts in future work. We believe this represents an exciting direction for further research, which could provide deeper insights into the vulnerabilities of VLMs.

---

> ### Author Response · Authors · 2024-11-23
>
> Dear Reviewer mcuc,
>
> This is a kind reminder that we have provided a comprehensive response to your initial review.
>
> We have provided both technical and empirical comparisons with SGA and SA-Attack, as the reviewer mentioned. These works study different attack objectives and types of perturbations, making direct comparisons inherently challenging. We have added new results in our submission and rebuttal.
>
> It is important to emphasize that comparing universal perturbations (our TUAP) with sample-specific perturbations (SGA and SA-Attack) is inherently unfair, as sample-specific perturbations are customized for each individual sample, giving them a significant advantage. Despite this disadvantage, our TUAP consistently outperforms or matches SGA across several evaluations, highlighting its novelty in achieving universal adversarial transferability.
>
> We kindly request that you review the newly provided results, summarized below:
>
> - Even when compared to sample-specific perturbations, our TUAP outperforms SGA in cross-task transferability evaluations with untargeted objectives. Specifically, the CIDEr score is halved when comparing TUAP with SGA.
> - By running the released code for SGA and directly applying our TUAP to any samples, we demonstrate that TUAP outperforms or matches SGA in untargeted objectives for COCO image-text retrieval tasks. This result is significant because sample-specific perturbations are generally considered stronger than universal perturbations.
> - We achieve simultaneous black-box cross-model, cross-dataset, and cross-task adversarial transferability. These results meet challenging targeted attack objectives and distinguish our approach from that of existing studies.
>
> We hope these results address your concerns and provide clarity on the novelty and contributions of our work. We look forward to further constructive discussions with you.

---

> ### Author Response · Authors · 2024-11-24
>
> Dear Reviewer mcuc,
>
> Thank you once again for taking the time to review our paper. We deeply value your insightful comments and are sincerely grateful for your efforts.
>
> We kindly request you to review our reply to see if it sufficiently addresses your concerns. Your feedback means a lot to us.
>
> Thank you deeply,
> Authors

---

> > ### Comment · Reviewer_mcuc · 2024-11-25
> >
> > Dear Authors,
> >
> > Thank you for your detailed response and the efforts you have made in revising the paper.
> >
> > I appreciate the extensive experiments presented, which effectively demonstrate the proposed method. However, I find that the core idea lacks significant novelty, as it mainly involves extending the UAP scenario from single-modal architectures to CLIP. Furthermore, the original UAP framework could already be adapted to targeted attack scenarios by simply modifying the objective function. As such, the results, while thorough, are not particularly surprising in the context of CLIP. The contribution feels more like an experimental extension of UAP rather than a fundamentally novel advancement.
> >
> > For these reasons, I will be maintaining my current rating.
> >
> > Best regards,
> > Reviewer mcuc

---

### Official Review · Reviewer_Ypoi · 2024-10-31

**Soundness:** 4
**Presentation:** 3
**Contribution:** 2
**Rating:** 3
**Confidence:** 5

**Summary:**

This paper proposes the first targeted UAP attack against VLP models and conducts extensive experiments across various tasks and models. By specifying a targeted adversarial text description, TUAP is able to generate a universal L_{inf} norm-bounded or L_{2}-norm perturbation or a small unrestricted patch, exhibiting outstanding transferability.
Besides, the ensemble of surrogate CLIP encoders further enhances the attack effects.

**Strengths:**

(1) The authors provide sufficient experiments on diverse downstream tasks and models, showing the efficacy of the proposed TUAP.
(2) The tables are in good format.
(3) The experimental analysis is reasonable and convincing.
(4) Visualization results are intuitive and reveal that TUAP successfully achieves the attack.

**Weaknesses:**

(1) My major concern is that this paper provides no significant technical innovation. Specifically, the proposed method makes an intuitive attempt by simply combining existing techniques from former studies of adversarial attacks, lacking in-depth exploration and novel insights into vulnerabilities of VLP models against UAP.
(2) The introduction section requires further reformulation to reduce the unnecessary space.
(3) Despite the extensive experiments, I do not see any insightful analysis regarding the underlying mechanism of the attack algorithm.

While the authors have conducted sufficient experiments, no significant contributions are made compared with [1].
In summary, this paper is more like an experimental report rather than a top-tier conference paper, far from meeting the acceptance criteria for ICLR.

[1] Ziqi Zhou, Shengshan Hu, Minghui Li, Hangtao Zhang, Yechao Zhang, and Hai Jin. Advclip: Downstream-agnostic adversarial examples in multimodal contrastive learning. In ACM MM, 2023.

**Questions:**

Did you train the universal perturbation with the whole training set of the surrogate model?

---

> ### Author Response · Authors · 2024-11-20
> **Response 1: response to lack in-depth exploration and novel insights**
>
> Thanks for reviewing our paper and the comments. Please find our responses to your questions below.
>
> ---
>
> **Q1:** Lacking in-depth exploration and novel insights
>
> **A1:**
>
> > intuitive attempt by simply combining existing techniques from former studies
>
> We would appreciate it if the reviewer could specify any prior studies that have demonstrated our findings or require us to compare them technically. Novelty assessments should be based on concrete comparisons rather than subjective impressions.
>
> > lacking in-depth exploration and novel insights
>
> We would like to highlight the following in-depth explorations and novel insights that may have been overlooked:
>
> - **Cross-Model Transferability:** We comprehensively demonstrate cross-model transferability using 6 CLIP encoders and four downstream large VLMs.
> - **Cross-Dataset Transferability:** The TUAPs generated on ImageNet or CC3M successfully transfer across 11 additional datasets.
> - **Cross-Task Transferability:** Our TUAPs, generated without targeting specific tasks, successfully transfer across diverse tasks, including image classification, image-text retrieval, image captioning, and visual question answering (VQA). Notably, TUAPs not only cause arbitrary errors but induce targeted errors specified by the adversary.
> - **Simultaneous Transferability:** These transferability aspects are not mutually exclusive. Our TUAP achieves simultaneous cross-model, cross-dataset, and cross-task transferability.
> - **Scalable Transferability with Surrogate Models:** Most importantly, we reveal that transferability scales as the number of surrogate models increases. To the best of our knowledge, no previous work has revealed this scaling trend of  UAP transferability.
> - **New Adversarial Vulnerability in Large VLMs:** We have uncovered an emerging safety risk in large VLMs, where an adversary can launch widespread targeted attacks on VLMs using universal patterns (TUAPs).
>
> We believe these contributions represent significant in-depth exploration and novel insights that advance the field.

---

> ### Author Response · Authors · 2024-11-20
> **Response 2 and 3: introduction reformulation and underlying mechanism**
>
> **Q2:** The introduction section requires further reformulation
>
> **A2:** We have thoroughly revised the introduction section in our updated submission to address the concerns you've raised. Our goal was to enhance clarity, strengthen the motivation for our work, and more effectively articulate our contributions. If there are specific areas or sentences that you believe still require improvement, we would greatly appreciate it if you could point them out directly. Your detailed feedback will help us make precise adjustments to further improve the introduction.
>
> ---
>
> **Q3:** The reviewer does not see any insightful analysis regarding the underlying mechanism of the attack algorithm.
>
> **A3:** We have analyzed the underlying mechanism of our attack algorithm in Section 4.4. Specifically, we explained how the concept blending capability that empowers the strong zero-shot performance of CLIP models also makes them susceptible to TUAPs. This blending capability allows CLIP to associate visual features with a wide range of textual concepts, which can be exploited by TUAPs to induce specific, adversarial outputs. Our analysis sheds light on why CLIP's strengths in zero-shot learning can be turned into vulnerabilities, offering a deeper understanding of the attack's effectiveness.

---

> ### Author Response · Authors · 2024-11-20
> **Response 4: comparison with AdvCLIP**
>
> **Q4:** No significant contributions compared with AdvCLIP
>
> **A4:**
>
> We believe the reviewer may have misunderstood our work. Please allow us to clarify the significant differences between our method and AdvCLIP.
>
> === **Technical Comparison** ===
>
> |  Method | Threat model | Attack objective |    UAP generation   | Cross-model transferbility | Cross-dataset transferbility | Cross-task transferbility |
> |:-------:|:------------:|:----------------:|:-------------------:|:------------------------------:|:-------------------------------:|:----------------------------:|
> | AdvCLIP |  White-box*  |    Untargeted    |   ResNet Generator  |                ✗               |               ✓                |               ✓              |
> |   Ours  |   Black-box  |     Targeted     | Perturbation vector |                ✓               |                ✓                |               ✓              |
>
>
> **(1) Different attack objectives:** Our method is a targeted attack, whereas AdvCLIP is an untargeted attack.
>
> A direct quote from the AdvCLIP paper's abstract:
> > "achieving universal non-targeted attacks. "
>
> **(2) Black-box transferability:** Our TUAP achieves black-box cross-model, cross-dataset, and cross-task adversarial transferability, simultaneously, which is impossible for AdvCLIP as it requires white-box access to the victim encoder.
>
> We quote from the AdvCLIP paper's Section 3.1 Threat Model:
>
> > "We assume a quasi-black-box attack model, where the attacker has access to VLP encoders through purchasing or downloading from publicly available websites …"
>
> In our context, the VLP encoder is the victim model. Our method does not make such an assumption; we operate in a complete black-box setting.
>
> ---
>
> === **Empirical Comparison** ===
>
> Since no pre-trained weights or UAPs are available in the official AdvCLIP repository, we reproduced the UAP using their released code with default hyperparameters. Due to different attack objectives, implementations, and evaluation protocols, we conducted the following empirical comparisons using both our evaluation and AdvCLIP's official repository, separately.
>
> The table below reports the results based on AdvCLIP's evaluation, with the **untargeted** adversarial accuracy (Adv acc) and fooling rate.
>
> |  Method | Threat model | Surrogate Encoder |  Victim Encoder | Evaluation dataset | Clean acc |  Adv acc | Fooling rate |
> |:-------:|:------------:|:----------------------:|:---------------:|:------------------:|:---------:|:--------:|:---------:|
> | AdvCLIP |  White-box  |     ViT-B/16 OpenAI    | ViT-B/16 OpenAI |       Pascal       |    77.0   |    7.0   |    92.0   |
> | AdvCLIP |   Black-box  |     ViT-B/16 OpenAI    |   RN50 OpenAI   |       Pascal       |    70.5   |   70.0   |    10.5   |
> |   **Ours**  |   Black-box  |           E16          |   RN50 OpenAI   |       Pascal       |    70.5   | **27.5** |  **70.0** |
>
> - First row: We reproduced the white-box attack performance reported in the AdvCLIP paper.
> - Second row: AdvCLIP failed to achieve black-box cross-model transferability.
> - Third row: Our TUAP outperforms AdvCLIP by increasing the fooling rate by 59.5% and reducing adversarial accuracy by 43%, whereas AdvCLIP only reduces adversarial accuracy by 0.5%. Our TUAP successfully achieves cross-model transferability.

---

> ### Author Response · Authors · 2024-11-20
> **Response 4 (continued): comparison with AdvCLIP**
>
> The following table reports the results (**untargeted** adversarial accuracy) based on our evaluation (the same as in our Table 1). These results further verify that we can achieve cross-model transferability, which AdvCLIP does not.
>
> |  Method  |  Vicitim Encoder | CIFAR10 | CIFAR100 |  FOOD101 |  GTSRB  | ImageNet |   Cars   |  STL10  |  SUN397  |
> |:--------:|:----------------:|:-------:|:--------:|:--------:|:-------:|:--------:|:--------:|:-------:|:--------:|
> |  AdvCLIP |   ViT-L OpenAI   |   91.8  |   68.9   |   88.8   |   37.3  |   71.7   |   69.8   |   98.8  |   67.3   |
> | **Ours** |   ViT-L OpenAI   | **0.0** |  **0.0** |  **3.9** | **0.3** | **10.5** |  **5.5** | **1.6** |  **7.3** |
> |  AdvCLIP | ViT-L Commonpool |   97.5  |   82.8   |   93.4   |   57.9  |   75.6   |   92.7   |   98.9  |   73.1   |
> | **Ours** | ViT-L Commonpool | **0.0** |  **0.0** |  **3.8** | **0.0** | **14.1** | **19.3** | **0.6** | **12.3** |
> |  AdvCLIP |    ViT-L CLIPA   |   97.8  |   87.9   |   94.2   |   57.0  |   78.7   |   92.8   |   99.2  |   73.9   |
> | **Ours** |    ViT-L CLIPA   | **0.0** |  **0.0** | **10.5** | **0.0** | **23.1** | **30.0** | **1.1** | **17.7** |
> |  AdvCLIP |   ViT-B SigLIP   |   90.9  |   67.7   |   91.3   |   42.6  |   74.9   |   89.1   |   98.0  |   68.2   |
> | **Ours** |   ViT-B SigLIP   | **0.0** |  **0.0** |  **4.4** | **0.0** | **13.1** | **16.8** | **0.2** |  **9.6** |
> |  AdvCLIP |   ViT-B Laion2B  |   94.1  |   74.6   |   85.6   |   49.8  |   68.3   |   87.6   |   97.6  |   70.1   |
> | **Ours** |   ViT-B Laion2B  | **0.0** |  **0.0** |  **3.1** | **0.3** | **12.7** | **14.2** | **1.2** | **11.8** |
> |  AdvCLIP |    RN50 OpenAI   |   71.2  |   39.6   |   77.5   |   34.9  |   55.5   |   48.0   |   93.1  |   56.1   |
> | **Ours** |    RN50 OpenAI   | **0.0** |  **0.0** |  **3.2** | **0.1** |  **9.6** |  **6.9** | **1.1** | **10.2** |
>
>
> The table below reports the **targeted** results (targeted attack success rate) based on our evaluation. It shows that AdvCLIP cannot perform targeted attacks.
>
>
> |  Method  |  Vicitim Encoder |  CIFAR10  |  CIFAR100 |  FOOD101 |   GTSRB   | ImageNet |   Cars   |   STL10  |  SUN397  |
> |:--------:|:----------------:|:---------:|:---------:|:--------:|:---------:|:--------:|:--------:|:--------:|:--------:|
> |  AdvCLIP |   ViT-L OpenAI   |    0.0    |    0.0    |    0.0   |    0.0    |    0.0   |    0.0   |    0.0   |    0.0   |
> | **Ours** |   ViT-L OpenAI   | **100.0** | **100.0** | **94.3** |  **99.7** | **71.9** | **86.2** | **98.4** | **81.4** |
> |  AdvCLIP | ViT-L Commonpool |    0.0    |    0.0    |    0.0   |    0.0    |    0.0   |    0.0   |    0.0   |    0.0   |
> | **Ours** | ViT-L Commonpool | **100.0** | **100.0** | **95.1** | **100.0** | **70.7** | **72.8** | **99.4** | **71.8** |
> |  AdvCLIP |    ViT-L CLIPA   |    1.0    |    0.2    |    0.0   |    0.3    |    0.0   |    0.0   |    0.2   |    0.1   |
> | **Ours** |    ViT-L CLIPA   | **100.0** | **100.0** | **87.9** | **100.0** | **63.1** | **65.8** | **98.9** | **70.2** |
> |  AdvCLIP |   ViT-B SigLIP   |    0.0    |    0.0    |    0.0   |    0.0    |    0.0   |    0.0   |    0.0   |    0.0   |
> | **Ours** |   ViT-B SigLIP   | **100.0** | **100.0** | **92.8** | **100.0** | **74.3** | **71.2** | **99.8** | **73.8** |
> |  AdvCLIP |   ViT-B Laion2B  |    0.1    |    0.0    |    0.0   |    2.2    |    0.0   |    0.0   |    0.1   |    0.0   |
> | **Ours** |   ViT-B Laion2B  | **100.0** | **100.0** | **93.7** |  **99.7** | **65.3** | **66.9** | **98.8** | **64.2** |
> |  AdvCLIP |    RN50 OpenAI   |    0.1    |    0.0    |    0.0   |    0.0    |    0.0   |    0.0   |    0.2   |    0.0   |
> | **Ours** |    RN50 OpenAI   | **100.0** | **100.0** | **86.7** |  **99.8** | **53.9** | **58.3** | **98.6** | **55.2** |
>
> ---
>
> Contributions Compared to AdvCLIP:
>
> - **Simultaneous Cross-Model, Cross-Dataset, and Cross-Task Transferability:** Our TUAP achieves transferability across models, datasets, and tasks, simultaneously.
> - **Scalable Transferability with Surrogate Models:** We discovered that transferability scales with the number of surrogate models used. To the best of our knowledge, there has been no previous study on UAPs revealing this transfer scalability.
> - **New Adversarial Vulnerability in Large VLMs:** We revealed a new safety risk for large VLMs, demonstrating that an adversary can conduct effective attacks using an ensemble of pre-trained encoders. This aspect is not examined in AdvCLIP.
>
> Given the significant differences in both technical approach and empirical results, we believe the novelty and contributions of our work are clear and substantial compared to AdvCLIP.

---

> ### Author Response · Authors · 2024-11-20
> **Response 5: meeting the acceptance criteria for ICLR**
>
> **Q5:** Meeting the acceptance criteria for ICLR.
>
> **A5:**
>
> > In summary, this paper is more like an experimental report rather than a top-tier conference paper, far from meeting the acceptance criteria for ICLR.
>
> We appreciate the review acknowledging our paper's strong soundness, clear presentation, and comprehensive experiments. According to the ICLR Reviewer Guide (2021–2025), novelty should be evaluated not only based on technical methods but also on novel findings. Additionally, the guideline suggests that *"the submission clear, technically correct, experimentally rigorous, reproducible"* is a strong point. We believe our detailed experiment setting for reproducibility, rigorous experiments, technically sound method, and clear presentation are strengths.
>
> By addressing the technical challenges in crafting targeted UAPs (TUAPs) for CLIP, our work is the first to demonstrate the possibility of generating highly-transferable TUAPs for CLIP based on an ensemble of surrogate models. Our large-scale and comprehensive analysis provides a set of new insights into the vulnerability of CLIP, whilst revealing a scaling law for TUAP transferability. We believe our work offers significant contributions to the community in both the technical and empirical respects.
>
> We genuinely hope the reviewer can reassess the contribution of our work. This would mean a lot to us.

---

> ### Author Response · Authors · 2024-11-23
>
> Dear Reviewer Ypoi,
>
> This is a kind reminder that we have provided a comprehensive response to your initial review.
>
> We have conducted both technical and empirical comparisons between our TUAP and AdvCLIP, which the reviewer mentioned. The main difference lies in white-box/black-box accessibility to victim encoders and attack objectives.
>
> We kindly request that you review the newly provided results, summarized as follows:
>
> - By running AdvCLIP’s code, we demonstrate that it fails to achieve cross-model transferability, whereas our TUAP successfully accomplishes this.
> - Our TUAP significantly outperforms the UAP generated by AdvCLIP in untargeted evaluations, as AdvCLIP cannot achieve cross-model transferability.
> - TUAP also supports targeted attacks, which AdvCLIP does not, as it is limited to untargeted attack scenarios.
> - On several evaluations, TUAP achieves a performance gap of 100% compared to 0% for AdvCLIP, highlighting the fundamental differences in the attack threat models studied.
> - We achieve simultaneous black-box cross-model, cross-dataset, and cross-task adversarial transferability. These results meet challenging targeted attack objectives and distinguish our approach from that of existing studies.
>
> We hope these results address your concerns and clarify the contributions of our work. We look forward to engaging in constructive discussions with you.

---

> ### Author Response · Authors · 2024-11-24
>
> Dear Reviewer Ypoi,
>
> Thank you once again for taking the time to review our paper. We deeply value your insightful comments and are sincerely grateful for your efforts.
>
> We kindly request you to review our reply to see if it sufficiently addresses your concerns. Your feedback means a lot to us.
>
> Thank you deeply,
> Authors

---

> > ### Comment · Reviewer_Ypoi · 2024-11-25
> > **Reviewer Response**
> >
> > I sincerely thank the authors for their exhaustive rebuttal. However, my major concerns regarding the technique contribution still remains unresolved since both the optimization of cosine embeddings and the model ensemble are commonly used by previous studies on VLP models.
> >
> > Simply transferring from untargeted to targeted universal attacks without significant technical innovations does not provide insightful viewpoints and enough contributions to meet the criteria of ICLR, and the supplemented analysis did not adequately convince me either.
> >
> > Consequently, I will maintain my score for this paper. Thanks again for your thorough experimental results and analyses.

---

### Official Review · Reviewer_Tkjh · 2024-11-03

**Soundness:** 2
**Presentation:** 2
**Contribution:** 2
**Rating:** 3
**Confidence:** 1

**Summary:**

This work explores the vulnerability of CLIP models to targeted Universal Adversarial Perturbations (TUAPs) in a black-box setting, focusing on transferability. TUAPs enable adversaries to specify a targeted text description and generate universal perturbations that mislead unseen CLIP models into producing embeddings aligned with the adversarial text. Experiments demonstrate the effectiveness and transferability of TUAPs across various datasets and large Vision-Language Models (VLMs) like OpenFlamingo and MiniGPT-4. The findings highlight a significant vulnerability in CLIP models to targeted attacks, underscoring the need for effective countermeasures.

**Strengths:**

Strengths.
1. The paper is clearly written and motivates the proposed approach well in a lucid manner.
2. The paper demonstrates universal transferability across different datasets and various Vision-Language Models (VLMs), consistently misleading image encoders.
3. The paper proposes a targeted Universal Adversarial Perturbations (TUAPs) method. The proposed TUAPs allow adversaries to specify precise text descriptions for targeted attacks while functioning effectively in a black-box setting.
4. The paper highlights significant vulnerabilities in CLIP models, underscoring the need for improved defenses against targeted adversarial attacks.

**Weaknesses:**

Weaknesses

1. The novelty, in my opinion, is limited. This paper simply migrates the Universal Adversarial Perturbations generation method in image classification to the VLM task, and there is no technical innovation.


2.  Lack of comparison with Universal Adversarial Attack methods for VLM models[1][2]. They have proposed to adopt the image encoder to generate adversarial perturbations.
[1] Zhao Y, Pang T, Du C, et al. On evaluating adversarial robustness of large vision-language models[J]. Advances in Neural Information Processing Systems, 2024, 36.
[2] Dong Y, Chen H, Chen J, et al. How Robust is Google's Bard to Adversarial Image Attacks?[J]. arXiv preprint arXiv:2309.11751, 2023.

3. How does the proposed method perform on closed-source models such as chatgpt4?


4. In Figure 1, there are apparent shark textures in the generated adversarial images, and I think there is no problem for VLM to identify them as sharks.

**Questions:**

Refer to Weaknesses.

---

> ### Author Response · Authors · 2024-11-20
> **Response 1 Technical novelty and comparison with UAP**
>
> Thank you very much for reviewing our paper and the valuable comments. Please find our response to your questions below:
>
> —
>
> **Q1:** Technical novelty and comparison with UAP
>
> > simply migrates the Universal Adversarial Perturbations generation method in image classification to the VLM task
>
> Adapting an existing method to a novel problem space is a meaningful contribution, and our approach is not a simple migration. As we explained in the introduction, previous studies on Universal Adversarial Perturbations (UAPs) for image classifiers have largely focused on untargeted attacks. Traditional targeted UAPs, on the other hand, are limited to fixed classes. In contrast, our TUAP method for CLIP is capable of targeting any arbitrary text sentence, which significantly expands the capability and complexity of UAP-based attacks.
>
> Furthermore, according to the ICLR Reviewer Guide (2021–2025), novelty should be evaluated not only based on technical methods but also on novel findings. Submissions bring value to the ICLR community when they convincingly demonstrate new, relevant, impactful knowledge. Drawing attention to a new application is also a valuable contribution.
>
> By overcoming the technical challenges of crafting adversarial perturbations based on surrogate ensembles, our work is the first to demonstrate the possibility of targeted UAPs on CLIP models, highlighting an emerging safety issue for large vision-language models. We believe our work offers significant contributions in both respects.
>
> Not every impactful work needs to propose a completely new method, nor does every method have to be completely different from others. We hope the reviewers can re-evaluate the contribution of our work based on the breadth and depth of our experimental analysis, as well as the practical significance of our findings.
>
> >  there is no technical innovation.
>
> We respectfully disagree with the assertion that our work has "no technical innovation." Eq. (2) to (7) are specifically designed for targeted UAP attacks against CLIP models. Our method is distinctive because it enables **targeted UAP** attacks towards any arbitrary text and achieves transferability through surrogate model ensembles, setting it apart from existing attacks.
>
> To the best of our knowledge, there is no existing work on targeted UAP attacks for CLIP in the current literature. Here, we provide the following empirical evidence to show the differences between our method and the original UAP attack [1]. Since UAP is an untargeted attack, it cannot produce the exact text target. Therefore, we compare adversarial accuracy (the lower, the better) to ensure a fair evaluation against UAP. A significant performance difference is observed between our TUAP and UAP, indicating that solving Eq. (2) to (7) in the context of CLIP presents unique technical challenges that we have successfully addressed.
>
> |  Method  |  Vicitim Encoder | CIFAR10 | CIFAR100 |  FOOD101 |  GTSRB  | ImageNet |   Cars   |  STL10  |  SUN397  |
> |:--------:|:----------------:|:-------:|:--------:|:--------:|:-------:|:--------:|:--------:|:-------:|:--------:|
> |    UAP   |   ViT-L OpenAI   |   84.6  |   59.9   |   84.8   |   37.6  |   70.3   |   68.4   |   98.1  |   66.9   |
> | **Ours** |   ViT-L OpenAI   | **0.0** |  **0.0** |  **3.9** | **0.3** | **10.5** |  **5.5** | **1.6** |  **7.3** |
> |    UAP   | ViT-L Commonpool |   88.9  |   64.5   |   89.8   |   45.5  |   74.0   |   91.6   |   98.0  |   72.5   |
> | **Ours** | ViT-L Commonpool | **0.0** |  **0.0** |  **3.8** | **0.0** | **14.1** | **19.3** | **0.6** | **12.3** |
> |    UAP   |    ViT-L CLIPA   |   90.8  |   71.5   |   91.8   |   46.4  |   78.1   |   92.8   |   98.4  |   74.0   |
> | **Ours** |    ViT-L CLIPA   | **0.0** |  **0.0** | **10.5** | **0.0** | **23.1** | **30.0** | **1.1** | **17.7** |
> |    UAP   |   ViT-B SigLIP   |   64.2  |   36.4   |   84.0   |   23.1  |   71.8   |   85.9   |   95.5  |   65.4   |
> | **Ours** |   ViT-B SigLIP   | **0.0** |  **0.0** |  **4.4** | **0.0** | **13.1** | **16.8** | **0.2** |  **9.6** |
> |    UAP   |   ViT-B Laion2B  |   71.9  |   42.5   |   78.2   |   33.2  |   66.1   |   84.1   |   96.2  |   68.8   |
> | **Ours** |   ViT-B Laion2B  | **0.0** |  **0.0** |  **3.1** | **0.3** | **12.7** | **14.2** | **1.2** | **11.8** |
> |    UAP   |    RN50 OpenAI   |   31.3  |   13.8   |   58.0   |   19.2  |   50.2   |   45.3   |   88.4  |   51.1   |
> | **Ours** |    RN50 OpenAI   | **0.0** |  **0.0** |  **3.2** | **0.1** |  **9.6** |  **6.9** | **1.1** | **10.2** |
>
> [1] Moosavi-Dezfooli, Seyed-Mohsen, et al. "Universal adversarial perturbations." CVPR, 2017.

---

> ### Author Response · Authors · 2024-11-20
> **Response 2: response to lack of comparison with Universal Adversarial Attack methods for VLM models [1][2]**
>
> **Q2:** Lack of comparison with Universal Adversarial Attack methods for VLM models [1][2]
>
> **A2:** The reviewer suggested papers that employ sample-specific perturbations; however, these are fundamentally different from universal attacks. Below, we provide both technical and empirical comparisons to clarify the distinctions:
>
> | Method |    Attack objective   | Type of perturbation | Cross-model transferability | Cross-dataset transferability | Cross-task transferability |
> |:------:|:---------------------:|:--------------------:|:------------------------------:|:-------------------------------:|:----------------------------:|
> |   [1]  | Untargeted / targeted |    Sample-specific   |                ✓               |                ✗                |               ✗              |
> |   [2]  | Untargeted / targeted |    Sample-specific   |                ✓               |                ✗                |               ✗              |
> |  Ours  |        Targeted       |       Universal      |                ✓               |                ✓                |               ✓              |
>
> One of the main differences between our work and the studies referenced by the reviewer [1,2] is the type of perturbation used. The works [1,2] utilize sample-specific perturbations that must be generated individually for each image, which can be time-consuming and impractical for large-scale attacks. In contrast, UAPs are generated once and can be applied to any image, making them more suitable for widespread and real-world scenarios.
>
> The reliance on sample-specific perturbations in [1,2] prevents them from achieving cross-dataset transferability because the perturbations are tailored to specific images and cannot be applied to others. Moreover, [1,2] did not demonstrate cross-task transferability in their studies.
>
> In our work, we comprehensively demonstrate that our Targeted Universal Adversarial Perturbation (TUAP) achieves cross-model, cross-dataset, and cross-task transferability simultaneously. This underscores the robustness and practical significance of our approach compared to sample-specific perturbations.
>
> > [1] Zhao Y, Pang T, Du C, et al. On evaluating adversarial robustness of large vision-language models[J]. Advances in Neural Information Processing Systems, 2024, 36.
>
> We followed the same experimental setting as in Table 2 of [1], which computes the CLIP score (higher is better) for the ImageNet validation set. Since we do not know the random 1K image split used by [1], we performed the evaluation on the full validation set using target text sentence No. 8.
>
> We observe that our TUAP outperforms [1] in the same type of modality attack (MF-it). For the best attack modality (MF-ii + MF-tt) in [1], our method outperforms theirs when BLIP2 is the victim model and achieves comparable results when MiniGPT4 is the victim model.
>
> Notably, our TUAP achieves this performance using a single perturbation applied universally to all images, whereas [1] employs sample-specific perturbations tailored to each image. This is a significant result, given that sample-specific perturbations are generally known to be stronger than universal ones.
>
> Additionally, we would like to point out that [1] was published at NeurIPS 2023, not 2024.
>
> |  Method  | VLM Model | Attack modality |   RN50   |   RN101  | ViT-B/16 | ViT-B/32 | ViT-L/14 |
> |:--------:|:---------:|:---------------:|:--------:|:--------:|:--------:|:--------:|:--------:|
> |    [1]   |   BLIP2   |      MF-it      |   49.2   |   47.4   |   52.0   |   54.6   |   38.4   |
> |    [1]   |   BLIP2   |   MF-ii+MF-tt   |   65.6   |   63.3   |   66.5   |   68.1   |   55.5   |
> | **Ours** |   BLIP2   |      MF-it      | **68.0** | **64.1** | **70.8** | **73.2** | **62.1** |
> |    [1]   |  MiniGPT4 |      MF-it      |   47.2   |   45.0   |   46.1   |   48.4   |   44.3   |
> |    [1]   |  MiniGPT4 |   MF-ii+MF-tt   | **63.3** | **61.1** |   63.1   |   66.8   | **61.4** |
> | **Ours** |  MiniGPT4 |      MF-it      |   62.4   |   60.0   | **65.1** | **68.1** |   55.3   |

---

> ### Author Response · Authors · 2024-11-20
> **Response 2 (continued): response to lack of comparison with Universal Adversarial Attack methods for VLM models [1][2]**
>
> > [2] Dong Y, Chen H, Chen J, et al. How Robust is Google's Bard to Adversarial Image Attacks?[J]. arXiv preprint arXiv:2309.11751, 2023.
>
> 1. Comparison of Untargeted Evaluation
>
> We used the adversarial perturbations generated on the NIPS17 dataset released by [2] and compared them with our TUAP applied to the same dataset. We report the adversarial accuracy (the lower, the better) based on the $L_\infty$-norm bounded threat model with target No. 8. The results show that our TUAP outperforms [2] by using a single universal perturbation, whereas [2] uses perturbations specifically designed for each individual image.
>
> | Method | ViT-L OpenAI | ViT-L CommonPool | ViT-L CLIPA | ViT-B SigLIP | ViT-B Laion2B | RN50 OpenAI |
> |:------:|:------------:|:----------------:|:-----------:|:------------:|:-------------:|:-----------:|
> |   [2]  |     10.5     |        9.5       |     33.5    |     17.0     |     15.0      |     10.5    |
> |  Ours  |    **5.0**   |      **8.0**     |   **14.0**  |    **5.5**   |    **7.5**    |   **9.5**   |
>
> ---
>
> 2. Comparison of Targeted Evaluation
>
> We also conducted a comparison in the targeted setting with [2]. Possibly due to the computational expense of generating perturbations for each image, [2] released only 20 adversarial images. We utilized these 20 images and their corresponding text sentences to evaluate using the same task as in Table 1 of our submission. The results indicate that our TUAP significantly outperforms [2] in achieving targeted objectives.
>
> | Method | ViT-L OpenAI | ViT-L CommonPool | ViT-L CLIPA | ViT-B SigLIP | ViT-B Laion2B | RN50 OpenAI |
> |--------|--------------|------------------|-------------|--------------|---------------|-------------|
> | [2]    | 20.0         | 40.0             | 25.0        | 5.0          | 0.0           | 5.0         |
> | Ours   | **86.0**     | **82.0**         | **79.5**    | **89.5**     | **76.0**      | **68.0**    |

---

> ### Author Response · Authors · 2024-11-20
> **Response 3 and 4 on GPT4o-mini and perceptibility**
>
> **Q3:** How does the proposed method perform on closed-source models such as chatgpt4?
>
> **A3:**
> Thanks for the thoughtful suggestion. We have now evaluated our method on GPT4o-mini using the same experimental settings as in Table 2 of our initial submission. Following the approach suggested in [2] by the reviewer, we also report the metric where an attack is considered successful if the keyword from the target text appears in the VLM's response. The results, based on the $L_\infty$-norm bounded threat model with target No. 8, are summarized below. It clearly demonstrates that the ensemble technique we proposed for generating TUAPs is highly effective in achieving targeted attacks against commercial models like ChatGPT. This underscores the practical applicability and robustness of our method, even when applied to closed-source models.
>
> | Method | OpenFlamingo |  LLaVA7B  |  MiniGPT4 |   BLIP2   | GPT4o-mini |
> |:------:|:------------:|:---------:|:---------:|:---------:|:----------:|
> |   E1   |     9.74     |    2.28   |    0.34   |   10.16   |     3.0    |
> |   E16  |   **71.62**  | **66.64** | **45.76** | **72.12** |  **79.00** |
>
> We also summarized the BLEU-4 metric (for targeted attack) in the table below:
>
> | Method | OpenFlamingo |  LLaVA7B | MiniGPT4 |   BLIP2  | GPT4o-mini |
> |:------:|:------------:|:--------:|:--------:|:--------:|:----------:|
> |   E1   |     10.1     |   13.5   |   12.5   |   12.8   |     4.1    |
> |   E16  |   **28.4**   | **28.4** | **23.3** | **27.0** |   **5.4**  |
>
> As an example, we asked GPT4o-mini using the same prompt shown in our Figure 1 (Section 4.4) to describe the image, and the following are the exact responses (the target words are boldfaced):
>
> *“The image shows a surreal landscape with a **bridge** in the foreground and a mountainous area with flowing lava in the background. A **shark** is depicted above the mountains, adding a fantastical element. The combination of lava, the **bridge**, and the **shark** creates a dream-like, otherworldly scene.”*
>
> ---
>
> **Q4:** In Figure 1, there are apparent shark textures in the generated adversarial images
>
> **A4:**
> The perceptibility of the shark textures in the generated adversarial images is due to the $\epsilon$ value used in the $L_\infty$-norm bounded attack. Our default choice of $\epsilon = 16/255$ is standard in black-box adversarial studies. The paper [2] referenced by the reviewer also uses $\epsilon = 16/255$ and even larger perturbations up to $\epsilon = 32/255$.
>
> Regarding why an actual shark pattern appears in the perturbation, we have provided novel insights in Section 4.4 of our submission. The concept blending capability that empowers CLIP's zero-shot generalization also makes it vulnerable to adversarial manipulation, resulting in perceptible patterns related to the target concept.
>
> The trade-off between the perceptibility of the perturbation and the attack success rate (ASR) is thoroughly analyzed in our initial submission. In Section 4.5, Figure 4(a), we present an ablation study on $\epsilon$ and ASR. Lower values of $\epsilon$ lead to better imperceptibility, as shown in the corresponding visualizations in Figure 25. Similar studies for $L_2$ and adversarial patch attacks are presented in Appendix B.10 and Figure 5, with visualizations in Figures 24 and 26.
>
> An attacker could choose a lower value of $\epsilon$ to achieve better imperceptibility at the cost of a slight decrease in ASR. For example, in Figure 25, under $\epsilon = 8/255$, it is difficult to observe any shark texture. This is a common trade-off in adversarial robustness studies.

---

> ### Author Response · Authors · 2024-11-23
>
> Dear Reviewer Tkjh,
>
> This is a kind reminder that we have provided a detailed, point-by-point response to your initial review.
>
> We have provided both technical and empirical comparisons with the related works [1,2] mentioned by the reviewer. These works study different types of perturbations, making direct comparisons inherently challenging. We have added new results in our submission and rebuttal.
>
> It is important to emphasize that comparing universal perturbations (our TUAP) with sample-specific perturbations [1,2] is inherently unfair, as sample-specific perturbations are customized for each individual sample, giving them a significant advantage. Despite this disadvantage, our TUAP consistently outperforms or matches [1,2] across several evaluations, highlighting its novelty in achieving universal adversarial transferability.
>
> We would appreciate it if you could review the newly provided results. Below is a summary:
>
> - Our TUAP demonstrates performance that is either superior to or on par with [1], despite using a single universal perturbation applicable to any sample.
> - Similarly, TUAP outperforms [2] on both untargeted and targeted objectives, again with a single universal perturbation.
> - Our objective functions, as defined in Equations (2)-(7), are specifically tailored for CLIP models. New results indicate that without these modifications, UAP achieves minimal success against CLIP models.
> - We achieve simultaneous black-box cross-model, cross-dataset, and cross-task adversarial transferability. These results meet challenging targeted attack objectives and distinguish our approach from that of existing studies.
>
> We hope these additions and results address your concerns. We are open to further constructive discussions and welcome your insights.

---

> ### Author Response · Authors · 2024-11-24
>
> Dear Reviewer Tkjh,
>
> Thank you once again for taking the time to review our paper. We deeply value your insightful comments and are sincerely grateful for your efforts.
>
> We kindly request you to review our reply to see if it sufficiently addresses your concerns. Your feedback means a lot to us.
>
> Thank you deeply,
> Authors

---

> > ### Comment · Reviewer_Tkjh · 2024-11-24
> > **Official Comment by Reviewer Tkjh**
> >
> > Thanks to the author for the reply. Combined with the comments of other reviewers, I think the novelty of the method proposed in this work is limited. I decided to maintain the original score.

---

### Author Response · Authors · 2024-11-20
**General response**

Our work introduces a previously unexplored adversarial threat that has not been effectively addressed by existing studies. We acknowledge that the lack of empirical comparisons to prior work in the initial submission may have made it difficult for reviewers to fully assess the novelty of our paper. These comparisons were initially omitted due to the fundamentally different threat model explored in our work.

To address this, we have incorporated both technical and empirical comparisons with all related works identified by the reviewers. The performance gap is substantial, often determining whether the attacker’s goal is achieved or not, underscoring the complexity and significance of the new adversarial threat identified in our study.

To enable these comparisons, we relaxed certain aspects of our threat model, including adjustments in **Black-box versus White-box access assumptions**, **Targeted versus Untargeted attack objectives**, and **Sample-specific versus Universal perturbations**. Despite these relaxations, our TUAP (Targeted Universal Adversarial Perturbation) consistently outperforms existing approaches. We have summarized the technical comparisons between our work and all the works mentioned by each reviewer in the table below.

|       Method      | Threat model |    Attack objective   | Perturbation type | Cross-model transferability | Cross-dataset transferbility | Cross-task transferability |
|:-----------------:|:----------------:|:---------------------:|:---------------------:|:-------------------------------:|:-------------------------------:|:-----------------------------:|
|  Bard Attack [1]  |     Black-box    | Untargeted / targeted |    Sample-specific    |                ✓                |                ✗                |               ✗               |
| Attack on VLM [2] |     Black-box    | Untargeted / targeted |    Sample-specific    |                ✓                |                ✗                |               ✗               |
|      SGA [3]      |     Black-box    |       Untargeted      |    Sample-specific    |                ✓                |                ✗                |               ✓               |
|   SA-Attack [4]   |     Black-box    |       Untargeted      |    Sample-specific    |                ✓                |                ✗                |               ✓               |
|    AdvCLIP [5]    |    White-box*    |       Untargeted      |       Universal       |                ✗                |                ✓                |               ✓               |
|        Ours       |     Black-box    |        Targeted       |       Universal       |                ✓                |                ✓                |               ✓               |

---

To the best of our knowledge, the TUAP presented in this work is the only one that is capable of achieving cross-model, cross-dataset, and cross-task adversarial transferability simultaneously, and achieving challenging targeted attack objectives. The targeted attack objective is only successful if the output is the one specified by the attacker. The untargeted attack is successful as long as the output is incorrect. If the targeted attack is successful, then it also means it can achieve untargeted objectives (as long as the attacker does not use the ground truth as the target).

We welcome constructive discussions with each reviewer.

---

### Author Response · Authors · 2024-11-20
**Summary of changes in the revision**

We have made the following revisions to address reviewers' concerns:

- Revised introduction as suggested by reviewer **Ypoi**.
- Added discussion in related work section for papers [3,4] referenced by reviewer **mcuc**.
- A comparison with the SGA attack [3] is added to Appendix B.7, Table 11 to address the concern by reviewer **mcuc**. Results show that even compared with sample-specific perturbations with an untargeted objective, our Targeted UAP still shows a significant advantage in cross-task transferability.
- A comparison with the Bard attack [1] to Appendix B.7, Tables 12 and 13 to address the concern by reviewer **Tkjh**. Our TUAP shows a significant advantage in cross-model transferability over the sample-specific perturbation used by [1].
- A comparison with AdvCLIP [5] has been added in Appendix B.8, Tables 14 and 15, to address the concern of reviewer **Ypoi**. The results reveal a substantial performance gap, which critically determines whether the attacker’s objectives are successfully achieved or not.
- An evaluation of applying TUAP on the medical datasets is added to Appendix B.9, Table 16 to address the concern of reviewer **QeU8**. Results show that TUAP is also effective in achieving the adversarial objective when applied to medical imaging datasets, further demonstrating cross-dataset transferability.

In summary, these newly added results further demonstrate our TUAP can achieve cross-model, cross-dataset, and cross-task adversarial transferability simultaneously.

[1] Dong, Yinpeng, et al. "How Robust is Google's Bard to Adversarial Image Attacks?." arXiv preprint arXiv:2309.11751(2023).\
[2] Zhao, Yunqing, et al. "On evaluating adversarial robustness of large vision-language models." In NeurIPS 2023.\
[3] Lu, Dong, et al. "Set-level guidance attack: Boosting adversarial transferability of vision-language pre-training models." In ICCV 2023.\
[4] He, Bangyan, et al. "Sa-attack: Improving adversarial transferability of vision-language pre-training models via self-augmentation." arXiv preprint arXiv:2312.04913 (2023).\
[5] Zhou, Ziqi, et al. "Advclip: Downstream-agnostic adversarial examples in multimodal contrastive learning." In ACMMM 2023.

---

### Meta-Review · Area_Chair_ZLFD · 2024-12-18

**Metareview:**

This work investigates the vulnerability of CLIP models to Targeted Universal Adversarial Perturbations (TUAPs) in a black-box setting, demonstrating their effectiveness and transferability across datasets, models, and downstream Vision-Language Models (VLMs). However, most of the reviewers raise a concern regarding lack of significant technical novelty, as the core techniques, such as the optimization of cosine embeddings and model ensemble, are commonly used in prior studies on VLP models. The proposed method primarily extends the Universal Adversarial Perturbation (UAP) framework from single-modal architectures to CLIP. Transferring from untargeted to targeted universal attacks involves standard modifications to the objective function and does not provide substantial conceptual advancements. While the experiments are extensive and well-executed, the results fail to introduce new insights or transformative contributions. Despite the rebuttal, the key concerns about the lack of originality and technical contributions remain unresolved. Therefore, we decide not to accept this work based on its current state.

**Additional Comments On Reviewer Discussion:**

Based on the discussion during rebuttal, most reviewers still have the solid concerns regarding the technical novelty and contribution of this work.

---

### Decision · Program_Chairs · 2025-01-22

Reject